



# Seasonal dynamics and annual budget of dissolved inorganic carbon in the northwestern Mediterranean deep convection region

Caroline Ulses[1], Claude Estournel[1], Patrick Marsaleix[1], Karline Soetaert[2], Marine Fourrier[3], Laurent Coppola[3,4], Dominique Lefèvre[5], Franck Touratier[6], Catherine Goyet[6], Véronique Guglielmi[6], Fayçal Kessouri[7], Pierre Testor[8], Xavier Durrieu de Madron[9]

[1]Laboratoire d'Etudes en Géophysique et Océanographie Spatiales (LEGOS), Université de Toulouse, CNES, CNRS, IRD, UPS, Toulouse, France
[2]Department of Estuarine and Delta Systems, NIOZ Royal Netherlands Institute for Sea Research, 4400 AC Yerseke, The Netherlands
[3]Sorbonne Université, CNRS, Laboratoire d'Océanographie de Villefranche (LOV), Villefranche-sur-Mer, France
[4]Sorbonne Université, CNRS, OSU STAMAR, Paris, France
[5]Aix-Marseille Université, Mediterranean Institute of Oceanography (MIO), 13288 Marseille Cedex 9, France
[6] IMAGES_ESPACE-DEV, Université de Perpignan Via Domitia, 52 Avenue Paul Alduy, 66860 Perpignan, France
[7]Southern California Coastal Water Research Project, Costa Mesa, CA, USA
[8]CNRS-Sorbonne Universités (UPMC Univ. Pierre et Marie Curie, Paris 06)-CNRS-IRD-MNHN, UMR 7159, Laboratoire d'Océanographie et de Climatologie (LOCEAN), Institut Pierre Simon Laplace (IPSL), Observatoire Ecce Terra, Paris, France
[9]CEFREM, CNRS-Université de Perpignan, 52 avenue Paul Alduy, 66860, Perpignan, France.

*Correspondence to*: Caroline Ulses (caroline.ulses@legos.obs-mip.fr)

**Abstract.** Deep convection plays a key role in the circulation, thermodynamics and biogeochemical cycles in the Mediterranean Sea, considered as a hotspot of biodiversity and climate change. In the framework of the DEWEX (Dense Water Experiment) project, the seasonal cycle and annual budget of dissolved inorganic carbon in the deep convection area of the northwestern Mediterranean Sea are investigated over the period September 2012-September 2013, using a 3-dimensional coupled physical-biogeochemical-chemical modeling approach. We estimate that the northwestern Mediterranean Sea deep convection region was a moderate sink of $CO_2$ for the atmosphere over the study period. The model results show the reduction of $CO_2$ uptake during deep convection, and its increase during the abrupt spring phytoplankton bloom following the deep convection events. We highlight the dominant role of both biological and physical flows in the annual dissolved inorganic carbon budget. The upper layer of the northwestern deep convection region gained dissolved inorganic carbon through vertical physical supplies and, to a lesser extent, air-sea flux, and lost dissolved inorganic carbon through lateral transport and biological fluxes. The region, covering 2.5 % of the Mediterranean, acted as a source of dissolved inorganic carbon for the surface and intermediate water masses of the western and southern Western Mediterranean Sea and could contribute up to 10 and 20% to the $CO_2$ exchanges with the Eastern Mediterranean Sea and the Atlantic Ocean.





## 1 Introduction

Quantifying the ocean carbon pump and its evolution under ongoing global warming and rising atmospheric $CO_2$ inventory is a challenging issue. Exchanges of $CO_2$ at the air-sea interface result from a complex interplay of chemical, biological and physical processes in the ocean. Physical mechanisms can play a comparable role as biological processes on carbon transfer in the ocean at regional and global scales (Ayers and Lozier, 2012; Lévy et al., 2013; Stukel and Ducklow, 2017). In particular, deep convection regions, such as the Labrador Sea located at high latitudes in the Atlantic Ocean, are considered

large sinks for atmospheric $CO_2$ due to strong cooling and high primary production leading to long, or even persistent, periods of deficit compared to the atmosphere (Takahashi et al., 2002). In these regions, large amounts of atmospheric $CO_2$ captured at the surface and biologically fixed carbon are transferred to the deep ocean during the intense vertical mixing periods (DeGranpre et al., 2006; Körtzinger et al., 2008a). On the other hand, respired organic carbon remaining above the winter mixing depth can be ventilated back to the surface during the following winter (DeGranpre et al., 2006; Körtzinger et

al., 2008a; Palevsky and Nicholson, 2018). Lateral transport, often associated with the restratification of the water column, the dispersion of the newly formed dense water and/or the exchanges with boundary currents, also greatly contributes to the budget of water masses and their biogeochemical contents (Wolf et al., 2018; Koelling et al., 2022).

The northwestern region of the semi-enclosed Mediterranean Sea (Fig. 1) located at mid-latitudes and connected to the Atlantic Ocean through the narrow Gibraltar Strait is one of the regions where deep convection occurs (Mertens and Shott,

1998; Béthoux et al., 2002). In this region, a basin-scale cyclonic gyre is associated with a doming of isopycnals. The density increase, induced in winter in surface waters by cold and dry northerly winds, produces instabilities of the water column leading to convective mixing of surface waters with deeper waters. The interannual variability of the magnitude and spatial extent of the convection process is driven by both the strength of the air-sea heat flux and the preconditioning corresponding to the pre-winter hydrological properties of the water masses (Houpert et al., 2016; Somot et al., 2016; Estournel et al., 2016;

Margirier et al., 2020). With regards to the biogeochemical processes, the region is characterized at the sea surface by a first phytoplankton moderate bloom in fall, interrupted by deep winter mixing, and a secondary abrupt phytoplankton bloom, following deep winter mixing which has supplied inorganic nutrients in the euphotic layer (Severin et al., 2014; Bernardello et al., 2012; Lavigne et al., 2013; Ulses et al., 2016; Kessouri et al., 2017). At the annual scale, the net community production (NCP, defined as the gross primary production minus the community respiration) was found positive leading to

an autotrophic status of the area (Ulses et al., 2016; Coppola et al., 2018). The downward export of organic carbon and its interannual variability have been related to the intensity of the deep convection and the bloom (Heimbürger et al., 2013; Herrmann et al., 2013; Ulses et al., 2016).

Observational studies that have documented the dynamics of the $CO_2$ system in this region mostly focused on the Ligurian Sea, at the EMSO-DYFAMED (European Multidisciplinary Seafloor and water column Observatory-Dynamique des Flux

Atmospheriques en MEDiterranee, 43°25'N, 7°52' E, 2350 m depth) and BOUSSOLE (43°22' N, 7°54' E, 2400 m depth) sites (Hood and Merlivat, 2001; Copin-Montégut and Bégovic, 2002; Bégovic and Copin-Montégut, 2002; Copin-Montégut





et al., 2004; Touratier and Goyet, 2009; Merlivat et al., 2018; Coppola et al., 2020), where the intensity of convection generally remains moderate compared to the Gulf of Lion. These studies showed a pronounced seasonal cycle of $pCO_2$ mostly controlled by the sea surface temperature. However, the thermal effect is counterbalanced in spring by the impact of

phytoplankton growth which leads to DIC (dissolved inorganic carbon) drawdown, and in winter, by intense mixing events which bring up to the surface DIC-enriched deep waters (Hood and Merlivat, 2001; Copin-Montégut et al., 2004). On an annual timescale, the Ligurian Sea was found to be a medium to minor sink for atmospheric $CO_2$ (Hood and Merlivat, 2001; Copin-Montégut et al., 2004; Merlivat et al., 2008). Based on measurements carried out during the CASCADE cruise in 2011, Touratier et al. (2016) complemented those observations from the Ligurian mooring sites, by describing the

distribution of the carbonate system properties in the central region of the deep convection region during and just after the deep convection event. The authors showed a rapid transfer of anthropogenic $CO_2$ to the ocean interior during the convection event and found an excess in $CO_2$ related to the atmosphere.

The observational description of the $CO_2$ system in the region was enriched with characterizations from modeling studies. Mémery et al. (2002), who applied a 1D coupled physical-biogeochemical-chemical model at the EMSO-DYFAMED fixed

site, found that the Ligurian Sea was a weak sink for atmospheric $CO_2$ in the period 1995-1997. Their study underlined the need for $pCO_2$ data in winter when vertical supply of DIC drives $pCO_2$ which is then not well correlated with SST or surface chlorophyll. To estimate the air-sea $CO_2$ fluxes and a carbon budget in the upper layer of the whole Mediterranean Sea, D'Ortenzio et al. (2008) used a 1D coupled physical-biogeochemical-chemical modeling with the assimilation of chlorophyll satellite data in unconnected grid cells with a horizontal resolution of 0.5° and a depth of 300 m. Regarding the northwestern

region, they concluded that the deep convection region is a major sink of atmospheric $CO_2$ in the open sea over the period 1998-2004. At the scale of the western basin, they found that air-sea $CO_2$ fluxes and biology processes both dominate the carbon cycle. Using a 3D high resolution (1/24° horizontal resolution) reanalysis over the whole Mediterranean Sea, Cossarini et al. (2021) also indicated that the northwestern deep convection area is one of the open sea regions, with the southern Adriatic and Aegean seas, characterized by maximum atmospheric $CO_2$ uptake, over the period 1999-2019.

The northwestern deep convection region where the Western Mediterranean Deep Water is formed plays a crucial role in the circulation and ventilation of the Mediterranean Sea (Schroeder et al., 2016; Li and Tanhua, 2020; Mavropoulou et al., 2020). Yet previous observational and modeling studies addressing the dynamics of $CO_2$ and carbon budget in this region were restricted to fixed sites, generally characterized by moderate convection, or limited at periods of a few weeks, or neglecting or considering only implicitly lateral advection, or extended to the whole western basin. The 3D dynamics of the

$CO_2$ system and an inorganic carbon budget have never been specifically explored and quantified for the whole northwestern convection region, nor their exchanges with the surrounding regions.

The aim of the DEWEX (Dense Water Experiment) project (Conan et al., 2018; Testor et al., 2018) was to investigate the deep convection process and its impact on biogeochemical fluxes based on observational platforms and numerical models. In this framework, two research cruises were carried out in winter and spring of the year 2012/13, completing the MOOSE-GE

(Mediterranean Ocean Observing System for the Environment-Grande Echelle) observational effort performed each year





during stratified periods since 2010. Due to extremely strong buoyancy loss, the 2012/13 winter was characterized by intense deep convection events, and is considered to be one of the five most intense events over the period 1980/2013 (Somot et al., 2016; Herrmann et al., 2017; Coppola et al., 2018). Using a 3D coupled physical-biogeochemical simulation, Kessouri et al. (2018) estimated that the deep convection region was characterized over the period 2012/2013 by a net community

production and showed higher rates of export of organic carbon below the euphotic layer compared to the surroundings. They suggested that due to high spatial and interannual variability, and dispersion of newly formed dense water to the southern Mediterranean Sea, a fraction of the exported carbon escapes a return into the surface layer during the following winter. Here we took benefit from the in situ measurements from the DEWEX project, the MOOSE-GE program, and the BOUSSOLE and EMSO-DYFAMED fixed sites to implement and constrain a model of the dynamics of the $CO_2$ system and

complete the first 3D coupled physical-biogeochemical modeling study on organic carbon by Kessouri et al. (2018). We (i) examined the seasonal cycle of DIC, (ii) estimated an annual carbon budget, and (iii) analyzed and quantified the contribution from air-sea $CO_2$ exchanges, biogeochemical and physical processes to the carbon budget.

## 2 Material and methods

### 2.1 The numerical model

#### 2.1.1 The coupled hydrodynamic-biogeochemical-chemical model

The biogeochemical model Eco3M-S (Auger et al., 2011; Ulses et al., 2021) was forced offline by daily outputs (current velocities, turbulent diffusion coefficient, temperature, and salinity) of the 3D hydrodynamic model SYMPHONIE (Marsaleix et al., 2008). The SYMPHONIE model, a 3D primitive equation model, with a free surface and generalized sigma vertical coordinate, has been used to investigate open-sea convection (Herrmann et al., 2008; Estournel et al., 2016; Damien

et al., 2017) and circulation in the northwestern Mediterranean Sea (Estournel et al., 2003; Ulses et al., 2008; Bouffard et al., 2008).

The biogeochemical model Eco3M-S is a multi-nutrient and multi-plankton functional type model that simulates the dynamics of the pelagic planktonic ecosystem and the cycles of several biogenic elements (carbon, nitrogen, phosphorus, silicon, oxygen) (Auger et al., 2011; Many et al., 2021; Ulses et al., 2021). The model has been used to study

biogeochemical processes in the NW (northwestern) Mediterranean deep convection area (Herrmann et al., 2013; Auger et al., 2014; Ulses et al., 2016; 2021; Kessouri et al., 2017; 2018).

To investigate the dynamics of the $CO_2$ system, the model was extended by implementing the carbonate chemistry model developed and described in detail by Soetaert et al. (2007) and applied by Raick-Blum (2005) in 1D in the northwestern Mediterranean Sea. The food-web structure of the upgraded model and the biogeochemical processes interacting between

compartments are schematically represented in Fig. 2. Two state variables were added in the upgraded version of the coupled model. The first added variable is the dissolved inorganic carbon concentration, the sum of the concentrations of the four




carbon dioxide forms, dissolved carbonate dioxide, bicarbonate, carbonate ion and carbonic acid. The rate of change of the concentration of DIC due to biogeochemical processes is governed by the following equation:

$$\frac{\partial DIC}{\partial t}|bio = \sum_{i=1}^{3}(-GPP_i + RespPhy_i) + \sum_{j=1}^{3} RespZoo_j + RespBac \tag{1}$$

where $GPP_i$ and $RespPhy_i$ are gross primary production and respiration flux, respectively, for phytoplankton size class i (size classes 1, 2, and 3 are pico-, nano-, and micro-phytoplankton, respectively, Fig. 2); $RespZoo_j$ is respiration flux for zooplankton size class j (size classes 1, 2, and 3 are nano-, micro-, and meso-zooplankton, respectively, Fig. 2), $RespBac$ is bacterial respiration. The second added state variable is the "excess negative charge" (denoted $\sum[-]$), which is the moles of negative charges over positive charges of the acid-base system (Table 2 in Soetaert et al. (2007)). As in Soetaert et al. (2007)

we use this excess negative charge instead of the total alkalinity, commonly measured for proton balance. Here we assume that uptake of ions is compensated by uptake or release of protons (electroneutrality), and that $\sum[-]$ is not impacted by changes in the concentrations of nitrate, phosphate, or ammonia/ammonium, which is not the case of total alkalinity. The total alkalinity is then deduced from $\sum[-]$:

$$TA = \sum[-] + \sum NH_3 - \sum NO_3 - \sum PO_4 \tag{2}$$

In this first study on DIC dynamics, we neglected calcium carbonate (CaCO₃) precipitation and dissolution. Schneider et al. (2007) indicated that the Mediterranean Sea is supersaturated with respect to calcite and aragonite and that calcium carbonate dissolution is thus not favored thermodynamically. The present knowledge on CaCO₃ precipitation makes it difficult to parametrize this term in a model (Aumont et al., 2005). We are aware that future refinements will have to take it into account as it could lead notably to an underestimation of air-to-sea CO₂ flux. Sensitivity tests to this term were

performed (see Sect. 2.1.4) and are presented in Sect. 5. In this study, in the carbonate chemistry model the dissociation equilibriums of carbonates, water, ammonium, phosphate, silicate, and borate were taken into account. The thermodynamic equilibrium constants of the carbonate system were calculated as a function of temperature, salinity, and pressure as in Millero (1995) with typographical correction from the CO2SYS program (Lewis and Wallace, 1998). In particular, carbonic acid dissociation constants are calculated as Mehrbach et al. (1973) constants as refit by Dickson and Millero (1987).

The flux of CO₂ at the air-sea interface, $CO_2flux$, was calculated using the following equation:

$$CO_2flux = \rho K_0 K_w (pCO_{2,atm} - pCO_{2,sea}) \tag{3}$$

where $pCO_{2,atm}$ and $pCO_{2,sea}$ (in µatm) are the atmospheric and sea surface partial pressure of CO₂, respectively, $K_0$ (in mol kg⁻¹ atm⁻¹) is the solubility coefficient, $K_w$ (in m s⁻¹) the gas transfer velocity and $\rho$ the sea surface density (in kg m⁻³). We calculated the solubility coefficient according to Weiss (1974) and the gas transfer velocity using the most often used

parameterization of Wanninkhof et al. (1992), with a quadratic dependency to the wind speed 10 m above the sea. In





addition, we performed sensitivity analyses using eight various parameterizations of the gas transfer velocity to estimate uncertainties of air-sea exchanges (see Sect. 2.1.4).

**2.1.2 Model setup**

The numerical domain covers most of the Western Mediterranean Sea, using a curvilinear grid (Bentsen et al., 1999) with a
horizontal resolution varying from 0.8 km in the north to 1.4 km in the south, and 40 vertical levels (Ulses et al., 2021). The implementation of the hydrodynamic simulation and the strategy of downscaling from the Mediterranean Basin to the western sub-basin scale in three stages have been described by Estournel et al. (2016) and Kessouri et al (2017). In a first step, the SYMPHONIE hydrodynamic model, implemented over the Western Mediterranean Sea (delimited by blue lines in the insert of Fig. 1), was initialized and forced at its lateral boundaries with daily hydrodynamic analyses of the
configuration PSY2V4R4, based on the NEMO ocean model at a resolution of 1/12° over the Mediterranean Sea by the Mercator Ocean International operational system (Lellouche et al., 2013). In a second step, the biogeochemical model was computed, in offline mode, at the Mediterranean basin scale, on the same 1/12° NEMO grid (delimited by orange lines in the insert of Fig. 1), using the same NEMO hydrodynamic fields as those used by the SYMPHONIE simulation in step 1. In the third step, the Eco3M-S biogeochemical model was implemented over the Western Mediterranean Sea, using the grid and
the hydrodynamics fields of the aforementioned SYMPHONIE simulation in offline mode. The initial state and lateral boundary conditions of the biogeochemical fields are provided by the biogeochemical simulation of the Mediterranean Basin of step 2. This nesting protocol ensures the coherence of the physical and biogeochemical fields at the open boundaries of the Western Mediterranean model. The carbonate system module of the basin configuration of the biogeochemical model was initialized in summer 2011, using mean values of dissolved inorganic carbon and total alkalinity observations carried out
in 2011 from the Meteor M84/3 (Alvarez et al., 2014), CASCADE (CAscading, Surge, Convection, Advection and Downwelling Events, Touratier et al., 2016), and MOOSE-GE cruises (Testor et al., 2010) and at the EMSO-DYFAMED mooring (Coppola et al., 2021) and BOUSSOLE buoy (Golbol et al., 2020) sites, over bio-regions defined in Kessouri (2015), based on Lavezza et al. (2011). Recently, Davis and Goyet (2021) showed a rigorous mathematical approach based upon the property variability, to precisely quantify the uncertainties at any point of an interpolated data field. This approach
could be used in the near-future to improve both, the at-sea sampling strategy (Guglielmi et al., 2022a; 2022b), and the accuracy of model initialization. The regional biogeochemical simulation started in August 2012. At the river mouths, we prescribed the mean DIC concentration measured by Sempéré et al. (2000) for the Rhone River and climatological values according to Ludwig et al. (2010) and Schneider et al. (2007) at the other river mouths. To compute the gas transfer velocity, we used the 3-hour wind speed provided by the ECMWF model on a 1/8° grid, in consistency with the hydrodynamic
simulation. The atmospheric $pCO_{2,atm}$ was deduced from the flask-air measurements of mole fraction, measured monthly at the Lampedusa site (World Data Centre for Greenhouse Gases: https://gaw.kishou.go.jp/, Lan et al., 2022).





### 2.1.3 Study area and computation of DIC balance

We computed DIC flows and variation in the DIC inventory for the whole deep convection area. The deep convection area was defined as the area that includes the model grid points where the mixed layer depth exceeded 1000 m at least during 1

day of the study period based on Kessouri et al. (2017; 2018). This area covers 70100 km$^2$. The budget was calculated for 2 vertical layers: the photic upper layer, where the photosynthesis process takes place, and the aphotic deeper layer. The base of the upper layer was set at 150 m based on the regional minimum value of diffuse attenuation coefficient of light at 490 nm derived from satellite observations (http://marine.copernicus.eu/, products: OCEANCOLOUR_MED_OPTICS_L3_REP_-OBSERVATIONS_009_095), and following the studies by Lazzari et al. (2012) and Kessouri et al. (2018).

The biological term of the budget was defined as the sum of DIC production through respiration by living organisms, and of DIC consumption through photosynthesis. The physical term was decomposed into two transport terms: a net lateral transport at the limit of the area (positive values correspond to fluxes towards the deep convection zone) and a net vertical downward transport at the base of the upper layer, at 150 m depth. The internal variation, air-sea flux, biological term and lateral physical term were calculated online, while the vertical transport, including advection and mixing, was calculated as

the residual based on values of all other terms. The balance equation for the upper layer is given in Supplementary Material (Text S1).

### 2.1.4 Sensitivity tests

We performed various sensitivity tests to estimate the uncertainties of the modeled air-sea $CO_2$ flux. A first set of tests was based on the parametrization of the gas transfer coefficient. For these tests, we used quadratic (Wanninkhof, 2014), cubic

(Wanninkhof and McGillis, 1999) and hybrid (Liss and Merlivat, 1986; Nightingale et al., 2000; Wanninkhof et al., 2009) wind speed dependency parameterizations of diffusive flux, as well as parameterizations explicitly including air-sea fluxes due to bubble formation (Woolf, 1997; Stanley et al 2009; Liang et al. 2013). In the second set of sensitivity tests we prescribed the atmospheric mole fraction by adding and subtracting an associated uncertainty of 3 ppm due to spatial variabilities (Keraghel et al., 2020). Finally, in a third set of sensitivity tests, we performed 2 simulations by adding simple

estimates of the calcium carbonate production in the equation of DIC biological dynamics. Following the study of Palevsky and Quay (2017), we first estimated it based on PIC:POC ratio and NCP. Miquel et al. (2011) estimated the ratio PIC:POC to 0.5 at 200 m depth based on sediment trap measurements at the EMSO-DYFAMED site. Besides, Kessouri et al. (2018) estimated that POC export represents ~70 % of the total OC (TOC) export (the remaining 30% being attributed to DOC export). Thus, if we assume the ratio of calcium carbonate production to NCP is close to PIC:TOC, we added a consumption

term representing 35% of NCP in Eq. 1. In a second sub-test, we added a $CaCO_3$ production term based on the parametrization used in the Gulf of Lion's shelf modeling study by Lajaunie-Salla et al. (2021).





## 2.2 Data used for model skill assessment

### 2.2.1 BOUSSOLE buoy and EMSO-DYFAMED mooring site observations

To assess the time evolution of surface sea properties, we used high frequency temperature, salinity, and $pCO_2$ data collected at 3 m depth at the BOUSSOLE mooring site (43◦ 22' N, 7°54' E, depth: ~2400 m, green star in Fig. 1), in the Ligurian Sea, in 2013 (Antoine et al., 2006; Merlivat et al., 2018). Temperature and salinity were measured using a Seabird SBE 37-SMP MicroCat instrument. The sensors were cross-calibrated before and after each mooring deployment with the ship CTD, by performing a high temporal resolution sampling cast with 30 min long time series at the fixed depths of 300 and 1000 m.

This allows for high accuracy of 0.001°C in temperature and 0.005 in salinity (Houpert, 2013). $fCO_2$ measurements were monitored using a CARIOCA sensor whose accuracy is estimated at 2 µatm. A detailed description of these data is given in Merlivat et al. (2018).

We also used monthly vertical profiles of temperature, salinity, dissolved oxygen, dissolved inorganic carbon, and total alkalinity collected from September 2012 to September 2013 at the EMSO-DYFAMED site (43°25' N; 7°52' E; depth: 2350

m, black star in Fig. 1) (Coppola et al., 2020), located 5 km from the BOUSSOLE site. Note that temperature and salinity were collected using a Seabird SBE911. Dissolved oxygen measurements were performed using Winkler titration at each CTD cast and were used to correct the SBE43 sensor data by adjusting the calibration coefficients (Coppola et al., 2018). DIC and total alkalinity were measured via potentiometric titration following the methods described by Edmond (1970) and DOE (1994) with an accuracy estimated between 1.5 and 3 µmol kg$^{-1}$. They were analyzed by the SNAPO-CO$_2$ national

service (Service National d'Analyse des Paramètres Océaniques du CO$_2$). $pCO_2$ and $pH_T$ (pH at total scale) were deduced from total alkalinity and total inorganic carbon using the carbonate system CO2SYS program (Lewis et Wallace, 1998; Heuven et al., 2011), as in the system carbonate module described in Sect. 2.1.1.

### 2.2.2 DEWEX and MOOSE-GE cruise observations

To assess the horizontal and vertical distribution of the simulated DIC concentration, we used in situ observations collected

during two cruises carried out in the framework of the DEWEX project on-board the RV *Le Suroît*. The first cruise, DEWEX Leg1, was carried out in February 2013, during the active phase of deep convection (Testor, 2013), and the second one, DEWEX Leg2, in April 2013, during the following spring bloom (Conan, 2013). In addition, we used observations from the 2013 MOOSE-GE cruise, conducted during the stratified, oligotrophic season, in June–July 2013, on-board RV *Tethys II* (Testor et al., 2013). During the three cruises, the total dissolved inorganic carbon measurements (DEWEX Leg1: 19

stations, DEWEX Leg2: 14 stations, MOOSE-GE: 20 stations) were collected into acid-washed 500 cm$^3$ borosilicate glass bottles and poisoned with 100 mm$^3$ of HgCl$_2$, following the recommendations of DOE (1994) and Dickson et al. (2007). Samples were stored in the dark at 4 °C pending analysis. Analyses were also performed by the national service SNAPO-CO2. Following Wimart-Rousseau et al. (2021), the values of total dissolved inorganic carbon and total alkalinity below 500





m depth, outside the range defined by ± 2 the standard deviation to the mean value, were not considered. The accuracy of the

measurements was estimated at 1.5-3 µmol kg$^{-1}$.

**2.3 CANYON-MED neural networks**

Since in situ observations of the carbonate system remain scarce, the comparison of model outputs versus in situ observations is completed with a comparison with dissolved inorganic carbon, total alkalinity, and pH$_T$ derived from the CANYON-MED neural networks developed by Fourrier et al. (2020) for the Mediterranean Sea. pCO$_2$ was derived from

CANYON-MED outputs using the carbonate system CO2SYS program. The neural networks were applied at the EMSO-DYFAMED and BOUSSOLE sites using as input parameters pressure, temperature, salinity, and dissolved oxygen measured between 3 and 14 m depth, as well as the geolocation and date of the sampling. The accuracy of the derived pH$_T$, dissolved inorganic carbon and total alkalinity was estimated at 0.014, 12 µmol kg$^{-1}$ and 13 µmol kg$^{-1}$, respectively, for the entire Mediterranean Sea area. However, the accuracy of the neural networks was greatly improved locally, as Fourrier et al.

(2022) showed specifically for the Gulf of Lion and Ligurian Sea over the period 2013-2020.

**3 Assessment of the model skills**

**3.1 Comparison at the BOUSSOLE and EMSO-DYFAMED sites**

Figure 3 shows the seasonal cycle of temperature, salinity, pCO$_2$, DIC, total alkalinity, and pH$_T$ observed and modeled at the surface, at the EMSO-DYFAMED and BOUSSOLE sites. The temperature is very well simulated, with a highly significant

correlation of 0.997 (p-value < 0.01), a RMSE (Root Mean Square Error) of 0.50 °C, and a bias of -0.31 °C, compared to BOUSSOLE buoy observations (Fig. 3a). Regarding the salinity, the model is close to the observations, except from December to February when it underestimates the BOUSSOLE observations (Fig. 3b). The correlation (R = 0.59, p-value < 0.01) remains significant compared to the observations at the buoy, where the model has a bias of -0.04 and a RMSE of 0.10 over the whole study period. The modeled pCO$_2$ is in good agreement with observations and values derived with CANYON-

MED neural networks, with a significant correlation of 0.90 (p-value < 0.01), a bias of 5.74 µatm and a RMSE of 25.57 µatm, compared to the BOUSSOLE observations (Fig. 3c). The model simulates low values in winter, when temperatures were minimum, and in spring, during the phytoplankton bloom identified by Kessouri et al. (2018). The maximum values are modeled in summer due to warming, as in observations and in CANYON-MED results. The seasonal variation of modeled DIC is in agreement with those observed and deduced from CANYON-MED (Fig. 3d), showing an increase in winter until

the end of the deep mixing period and a drop in spring when the growth of phytoplankton was maximum. The seasonal dynamics of modeled alkalinity shows minimum values in November/December, an increase in winter and low variations in spring (Fig. 3e). The increase in winter is also found in observations and CANYON-MED results. In summer the model underestimates both datasets by ~10-15 µmol kg$^{-1}$. The pH$_T$ seasonal variation in observations, simulation and CANYON-





MED results all indicate a drop in summer, period of oligotrophy, high stratification, and domination of respiration over

photosynthesis according to the study of Kessouri et al. (2018). Finally, the model results for the variables of the carbonate system are also consistent with the seasonal variability derived by Coppola et al. (2020) from monthly mean DYFAMED observations over the period 1998-2016. The modeled variables fall within the range of the observed values gathered in this synthesis study: 300-570 µatm for $pCO_2$, 2200-2340 µmol kg$^{-1}$ for DIC, 2510-2600 µmol kg$^{-1}$ for alkalinity and 7.9-8.2 for $pH_T$.

**3.2 Comparison with DEWEX and MOOSE-GE cruise observations**

Previous studies based on the present coupled model concluded that the model shows good performance in reproducing fall and winter mixing (Estournel et al., 2016), as well as the timing and intensity of the phytoplankton blooms (Kessouri et al., 2018), the seasonal dynamics of the dissolved oxygen (Ulses et al., 2021) and inorganic nutrients (Kessouri et al., 2017) over the three cruise periods. Here, we focus the assessment of the model skills on the seasonal dynamics and spatial variability of

the carbonate system, especially of the DIC concentration.

A comparison of modeled surface (from 5 to 10 m depth) DIC concentration with DEWEX Leg1, DEWEX Leg2, and MOOSE-GE cruise observations is shown in Fig. 4. Figure 5 shows the modeled and observed DIC vertical profiles in the deep convection area (indicated in Fig. 1 and defined in Sect. 2.1.3) and south of this zone (latitude < 41°N), in the Balearic Front, where winter vertical mixing is shallower (Ulses et al., 2021), during the 3 cruise periods. The statistical analysis for

surface DIC concentrations indicates significant spatial correlations of 0.78, 0.67 and 0.54 (p-value < 0.01), a RMSE of 18.71, 24.25 and 12.04 µmol kg$^{-1}$ and a bias of 5.25, 15.27 and -1.94 µmol kg$^{-1}$, compared, respectively, to DEWEX Leg1, DEWEX Leg2 and MOOSE-GE observations. The model correctly represents observed spatial variability. In winter, during the intense vertical mixing period, maximum values near the sea surface are found in the deep convection zone (Fig. 4a-b) where the vertical profiles are almost homogeneous over the whole water column (Fig. 5a). In spring, the model represents

the drops observed in the surface layer in both zones (Fig. 4c-d and Fig. 5b) due to phytoplankton growth (Kessouri et al., 2018). Finally, the model reproduces the low values observed at the surface in the deep convection zone during the stratified period where the vertical profiles approach those observed in the southern zone (Fig. 4e-f and 5c). The statistical analysis based on the whole vertical profiles shows that the model is significantly correlated with the observations (R > 0.7, p-value < 0.01), has a RMSE smaller than 20 µmol kg$^{-1}$ and a standard deviation smaller 25 µmol kg$^{-1}$ and close to observations

(bottom panel in Fig. 5a-c).





## 4 Results

### 4.1 Seasonal cycle of dissolved inorganic carbon

We analyze here the seasonal cycle of the modeled dissolved inorganic carbon, over the period September 2012-September 2013. Figure 6 shows the time evolution of atmospheric and hydrodynamic conditions as well as of surface $pCO_2$ and DIC flows, while Fig. 7 displays the cumulative DIC flows and the resulting change in DIC inventory for the upper (surface-150 m) and deeper (150 m-bottom) layers. Figures 8 and 9 show maps of the seasonal mean $pCO_2$ difference between the atmosphere and surface seawater ($pCO_{2,atm}$ -$pCO_{2,sea}$) and air-to-sea $CO_2$ flux, respectively. Finally, the time evolution of the

DIC concentration profile averaged over the deep convection area is shown in Fig. 10.

The study year was divided into seasonal periods defined according to the timing of stratification and biogeochemical processes, specific to this year, according to the studies of Kessouri et al. (2017; 2018).

**Autumn** (1 September- 27 November, 88 days) - After a period of alternative heat gain and loss events, and from the intense

northerly wind event occurring at the end of October (spatial mean heat loss reached 1000 W m$^{-2}$), the deep convection area was continuously transferring heat to the atmosphere (Fig. 6a and 6e). The sea heat loss induced drops of surface temperature (Fig. 6c) and vertical mixing with a mixed layer that, on average, was shallower than 50 m during the whole autumn period (Fig. 6b).

Over this whole autumn period, sea surface $pCO_2$ decreased with temperature (Fig. 6c and 6d, coefficient correlation of 0.99

with p-value < 0.01). The air-sea $CO_2$ flux displayed strong outgassing peaks exceeding -20 mmol C m$^{-2}$ day$^{-1}$ during the northerly wind events occurring early and mid-September (Fig. 6f), when the $pCO_2$ difference was greater than 60 µatm (Fig. 6d). During the intense event of heat loss and cooling at the end of October, the deep convection area became in deficit compared to the atmosphere (sea surface $pCO_2$ smaller than atmospheric $pCO_2$) (Fig. 6d) and started to absorb atmospheric $CO_2$ (Fig. 6f). The air-sea flux displayed ingassing peaks smaller than 6 mmol C m$^{-2}$ day$^{-1}$ in November (Fig. 6f),

characterized by $pCO_2$ differences smaller than 30 µatm (Fig. 6d) and moderate wind speeds (Fig. 6e). Considering the whole autumn period (88 days), the deep convection area was a weak source of $CO_2$ for the atmosphere with a cumulative air-sea flux that amounted to -0.19 mol C m$^{-2}$ (Fig. 7a-b). Characterized by low temperature, the deep convection area, outside its south-western region, showed small averaged $pCO_2$ differences (< 25 µatm, Fig. 8a) and weak mean outgassing fluxes compared to the surrounding open-sea (Fig. 9a). In the southwestern region, intrusions of warm waters from the

Balearic Sea characterized by higher $pCO_2$, associated with strong wind speeds (not shown), favored maximum outgassing fluxes.

Regarding biogeochemical processes, the shallowing of the nutricline and vertical mixing events induced nutrient supplies into the upper layer that favored primary production and growth of phytoplankton near the surface from the end of October (Kessouri et al., 2017; 2018 and Fig. 10a). This led notably to a temporal decrease in DIC concentration into the mixed layer





end of October and November (Fig. 10b). However, over the whole fall period, respiration dominated primary production (Kessouri et al., 2018) (Fig. 6h), yielding a net production of DIC of 0.56 mol C m$^{-2}$ and 0.30 mol C m$^{-2}$ in the upper (surface-150 m) and deeper (150 m-bottom) layers, respectively (Fig. 7).

The net physical fluxes of DIC at the deep convection area boundaries were fluctuating between -150 and 150 mmol C m$^{-2}$ day$^{-1}$ in the upper and deeper layers (Fig. 6g). More specifically, our model results show a net upward transport of DIC of 35.08 mol C m$^{-2}$ into the upper layer of the area while the lateral transport led to a loss of DIC of 34.59 mol C m$^{-2}$ in the upper layer and a gain of DIC of 33.70 mol C m$^{-2}$ in the deeper layer of the deep convection area (not shown). This is probably induced by the upwelling and the surface divergence associated with the dynamics of the cyclonic gyre (Estournel et al., 2016). The lateral and vertical physical transfers at the boundaries resulted in a net increase in DIC inventory in the upper layer of 0.49 mol C m$^{-2}$ and a net decrease in DIC inventory in the deeper layer of -1.38 mol C m$^{-2}$ (Fig. 7).

Finally, the inventory of DIC changed by 0.85 and -1.08 mol C m$^{-2}$ in the upper and deeper layers, respectively (Fig. 7). The upper layer gained DIC through biogeochemical and physical processes and lost DIC through outgassing to the atmosphere.

**Winter** (28 November - 23 March, 116 days) **-** The winter period can be further divided into two sub-periods based on the magnitude of vertical mixing (Kessouri et al., 2017).

*Winter sub-period 1:* During the first winter period (end of November - mid-January), heat loss events induced an intensification of vertical mixing that remained moderate with the spatially averaged mixed layer depth above the euphotic layer depth (150 m) (Fig. 6a, 6b and 10). Vertical mixing induced new supplies of inorganic nutrients into the upper layer supporting primary production near the surface (Kessouri et al., 2017; 2018). From mid-December a net consumption of DIC is modeled in the whole upper layer (Fig. 6h). The cumulative biogeochemical fluxes in the upper layer showed a progressive decrease during the first winter period leading to a net weak consumption of DIC of 0.05 mol C m$^{-2}$ (Fig. 7a).

Until mid-January, sea surface pCO$_2$ continued to decrease with temperature, yielding a reinforcement of the pCO$_2$ difference from 30 to 40 μatm (Fig. 6c and 6d). The spatial mean air-sea CO$_2$ flux reached 15 mmol C m$^{-2}$ day$^{-1}$ during the northerly wind events (Fig. 6e and 6f). Over the first winter period the deep convection area absorbed 0.35 mol m$^{-2}$ of atmospheric CO$_2$ (Fig. 7).

The physical fluxes at the limit of the upper layer of the deep convection area showed similar patterns as during autumn, with an upward flux of DIC into the upper layer of 41.40 mol C m$^{-2}$, almost counterbalanced by a lateral outflow of DIC of 40.44 mol C m$^{-2}$ in the upper layer and a lateral inflow of DIC of 39.90 mol C m$^{-2}$ in the deeper layer. The net physical fluxes led to a gain of 0.97 mol C m$^{-2}$ in DIC inventory of the upper layer and a loss of DIC of 1.51 mol C m$^{-2}$ in the deeper layer (Fig. 7). Globally, the euphotic layer showed an increase in the DIC inventory of 1.27 mol C m$^{-2}$, resulting from a gain through air-sea fluxes and physical supplies, and a weak net biological flux (Fig. 7).

*Winter sub-period 2:* The second winter period, which began in mid-January, was characterized by deep convection. The mixed layer deepened strongly during the intense heat loss events that occurred until the end of February (Fig. 6a-b). After a





16 day pause during which surface restratification caused the mixed layer to be shallower, a new northerly wind generated a

secondary deep convection event in late March. During the second winter period (mid-January/end of March), vertical mixing reached deep water masses (Fig. 6b). Surface temperature remained relatively constant, at a value of ~12.9 °C close to deep water temperature (Fig. 6c). Vertical and horizontal exchanged transports showed same patterns as during the preconditioning period (fall and first period of winter) but the difference between vertical and horizontal transfers became more pronounced leading to net physical fluxes exceeding 110 mmol C m$^{-2}$ day$^{-1}$ in the upper layer and -160 mmol C m$^{-2}$

day$^{-1}$ in the deeper layer, during the four northerly wind events. The physical fluxes integrated over the second winter period reached 3.48 and -5.36 mol C m$^{-2}$ in the upper and deeper layers, respectively.

Regarding the biogeochemical fluxes, the net consumption of DIC progressively increased in the upper layer with the decrease in heterotrophic respiration and the moderate increase in primary production rates (Kessouri et al., 2018). It accelerated when vertical mixing ceased in mid-March and remained high until the end of the period. The cumulative

biological flux reached -1.49 mol C m$^{-2}$ over this sub-period.

Despite the biological consumption of DIC, a progressive increase in DIC concentration in the upper layer is clearly visible on Fig. 10b due to vertical transport. The upward fluxes led to an increase in sea surface pCO$_2$ showing values on average close to equilibrium (Fig. 6d). The pCO$_2$ difference decreased and, despite intense wind events, air-sea flux peaks remained lower than 12 mmol C m$^{-2}$ day$^{-1}$ and finally cumulative air-sea flux reached 0.28 mol C m$^{-2}$ over the second winter sub-

period (a lower value than over the first winter period).

To summarize, the upper layer showed a gain in DIC inventory through vertical transport and, to a lesser extent, uptake of atmospheric CO$_2$, while it lost DIC through lateral transport and net biological processes (Fig. 7a).

*All winter period:* The pCO$_2$ difference and air-sea fluxes averaged over the whole winter period (end November-end March,

Fig. 8b and 9b, respectively) integrate various processes: (1) Mistral and Tramontane northerly winds blowing on average over a northwest/southeast axis over the Gulf of Lion (not shown) intensified air-sea fluxes on this axis, (2) low sea surface temperature in the deep convection region favored an amplification of the pCO$_2$ difference and maximum air-sea fluxes during the first winter period, especially at the northern edge of the convection zone (Fig. S1a and S1c), while (3) high surface DIC concentrations in the regions of intense vertical mixing generated a reduction of pCO$_2$ difference and air-sea

fluxes, especially during the second winter period (Fig. S1b and S1d). The maxima of the lateral DIC transport in the upper layer of the water column averaged over the whole winter period are found in the general circulation, especially in the Northern Current, the Balearic Current and the Balearic Front, separating the southern less salty Atlantic waters from the deep convection salty waters (Fig. 11a and 11c). The instabilities developing at the periphery of the deep convection area favored the incorporation of saltier and DIC-enriched waters in the general circulation through a bleeding effect, similarly as

described by Herrmann et al. (2008) for the export of newly-formed dense waters from the deep convection area, (i) at the western boundaries of the deep convection area towards the Balearic Sea, and towards the Algerian basin by the southern extension of the Balearic Current, as well as (ii) along the Balearic Front between the Minorca Balearic Island and Corsica,





as illustrated in Fig. 11c and 11d. Finally, Figure 11b shows that the vertical DIC supply into the upper layer during winter resulted from upward and downward vertical fluxes of small scales due to the absence of stratification.


**Spring** (24 March - 5 June, 74 days) - From the end of March, total surface heat flux remained mostly positive (Fig. 6a). The water column rapidly stratified with a mixed layer thickness lower than 50 m (Fig. 6b). The cessation of deep vertical mixing favored the onset of a spring phytoplankton bloom with a peak of primary production and phytoplankton concentration at the surface mid-April (Fig. 10a; Kessouri et al., 2018), and leading to a sustained consumption of DIC (Fig. 6h, 7a, and 10b).

From mid-April, the near-surface layer became depleted in nutrients, a deep chlorophyll maximum (DCM) formed and progressively deepened (Fig. 10a; Kessouri et al., 2018), the consumption of DIC was slowed down in the upper layer (Fig. 6h and 7a). In the deeper layer the rate of DIC production through heterotrophic remineralization of organic carbon, exported during the deep convection and the bloom, was maximum during this period (Fig. 6h). The contribution of biological processes on the DIC inventory over the spring period resulted in a loss of 2.19 mol C m$^{-2}$ in the upper layer and a gain of

DIC of 0.91 mol C m$^{-2}$ in the deeper layer (Fig. 7).

The net exchanged transports of DIC at the deep convection area boundaries weakened in both the upper and deeper layers compared to the previous period (Fig. 6g). Over this restratification period characterized by baroclinic instabilities (Jones and Marshall, 1997), the cumulative net physical exchange flux was negative leading to a loss of DIC of 0.80 mol C m$^{-2}$ in the upper layer and of 1.62 mol C m$^{-2}$ in the deeper layer (Fig. 7).

Sea surface $pCO_2$ decreased in early spring (Fig. 6d), when DIC was consumed through strong primary production. The $pCO_2$ difference reached a maximum positive value of 75 µatm mid-April, at the peak of the bloom. Afterwards, when the DCM was formed (Fig. 10a), sea surface $pCO_2$ varied with sea surface temperature that increased until early May and remained around 15 °C until the end of the spring period, when the $pCO_2$ difference was around 30 µatm (Fig. 6c and 6d). The air-sea flux showed positive values with a maximum uptake of 30 mmol C m$^{-2}$ day$^{-1}$ mid-April during the bloom. Over

the whole spring period, the deep convection area absorbed 0.45 mol C m$^{-2}$ of atmospheric $CO_2$ (Fig. 7). The mean spring $pCO_2$ difference varied between 30 and 40 µatm over the deep convection area (Fig. 8c). The mean air-sea $CO_2$ flux was the strongest in the Gulf of Lion where both wind speed and $pCO_2$ difference were maximum (Fig. 9c). To sum up, the upper layer gained DIC through air-sea flux and lost DIC through biogeochemical processes and physical fluxes (Fig. 7b). The loss of DIC by physical fluxes resulted from a loss by lateral transport and a gain through upward transport.


**Summer** (6 June - 31 August, 87 days) - Two long episodes of heat in June and July (Fig. 6a) generated a strong stratification of the water column with a mixed layer shallower than 20 m (Fig. 6b and 10), and increases in surface temperature (Fig. 6c). Early August surface temperature started to slowly decrease. Sea surface $pCO_2$ shows a similar evolution as the one of temperature (correlation coefficient of 0.99 (p-value < 0.01), Fig. 6c and 6d). Very quickly at the

beginning of the summer period the deep convection area became in excess in $CO_2$ relative to the atmosphere. The $pCO_2$ difference reached a maximum absolute value of 135 µatm at the end of July. The model outputs show peaks of $CO_2$



outgassing varying between 18 and 28 mmol C m$^{-2}$ day$^{-1}$ during the northerly wind events that occurred from the end of July. The deep convection area released 0.42 mol C m$^{-2}$ over the whole summer period (Fig. 7a-b). Similarly to the fall situation, the outgassing fluxes were maximum in the western part of the delimited deep convection area (Fig. 9d), influenced by both

strong wind speeds and arrivals of warm waters from the Balearic Sea through anticyclonic circulations, which were characterized by higher $pCO_2$ differences (between 70 and 90 µatm, Fig. 8d).

From August onwards, the DIC drawdown due to biological processes decreased and net DIC production events took place (Fig. 6h). We estimate the contribution of biological processes in summer to be -0.57 mol C m$^{-2}$ in the surface layer and 0.39 mol C m$^{-2}$ in the deeper layer (Fig. 7). The net physical fluxes were again negative in both layers (Fig. 7). On average, the

lateral export of DIC from the upper layer prevailed over the vertical supply of DIC into this layer.

Thus, in summer the upper layer of the deep convection area decreased its DIC inventory in response to physical and biogeochemical processes, as well as outgassing towards the atmosphere (Fig. 7a and 7b).

## 4.2 Annual carbon budget

Figure 12 shows a schematic of the annual budget of dissolved inorganic carbon in the deep convection zone. Our model

results show that the deep convection area acted as a moderate $CO_2$ sink for the atmosphere on an annual scale, over the period September 2012-September 2013. We estimate that it absorbed 0.5 mol C m$^{-2}$ of atmospheric $CO_2$. This uptake of atmospheric $CO_2$ displayed spatial variability (Fig. 13). It was greater than 1 mol C m$^{-2}$ in the northern edge of the area along the Northern Current flowing over the Gulf of Lion continental slope, and became less than 0.25 mol C m$^{-2}$ in the western and eastern edge areas. One can notice that the annual rate remained lower than on the Gulf of Lion's shelf, which is beyond

the scope of this study. Within the sea, biogeochemical processes induced an annual consumption of 3.7 mol C m$^{-2}$ of DIC in the upper layer and a production of 2.3 mol C m$^{-2}$ in the deeper layers. The deep convection area thus appears as a net autotrophic region from a biological point of view, with a DIC consumption of 1.5 mol C m$^{-2}$ considering the whole water column.

Our estimate of net physical fluxes (lateral plus vertical) is 3.3 mol C m$^{-2}$ in the upper layer and -11.0 mol C m$^{-2}$ in the

deeper layer. Specifically, the model indicates a vertical DIC supply of 133.2 mol C m$^{-2}$ from the deeper layer to the upper layer, partly offset by a lateral outflow of 129.8 mol C m$^{-2}$ in the upper layer and an inflow of 122.2 mol C m$^{-2}$ in the deeper layer. The budget in the deep layer masks different signs of physical fluxes: if the deeper layer is subdivided into an intermediate layer (150 m-800 m) and the deeper most layer (800 m-bottom), we find that the former, the intermediate layer, gained an amount of 83.1 mol C m$^{-2}$ of DIC through vertical transport, while it lost 87.6 mol C m$^{-2}$ of DIC through lateral

export. Finally, our model shows that the convection zone was a source of DIC of 8.7 mol C m$^{-2}$ for the rest of the western Mediterranean Sea. While the DIC inventory in the upper layer remained stable (decrease of 0.07 mol C m$^{-2}$), the DIC inventory in the deeper layer experienced a decrease of 8.7 mol C m$^{-2}$. This loss occurred mainly during deep convection, and to a lesser extent during the preconditioning period (in autumn and early winter).



Finally, we complete the inorganic carbon budget with the labile organic carbon fluxes (refractory organic carbon is not
considered in our model). We estimate that during the studied period a lateral export of organic carbon of 1.1 mol C m$^{-2}$ and

0.3 mol C m$^{-2}$ took place in the upper and deeper layers, respectively. The modeled downward export of organic carbon

amounted to 2.3 mol C m$^{-2}$.

## 5 Discussion

Based on high-resolution 3D modeling, we have estimated a DIC budget in the northwestern Mediterranean deep convection
zone over an annual period, September 2012-September 2013. Our results show that both biological and physical processes

dominate the $CO_2$ budget in the upper layer (0-150 m) of the convection zone for the study period. Through their impacts on

DIC concentration, biological and physical flows have both a major role in the intensity and sign of the air-sea exchanges in

the deep convection area.

### 5.1 The pCO2

The modeled spatial mean $pCO_2$ at the sea surface of the deep convection area displays a strong seasonal cycle. The area was

in deficit compared to the atmosphere from November 2012 to early June 2013 and in excess relative to atmosphere the rest

of the year. The spatial mean sea surface $pCO_2$ ranged between 322 and 515 µatm, resulting in a variation of the $pCO_2$

difference at the air-sea interface between -75 µatm during the spring phytoplankton bloom, to 135 µatm in summertime,

when sea surface temperature was maximum. Figure 6 shows that sea surface $pCO_2$ was mostly controlled by thermal
variations outside two periods characterized by changes in surface DIC concentrations: the deep convection period, from

mid-January to late March, and the phytoplankton bloom occurring in late winter/early spring. During deep convection,

vertical mixing of surface water with intermediate and deep water, enriched in DIC, led to an increase in the near-surface

DIC concentration. This induced an increase in sea surface $pCO_2$ and a reduction of the $pCO_2$ difference. Locally, intense

vertical mixing also led to short events of excess (with a maximum $pCO_2$ difference of 7 µatm), while on average the effect
of cooling counterbalanced the effect of upward transport of DIC-enriched waters and remained dominant. Then the late

winter/early spring bloom induced a strong DIC drawdown near the surface that led to annual minimum values of seawater

$pCO_2$.

The seasonal pattern of the simulated sea surface $pCO_2$ averaged over the deep convection area is similar to the one

simulated at the EMSO-DYFAMED and BOUSSOLE sites (Fig. 3) and is in good agreement with those described in
previous observational and modeling studies at these sites (Hood and Merlivat, 2001; Copin-Montégut and Bégovic, 2002;

Bégovic and Copin-Montégut, 2002; Mémery et al., 2002; Copin-Montégut et al., 2004; Merlivat et al., 2018; Coppola et al.,

2020). The high frequency measurements at the CARIOCA buoy described by Hood and Merlivat (2001) and Merlivat et al.

(2018) indicated that the date of the change of sign of the $pCO_2$ difference shows interannual variability and is within a

period lasting for more than a month depending on air-sea heat flux variations and the timing of the bloom onset. They





showed that in autumn the change of sign extends from mid-September to the end of October and in spring from early May to mid June. The magnitude of the variation of the modeled sea surface $pCO_2$ (spatial mean: 193 µatm, at EMSO-DYFAMED site: 192 µatm) is in the range of those deduced from observations (120 µatm by Bégovic and Copin-Montégut (2002), 230 µatm by Hood and Merlivat (2001), ~ 200 µatm by Merlivat et al. (2018)). More specifically, the impact of deep convection on sea surface $pCO_2$ through a large upward transport of $CO_2$-enriched waters, leading to its increase, is

consistent with previous studies. The measurements at the BOUSSOLE site gave evidence to brief windy periods marked by sea surface $pCO_2$ higher than atmospheric $pCO_2$ (Hood and Merlivat, 2001; Copin-Montégut and Bégovic, 2002; Copin-Montégut et al., 2004; Merlivat et al., 2018). The observations of the CASCADE cruise in March 2011 in the Gulf of Lion also showed high surface concentration of DIC and sea surface $pCO_2$ higher than atmospheric $pCO_2$ in deep convection cells (Touratier et al., 2016).

**5.2 The air-sea $CO_2$ flux**

Following the variability of $pCO_2$, the model results show that the deep convection zone absorbed atmospheric $CO_2$ over a 7-month period, from November 2012 to early June 2013, and released $CO_2$ to the atmosphere during the rest of the annual period studied (5 months: from September to October 2012, and June to August 2013). During deep convection, the vertical physical supplies of DIC into the surface layer reduced the uptake of atmospheric $CO_2$ and also generated locally short

events of outgassing (not shown). Considering the whole deep convection area and period, those events did not compensate for the effect of cooling, as found in sub-tropical regimes by Takahashi et al. (2002). The late winter/early spring bloom induced maximum ingassing in mid-April.

On an annual scale, our results indicate that the deep convection area was a sink for atmospheric $CO_2$. Our estimate of the annual air-to-sea flux is 0.47 mol C $m^{-2}$, which, considering the area of the zone, corresponds to an uptake of atmospheric

$CO_2$ of 0.4 Tg $yr^{-1}$.

Our estimate of the air-sea flux is associated with various sources of uncertainties related to the modeling of the different physical, biogeochemical and air-sea exchange processes. Regarding the wind speed accuracy, Ulses et al. (2021) calculated a percentage bias of -0.5% and a normalized RMSE of 13.9%, based on comparisons between ECMWF forcing fields and high-frequency measurements during the DEWEX cruises. The statistical analysis in Sect. 3 indicates that the model has low

to moderate RMSE for surface temperature (0.50 °C), salinity (0.10), $pCO_2$ (< 26 µatm) and DIC (< 24 µmol $kg^{-1}$) and low biases (respectively: -0.31 °C, -0.04, 6 µatm, < 15 µmol $kg^{-1}$).

To assess the uncertainties linked to the calculation of the gas transfer coefficient, we performed sensitivity tests using eight other parameterizations of this coefficient (Sect. 2.1.4). The estimates of the annual air-sea flux using these parameterizations are displayed in Fig. 14 and Table S1. The results of these tests indicate that the deep convection zone is

found as a moderate $CO_2$ sink for the atmosphere using all parametrizations. The estimate in the reference run, based on the wind speed quadratic-dependency relation established by Wanninkhof (1992), is close to the mean value of all the estimates, 0.43 (± 0.12) mol C $m^{-2}$ $yr^{-1}$. The highest estimates were obtained using the relation from the cubic-dependency



parametrization of Wanninkhof and McGillis (1999) and the bubble-inclusive parametrizations of Woolf (1997) and Stanley et al. (2009). The lowest estimate, which is almost twice as small as the mean value, was obtained using the parametrization

of Liss and Merlivat (1986). The second set of tests of sensitivity on atmospheric $CO_2$ forcing, shows that the annual air-sea flux varies between 0.33 and 0.61 mol C m$^{-2}$ yr$^{-1}$ (SD of 0.14 mol C m$^{-2}$ yr$^{-1}$) if an uncertainty value of 3 ppm to the atmospheric mole fraction is constantly subtracted and added, respectively. Finally, sensitivity tests taking into account a supplementary consumption term for $CaCO_3$ precipitation (Sect. 2.1.4) were performed to assess its potential influence on air-sea $CO_2$ flux. They show that calcification processes could lead to an underestimation of the annual air-sea $CO_2$ uptake

by 23 to 58% with estimates of 0.72 mol C m$^{-2}$ yr$^{-1}$, based on PIC:OC ratio, and 0.58 mol C m$^{-2}$ yr$^{-1}$, based on the parametrization used in Lajaunie-Salla et al. (2021). This demonstrates the need to better constrain this term in future studies on carbonate system dynamics.

Our estimates of the annual air-sea flux over the whole deep convection area and at the DYFAMED site, 0.47 and 0.33 mol

C m$^{-2}$ yr$^{-1}$, respectively, are close to those provided in previous observational and modeling studies at the DYFAMED site. Based on the parametrization of Liss and Merlivat (1986) and for the period 1995-1997, Hood and Merlivat (2001) found a value of 0.10-0.15 mol C m$^{-2}$ yr$^{-1}$ using hourly measurements, while Mémery et al. (2002) found a value of 0.15 ± 0.07 mol C m$^{-2}$ yr$^{-1}$ using a 1-D model. We obtained an annual flux of 0.17 mol C m$^{-2}$ yr$^{-1}$ at the DYFAMED site using the same gas transfer relationship. Bégovic (2001) and Copin-Montégut et al. (2004) estimates varied between 0.42 and 0.68 mol C m$^{-2}$ yr$^{-1}$

for the period 1998-2000, using monthly $CO_2$ measurements and the parametrization proposed by Wanninkhof and McGillis (1999). Using the same parametrization, our estimate amounts to 0.40 mol C m$^{-2}$ yr$^{-1}$ at DYFAMED. Finally, Merlivat et al. (2018) estimated a close annual $CO_2$ air-sea flux of 0.45 mol C m$^{-2}$ yr$^{-1}$, using hourly measurements for the period 2013-2015.

Based on a 1D satellite data approach (Antoine and Morel, 1995) applied with a horizontal resolution of 0.5° to the whole

Mediterranean Sea over the period 1998-2004, D'Ortenzio et al. (2008) estimated an annual mean $CO_2$ air-sea flux ranging between 0 and 4 mol C m$^{-2}$ over the NW deep convection area. The larger homogeneity in our estimates could be partly ascribed to the horizontal diffusion and advection that were accounted for in our model. Using a 3D coupled physical-biogeochemical reanalysis of the Mediterranean Sea, von Shuckmann et al. (2018), over the period 1999-2016, and Cossarini et al. (2021), over the period 1999-2019, estimated a mean annual air-sea flux in the deep convection zone ranging between

0 and 0.5 mol C m$^{-2}$ yr$^{-1}$, and 0 and 1 mol C m$^{-2}$ yr$^{-1}$, respectively. Our results in terms of spatial distribution, with minimum values in the western edge of the deep convection zone and maximum values in the northern area of the Gulf of Lion, are also consistent with their results.






### 5.3 Physical flows in the deep convection area

Our study confirms the importance of deep convection on DIC vertical distribution and surface $pCO_2$ in the study area, as shown in previous observational studies (Copin-Montégut et al., 2004; Touratier et al., 2016), and highlights that physical flows play a crucial role in the DIC budget in this highly energetic region.

Our 3D model results allowed us to distinguish the contribution of vertical and lateral transports in the net physical exchange flux. They both show a similar seasonal cycle with greater magnitude in fall, the preconditioning phase, and in winter, the convection period, being both sea heat loss periods. During those periods, vertical supply overwhelmed lateral export in the upper layer. Conversely, during the stratification phase and stratified period, in spring and summer, lateral export prevailed over vertical supply. At the annual scale, we estimate that the vertical supplies amounted to 133 mol C m$^{-2}$ yr$^{-1}$. They were

almost counterbalanced by a lateral transfer of 130 mol C m$^{-2}$ yr$^{-1}$ to adjacent upper layer areas, which acted as a major sink of DIC for the deep convection upper layer.

By estimating a water mass budget, we found that lateral and vertical DIC flows are highly significantly correlated with lateral (R=0.9998, p-value < 0.01) and vertical (R = 0.9998, p-value < 0.01) water flows, respectively. Moreover, a detailed calculation of the water and DIC exchanges flows allowed us to evaluate the contribution of (1) the difference in inflowing

and outflowing water fluxes, at constant mean DIC concentration, and (2) the difference in DIC concentrations, at constant flux. We found that the first contribution is largely dominant compared to the second one, highlighting that the lateral flows of DIC are essentially related to the difference of inflows and outflows of water, rather than to DIC concentration differences between deep convection waters and surrounding waters. Strong mesoscale activities and instabilities within and on the edge of the mixed patch that characterized the convection zone shown in previous works (Marshall and Shott, 1999; Testor et al.,

2018) could lead to this strong lateral transfer of water and associated DIC, as illustrated on Fig. 11. The findings of Waldman et al. (2018) showing water sinking in the general circulation suggest that DIC could be partly transferred back in deep waters in the boundary current. Studying the dissolved oxygen dynamics, Wolf et al. (2018) also found that lateral processes could play a major role in the biogeochemical annual budget in the deep convection located in the central Labrador Sea. We consider that the study of the contribution of the various lateral exchange mechanisms in the lateral DIC transfer,

such as Ekman-driven transport, geostrophic advection, frontal processes, submesoscale coherent vortices, is out of scope of this first work on the DIC budget, but further complementary works will be dedicated to this subject.

### 5.4 Net community production and air-sea fluxes

Previous modeling studies (Herrmann et al., 2013; Ulses et al., 2016) showed that the northwestern Mediterranean deep convection area acts as an autotrophic ecosystem with, on an annual timescale, gross primary production dominating

respiration and hence a positive NCP. The present modeling study displays a NCP in the upper layer of 3.7 mol C m$^{-2}$ yr$^{-1}$ and a DIC buildup of 2.3 mol C yr$^{-1}$ through respiration of heterotrophic organisms in the deeper layer. Our budget shows that, in the upper layer, the net biological loss term of DIC is counterbalanced for 88% by physical gain fluxes, and only for



12% by air-sea gain fluxes. It clearly appears that deep vertical mixing and advection significantly braked the atmospheric $CO_2$ uptake in winter, by bringing into the upper layer remineralized organic carbon. We quantify here that the annual air-sea

$CO_2$ flux is ~ 8 times smaller than the annual NCP in the upper layer. These results are in line with previous findings on the reducing effect of winter ventilation on atmospheric $CO_2$ uptake of Oschlies and Kähler (2004), Körtzinger et al. (2008a; 2008b) and Palevsky and Nicholson (2018) in the northern Atlantic Ocean, and Palevsky and Quay (2017) in the Pacific Ocean.

In our simulation, the downward export of organic carbon (OC) at the base of the photic zone of the deep convection area is

estimated at 2.3 mol C m$^{-2}$ yr$^{-1}$. The results of Herrmann et al. (2013) and Ulses et al. (2016) showed, using a similar coupled modeling approach, that OC export is characterized by high interannual variability, with standard deviation between 24 to 37 %, linked to the variability of the convention strength. The intensity of the winter vertical mixing has been shown to be subject itself to high interannual variability (Houpert et al., 2016; Margirier et al., 2020). Observations in the core of the convection zone by Margirier et al. (2020) evidenced that only intermediate convection events took place in the four years

following the 2013 events. Thus, organic carbon transferred into the deeper layer could either have been stored in the deep convection area until the next 2018 events (Fourrier et al., 2022) when it could have been reinjected in its remineralized form, or it could have been transferred, partly under remineralized form, towards the southwestern Mediterranean through the dispersion of the newly-formed dense water (Schroeder et al., 2008; Beuvier et al., 2012). Here we estimate that an amount of 0.3 mol C m$^{-2}$ yr$^{-1}$ of organic carbon was laterally exported from the deeper layer. We found that the vertical

supply of DIC into the upper layer is two orders of magnitude higher than OC export, or the upper layer and depth-integrated NCP. This is explained by the equilibrium role of the DIC lateral transfers towards the surrounding zone. This shows that a 1D approach would not be appropriate to take into account the complexity of the 3D mechanisms of exchanges for the DIC budget of this deep convection area which has to be considered integrated into a whole regional system, especially in a context of a changing atmosphere and ocean.

**5.5 Contribution of the northwestern deep convection region to the carbon budget of the Mediterranean Sea**

Our results indicate that the NW Mediterranean deep convection zone was a sink of carbon for the atmosphere and a source of carbon for the Western Mediterranean Sea over the period September 2012 to September 2013. More specifically, we found that the lateral exchanges with the surrounding region were characterized by an inflow of carbon into the deep layers, although organic carbon was exported during deep convection and restratification periods, and an outflow of both organic

and inorganic carbon in upper water masses.

Previous studies investigating the air-sea $CO_2$ flux at the scale of the whole Mediterranean Sea showed that this sea acted as a moderate sink of atmospheric $CO_2$ over the past decades (Copin-Montégut, 1993; D'Ortenzio et al., 2008; Cossarini et al., 2021). According to those studies, the northern continental shelves and open seas (Gulf of Lion, Adriatic and Aegean seas) absorbed atmospheric $CO_2$, while the southeastern Mediterranean was in excess in $CO_2$ relative to the atmosphere and



released $CO_2$ to the atmosphere. The water formation areas and, in particular, the northwestern Mediterranean deep convection area, were shown to be regions of relatively strong atmospheric $CO_2$ uptake (Cossarini et al., 2021). Estimates of air-sea flux for the whole Mediterranean Sea varied between 0.2 Tg C yr$^{-1}$ (D'Ortenzio et al., 2008) and 2.6 Tg C yr$^{-1}$ (Cossarini et al., 2021) if only the open seas are considered, and between 4.2 Tg C yr$^{-1}$ (Copin-Montégut, 1993; Cossarini et al., 2021) and 12.6 Tg C yr$^{-1}$ (Solidoro et al., 2022) including the continental shelves. Thus the NW Mediterranean deep

convection area, which represents 2.5% of the Mediterranean Sea surface, and which, we estimate, absorbed 0.4 Tg C yr$^{-1}$, could strongly contribute to the uptake of atmospheric $CO_2$ in the open Mediterranean Sea.

  Our results show that DIC was transferred from the deep depths, and to a much lesser extent from the atmosphere, to the surface and intermediate water masses and then transferred laterally to the neighboring sub-basins. The transfer of DIC in surface water masses which is here estimated at 109 Tg C yr$^{-1}$ could mitigate the air-sea $CO_2$ uptake in winter and spring also

in the surrounding western and southern seas. It could represent 21% of the DIC outflow from the western to the eastern Mediterranean sub-basin estimated by Solidoro et al. (2022) at 509 Tg C yr$^{-1}$, and between 8 to 22% of the Atlantic $CO_2$ surface inflow estimated between 660 to 1310 Tg C yr$^{-1}$ by Aït-Ameur and Goyet (2006) and at 487 Tg C yr$^{-1}$ by Solidoro et al. (2022). Finally, the transfer of DIC in intermediate waters, estimated here at 73 Tg C yr$^{-1}$, could represent up to 11% to the Mediterranean DIC export at the Gibraltar Strait towards the Atlantic Ocean, estimated to range between 680 and 1380

Tg C yr$^{-1}$ (Aït-Ameur and Goyet, 2006), and 100% of the net (difference between Atlantic surface inflow and Mediterranean outflow) DIC outflow, estimated between 20 and 70 Tg C yr$^{-1}$ (Huertas et al., 2009).

  Our DIC budget assessment is limited to a single year and will need to be extended to a longer period to investigate in particular the question of carbon sequestration. The deep convection process exhibits strong interannual variability related to

that of air-sea heat flux intensity and hydrological properties of water masses (Houpert et al., 2016; Somot et al., 2016; Estournel et al., 2016; Margirier et al., 2020). The interannual variability in the intensity and extent of deep convection is expected to directly impact the vertical supply of respired carbon into the upper layer. On the other hand, through variability in temperature, and in vertical supply of dissolved inorganic nutrients into the upper layer, it should also alter biogeochemical and air-sea fluxes of $CO_2$.

Along with interannual variability, the Mediterranean Sea is experiencing changes in atmospheric forcing and water mass properties in response to global warming. The findings of Merlivat et al. (2018) based on $pCO_2$ measurements over the 2 periods 1995-1997 and 2013-2015 suggested an increasing trend in surface DIC concentration. Based on observations from EMSO mooring sites and MOOSE cruises, Coppola et al. (2020) confirmed this trend and evidenced increasing trends also in intermediate (300-800 m) and deep (> 2000 m) waters of the Ligurian Sea, over the period 1998-2016. The increasing

trend in surface DIC concentration could not only be explained by the trend in atmospheric $pCO_2$ and is also expected to be influenced by biogeochemical changes in water masses inflowing at the Gibraltar Strait (Merlivat et al., 2018), as well as over the entire Mediterranean basin, especially in the intermediate and deep water formation areas of the eastern basin (Wimart-Rousseau et al., 2021).



Finally, the reduction of winter mixing and the intensification of marine heat waves predicted by models in the second half
of the 21$^{st}$ century (Darmaraki et al., 2019; Soto-Navarro et al., 2020) should clearly modify the contribution of the NW deep
convection zone in the Mediterranean. Based on coupled models over the entire Mediterranean Sea, Solidoro et al. (2022)
and Reale et al. (2022) predicted, in response to the increase in atmospheric $pCO_2$, temperature and stratification, an increase
in atmospheric $CO_2$ uptake, in DIC inventory in the whole Mediterranean Sea and modifications of the exchange fluxes
between the eastern and western sub-basins. 3D coupled models clearly constitute useful tools to gain insight into carbon
budget and multi-model ensemble exercises on these issues, as performed by Friedland et al. (2021) on the influence of
inorganic nutrient river inputs, could allow a refinement of the carbon budget terms and their evolution, together with an
assessment of their uncertainties.

## 6 Conclusion

We have estimated for the first time a $CO_2$ budget for the whole northwestern Mediterranean deep convection zone over an
annual period using a high-resolution 3D coupled hydrodynamic-biogeochemical-chemical modeling. An assessment of the
model results through their comparisons with DEWEX and MOOSE-GE cruise observations, as well as EMSO-DYFAMED
mooring and BOUSSOLE buoy site observations and outputs of the CANYON-MED neural networks, shows the ability of
the model to describe the seasonal cycle and spatial variability of the DIC dynamics in this region with good accuracy. Based
on the present study over the year 2012/13, we can draw the following conclusions for this key region in the ocean
circulation and biogeochemical cycles in the Mediterranean Sea:

-    The $CO_2$ dynamics in the NW Mediterranean deep convection area underwent large seasonal variation. The region
      was marked by a deficit of $CO_2$ compared to the atmosphere from the second part of fall to the first part of spring,
      which led to a 7-month ingassing of atmospheric $CO_2$. The deficit situation, to a large extent controlled by
      temperature variability, was, on the one hand, reduced by vertical supply of DIC during the period of deep
convection, and on the other hand, accentuated and extended by the spring phytoplankton bloom. This underlines
      the findings of Mémery et al. (2002) on the importance of data of sea surface DIC or $pCO_2$ data during deep
      convection for precise estimates of air-sea $CO_2$ flux in this area, in addition to sea surface temperature observations
      and spring NCP estimates.
-    On an annual basis, the NW Mediterranean deep convection area acted as a sink of atmospheric $CO_2$. We estimate
an annual uptake of 0.47 mol C m$^{-2}$ yr$^{-1}$. The maximum fluxes (> 1 mol C m$^{-2}$ yr$^{-1}$) occurred in the northern Gulf of
      Lion region, submitted to strong northerly continental winds and located at the edge of the deep convection between
      the Northern Current and the core of the deep convection, while minimum values (close to null values) are found in
      the western zone where warm anticyclonic gyres developed. The sensitivity tests on the parametrization of gas
      transfer velocity indicate an uncertainty on the annual estimate of 28%. Moreover, we displayed that calcification



processes could lead to an underestimation by 23 to 58% of the annual uptake, highlighting the need for the refinement of the model in future studies.

- The annual DIC budget in the upper layer of the deep convection area was co-dominated by biogeochemical and physical fluxes, estimated here at -3.7 and 3.3 mol C m$^{-2}$ yr$^{-1}$, respectively. The net physical flux resulted from a balance of a net upward transfer and a net lateral export, both exhibiting maximum intensity during the
preconditionning and deep convection period. The air-sea $CO_2$ flux only represents 13% of the upper layer NCP and 31% of NCP integrated over the whole water column. These results confirm that the DIC budget in this region should be addressed with a 3D approach considering the complex physical mechanisms taking place.

- The NW Mediterranean deep convection area acted as a source of DIC for the surface and intermediate water masses flowing towards the southern Western Mediterranean. The transfer of DIC in the adjacent surface and
intermediate water masses could mitigate the atmospheric $CO_2$ uptake also in the surrounding open sea of the sub-basin, and contribute up to 10 and 20% to the DIC exchanges with the Eastern Mediterranean and Atlantic Ocean.

**Author contributions**

CU, CE, PM, KS, and FK developed the coupled model. CU designed the simulations. CU performed model simulations. CU and CE performed the analyses of the model outputs. MF, LC, DL, FT, CG, and VG provided the observational and
CANYON-MED data. MF, LC, and DL helped with data interpretation. PT and XDM contribute to the experimental design and carrying out of DEWEX cruises, and analysis of the hydrological data. CU wrote the initial version of the manuscript. All authors discussed the results and revised the manuscript.

**Competing interests**

The authors declare that they have no conflict of interest.

**Acknowledgments**

We want to acknowledge the scientists and crews of the Flotte Océanographique Française (https://www.flotteoceanographique.fr/) who contributed to the cruises carried out in the framework of the DEWEX project and MOOSE program (CNRS-INSU). We thank Nicolas Metzl from LOCEAN for helpful discussions, and the Service National des Paramètres Océaniques du CO2 (SNAPO-CO2, LOCEAN, Sorbonne University-CNRS, France) for the total
inorganic carbon and total alkalinity analyses. We thank Thibaut Wagener from MIO for his help on data analysis and his useful comments on an earlier version of the manuscript. We are grateful to Dariia Atamanchuk from Department of



Oceanography at Dalhousie University for her help in the calculation of air–sea $CO_2$ fluxes. We also thank Xin Lan and Edward Dlugokencky from NOAA Global Monitoring Laboratory for the flask-air $CO_2$ data. We thank Marta Álvarez from IEO and collaborators for making available the CARIMED database to us. This study is a contribution to the MerMex (Marine Ecosystem Response in the Mediterranean Experiment) project of the MISTRALS international program and has been supported by the European Union's Horizon 2020 EuroSea project (grant agreement No 862626). The numerical simulations were performed using the SYMPHONIE model, developed by the SIROCCO group (https://sirocco.obs-mip.fr/), and computed on the cluster of Laboratoire d'Aérologie and HPC resources from CALMIP grants (P1325, P09115 and P1331).

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

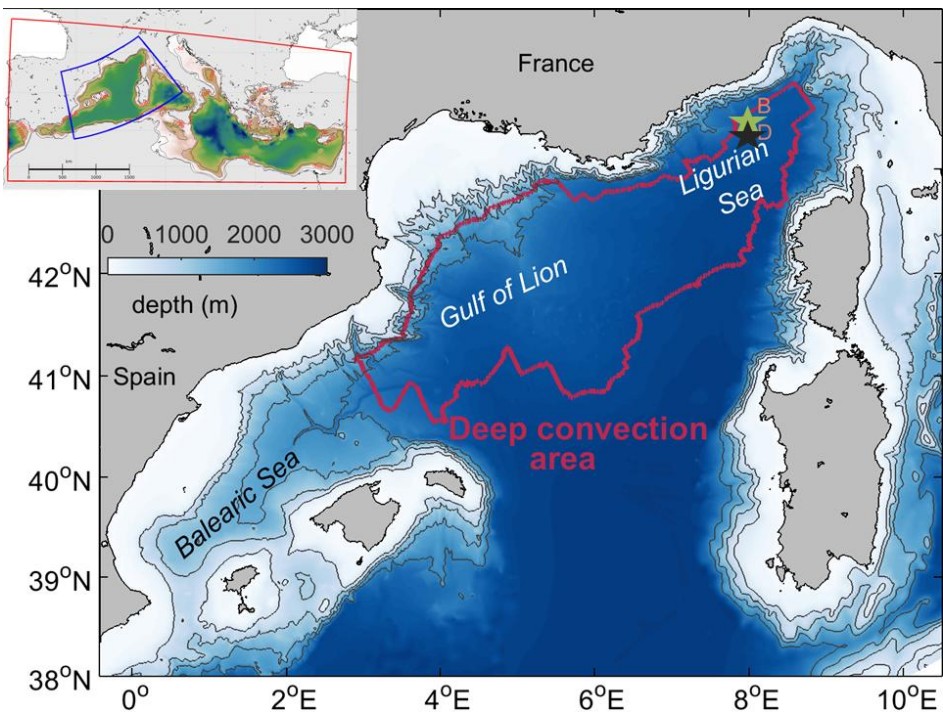

**Figure 1: Bathymetry (m) of the study area. The red contour indicates the limit of the deep convection area defined for the budget calculation (see Sect. 2.1.3). The location of the BOUSSOLE buoy and EMSO-DYFAMED mooring sites in the Ligurian Sea are**
**indicated with a green and black star, respectively. The insert representing the Mediterranean Sea indicates the limits of the coupled model used for this study in blue and for the forcing simulation in orange (see Sect. 2.1.2).**


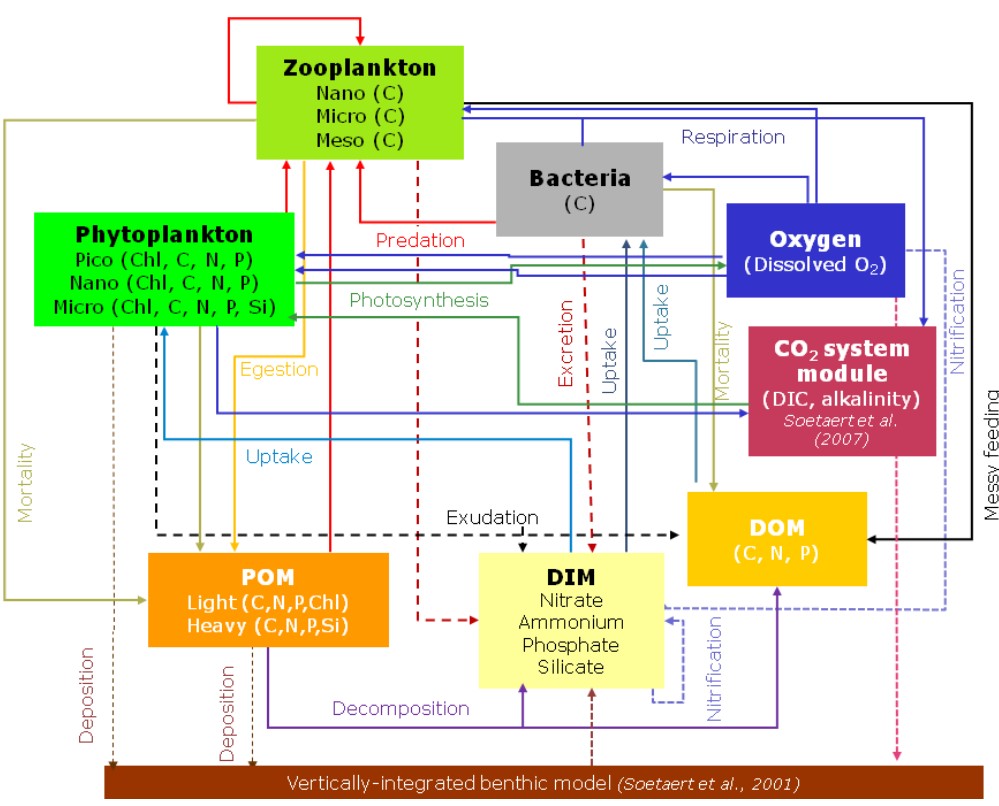

**Figure 2: Scheme of the upgraded biogeochemical model Eco3M-S (redrawn from Ulses et al., 2021).**





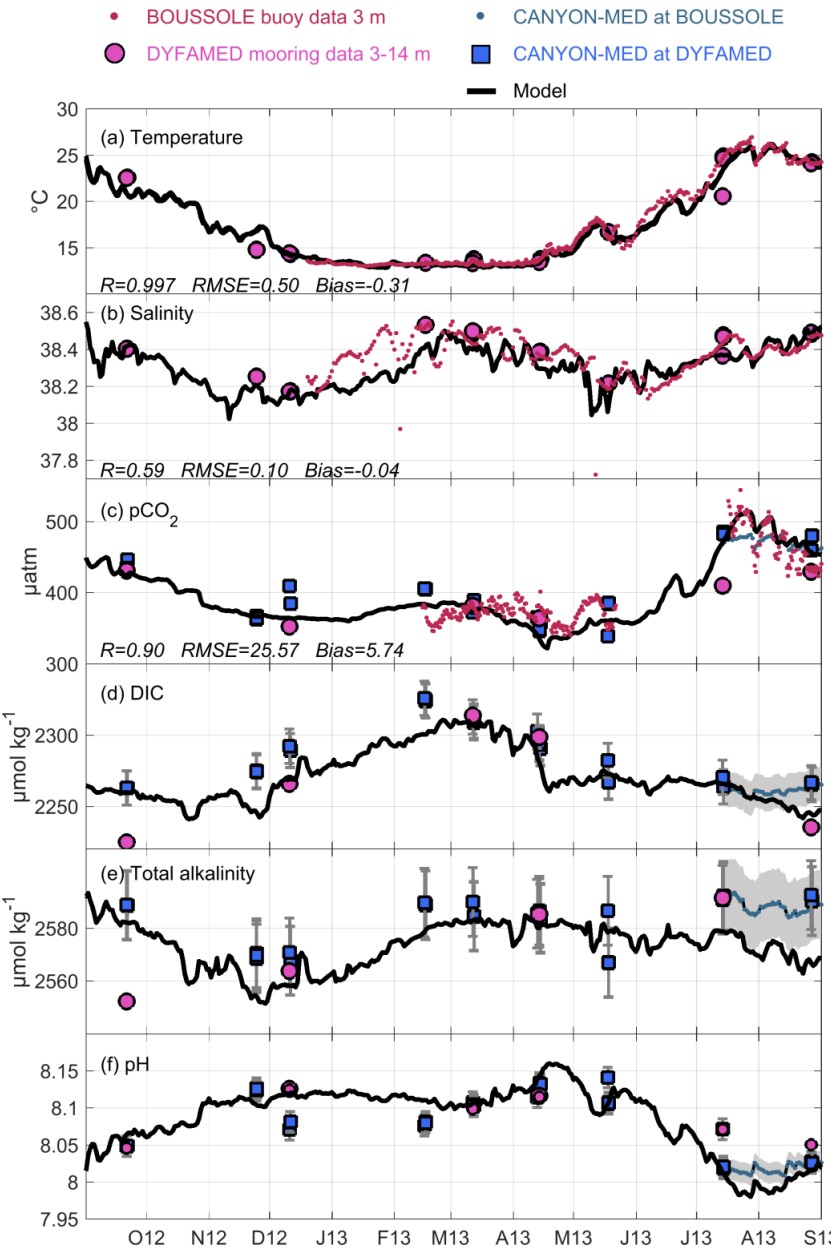

**Figure 3: Time series of (a) temperature, (b) salinity, (c) pCO₂, (d) DIC, (e) total alkalinity, and (f) pH at total scale, modeled at 3 m depth (line in black), observed (small red dots at BOUSSOLE site and pink points at EMSO-DYFAMED site between 3 and 14 m depth) and computed with CANYON-MED neural networks (small blue dots at BOUSSOLE at 3 m, blue squares at EMSO-DYFAMED site between 3 and 14 m depth, error bars are indicated in gray). Correlation coefficient, RMSE and bias between model outputs and BOUSSOLE observations are indicated in (a), (b) and (c).**








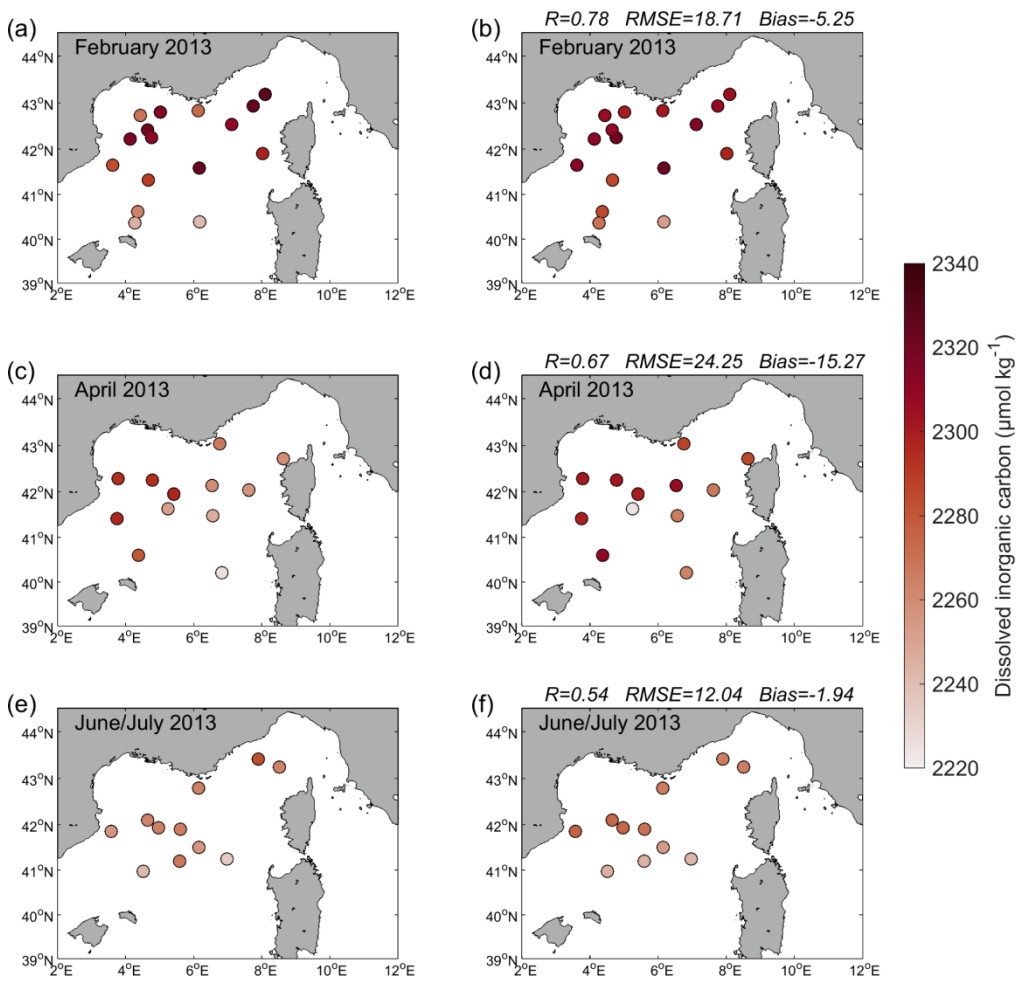

**Figure 4: Surface dissolved inorganic carbon (DIC) concentration (μmol kg⁻¹) observed (left) and modeled (right) over the (a,b) DEWEX Leg1 (1-21 February 2013), (c,d) DEWEX Leg2 (5-24 April 2013), and (e,f) MOOSE-GE (11 June-9 July 2013) cruise periods. The correlation coefficient (R), root mean square error (RMSE), and bias between surface observed and modeled DIC are indicated in (b,d,f).**






**Figure 5: Comparison between observed and modeled dissolved inorganic carbon (DIC) in the northwestern Mediterranean Sea over the (a) DEWEX-Leg1 (10-12 February 2013), (b) DEWEX-Leg2 (8-10 April 2013) and (c) MOOSE-GE (27 June-5 July 2013) cruise periods. Top: Observed (blue and red, mean in dashed lines and shaded areas for standard deviation) and modeled (green and orange, mean in solid lines and shaded areas for standard deviation) profiles in the deep convection area and south of it (latitude < 41°N); Bottom: Taylor diagram summarizing the statistical comparisons between the whole observations (noted O) collected during the three cruises and the corresponding model outputs (noted M): radius is standard deviation, angle is correlation coefficient and distance from the origin is root mean square error (RMSE).**







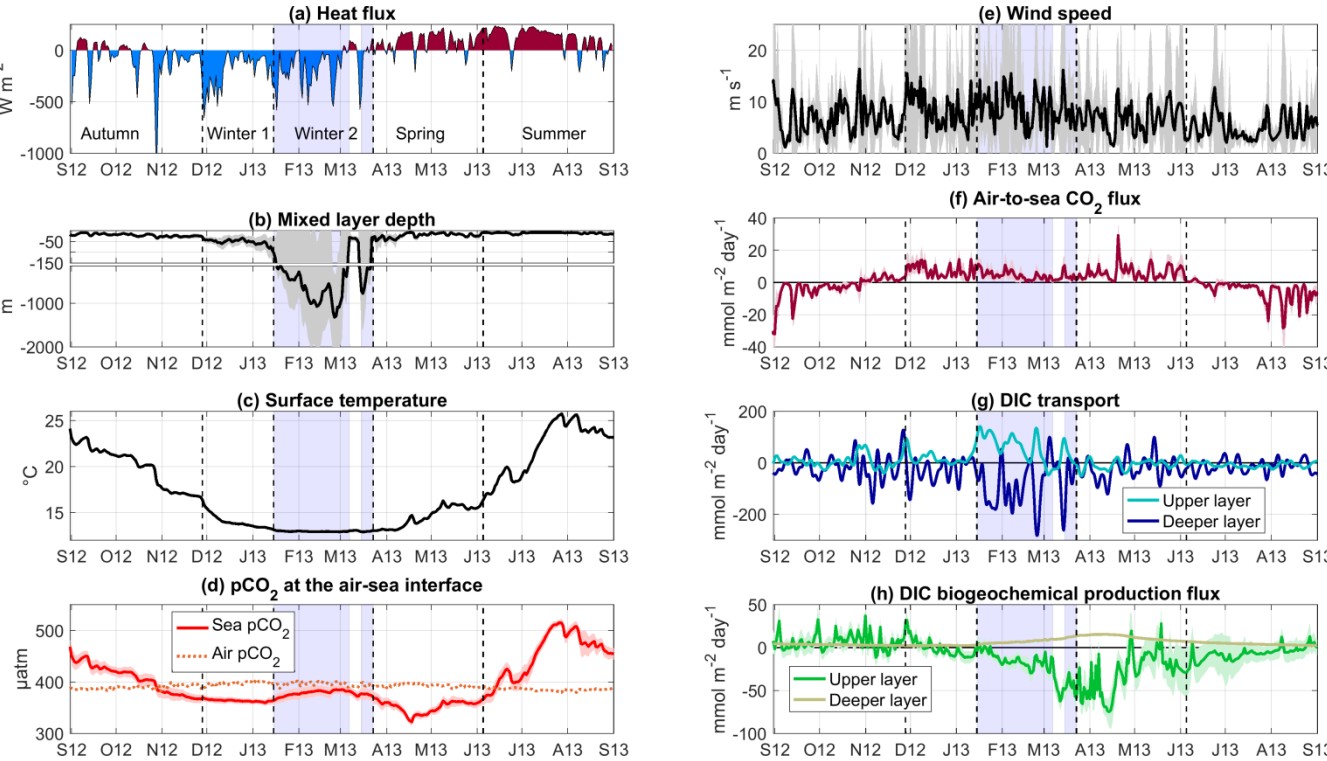

**Figure 6: Time series of modeled (a) total surface heat fluxes (W m$^{-2}$), (b) mixed-layer depth (m), (c) sea surface temperature (°C),**
**(d) sea surface and atmospheric pCO$_2$ (µatm), (e) wind speed (m s$^{-1}$), (f) air-to-sea CO$_2$ flux (mmol C m$^{-2}$ day$^{-1}$), (g) DIC total**
**(vertical plus lateral) transport in the upper (light blue) and deeper layer (dark blue) towards the deep convection area (mmol C**
**m$^{-2}$ day$^{-1}$), and (h) DIC biogeochemical production (see Eq. 1, mmol C m$^{-2}$ day$^{-1}$) in the upper (green) and deeper (brown) layer.**
**All the parameters are spatially averaged over the defined deep convection area (spatial mean in solid line and shaded area for**
**SD). Sources: ECMWF for air-sea heat flux and wind speed, SYMPHONIE/Eco3M-S for the other parameters and fluxes. The**
**blue shaded area corresponds to the deep convection period (period when spatially averaged mixed layer depth > 100 m). Note**
**that the range of the y axis varies for the different carbon fluxes, and due to higher values, SD for transport is not shown.**



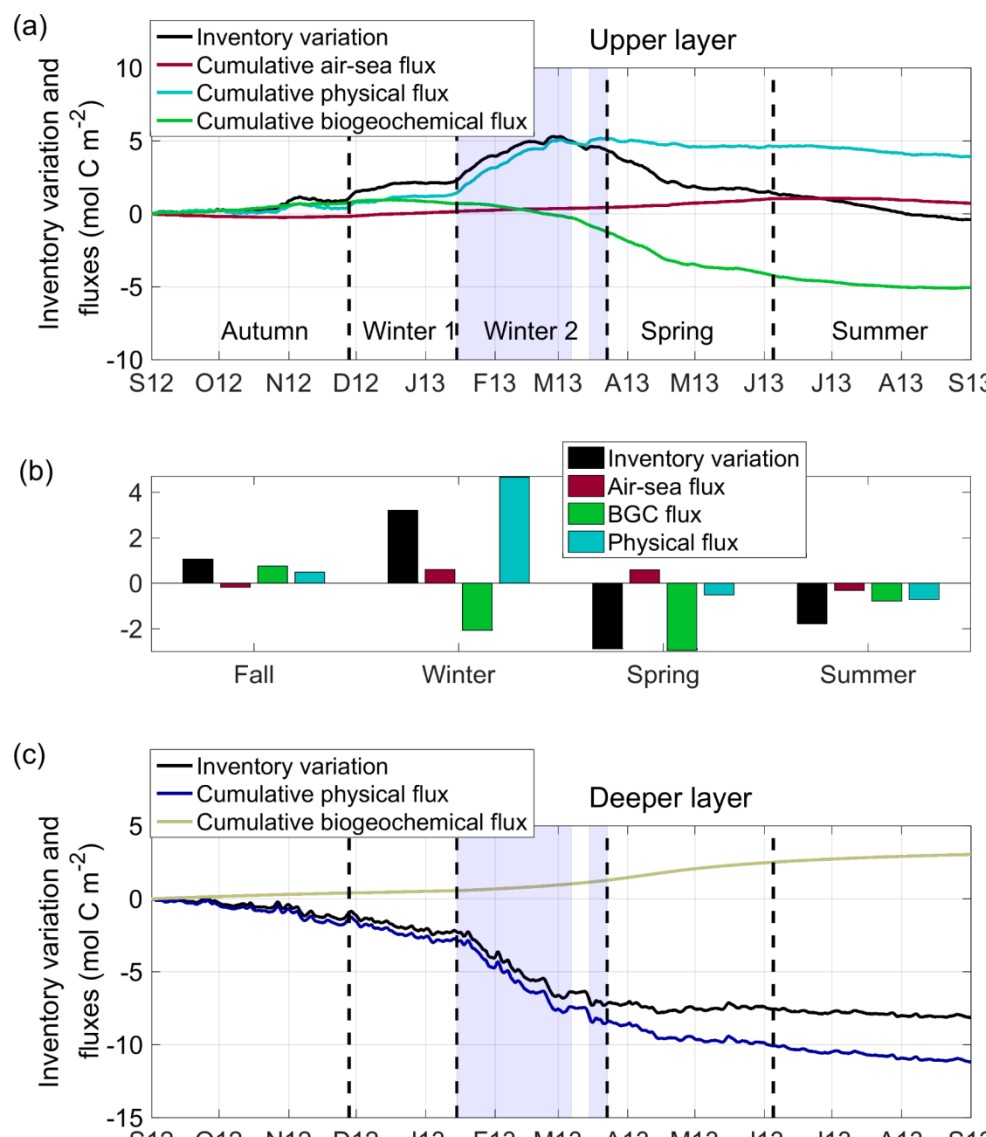


**Figure 7: Dissolved inorganic carbon (DIC) inventory (black) and cumulative air–sea (red), physical transfer (light and dark blue), and biogeochemical (bright and brown green) flux of dissolved inorganic carbon in the (a, b) upper (surface to 150 m) and (c) deeper (150 m to bottom) layers. Time series from September 2012 to September 2013 in (a) and (c) and seasonal cumulative fluxes and internal variation in (b). Unit: mol C m$^{-2}$. Positive values represent inputs for the deep convection area. The blue shaded area**
**corresponds to the deep convection period (period when spatially averaged mixed layer depth > 100 m). The DIC inventory on 1$^{st}$ September 2012 was 353 and 5560 mol C m$^{-2}$ in the upper and deeper layers, respectively.**

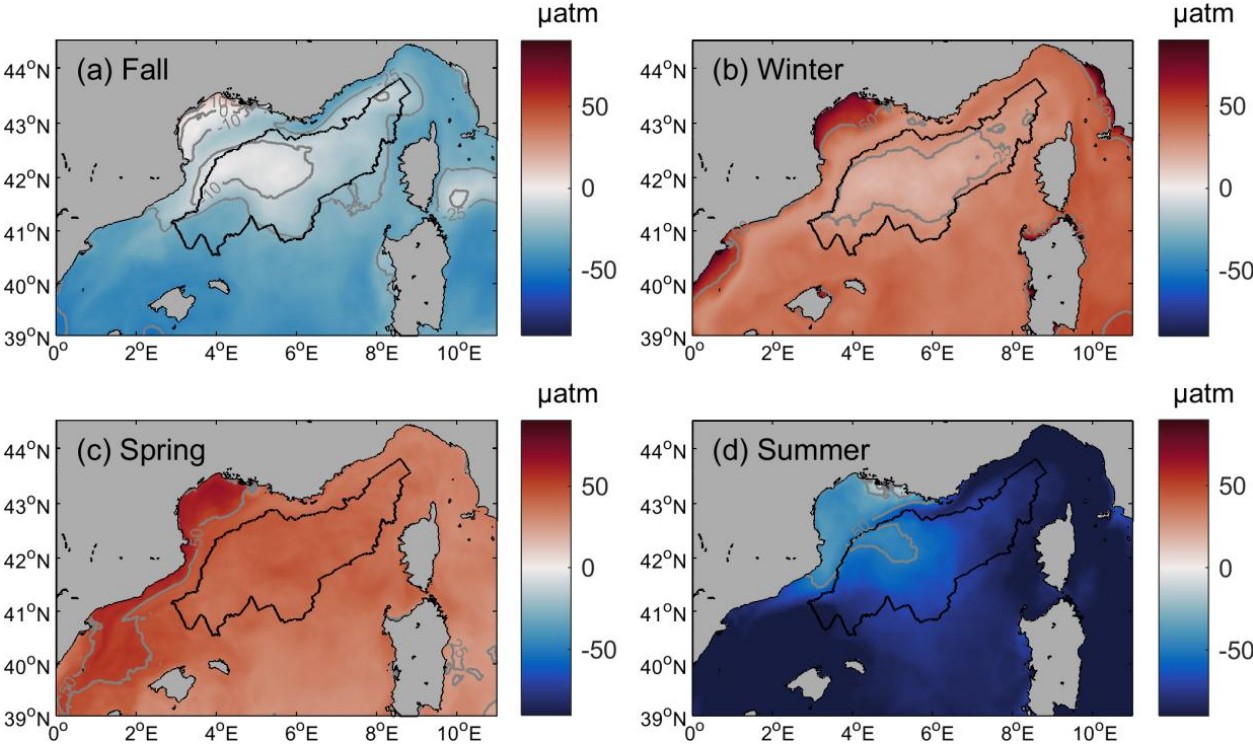

**Figure 8: Seasonal averages of modeled pCO$_2$ difference (pCO$_{2,atm}$ - pCO$_{2,sea}$, in μatm). Note that the periods of seasons here are defined in Sect. 4.1 according to mixed layer depth and biogeochemical processes (Fall: 1 September-27 November, Winter: 28 November-23 March, Spring: 24 March-5 June, Summer: 6 June-31 August). Grey lines indicate pCO$_2$ difference isolines (-50, -25, -10, 0, 10, 25, 50 μatm) and the black line the limit of the deep convection area.**






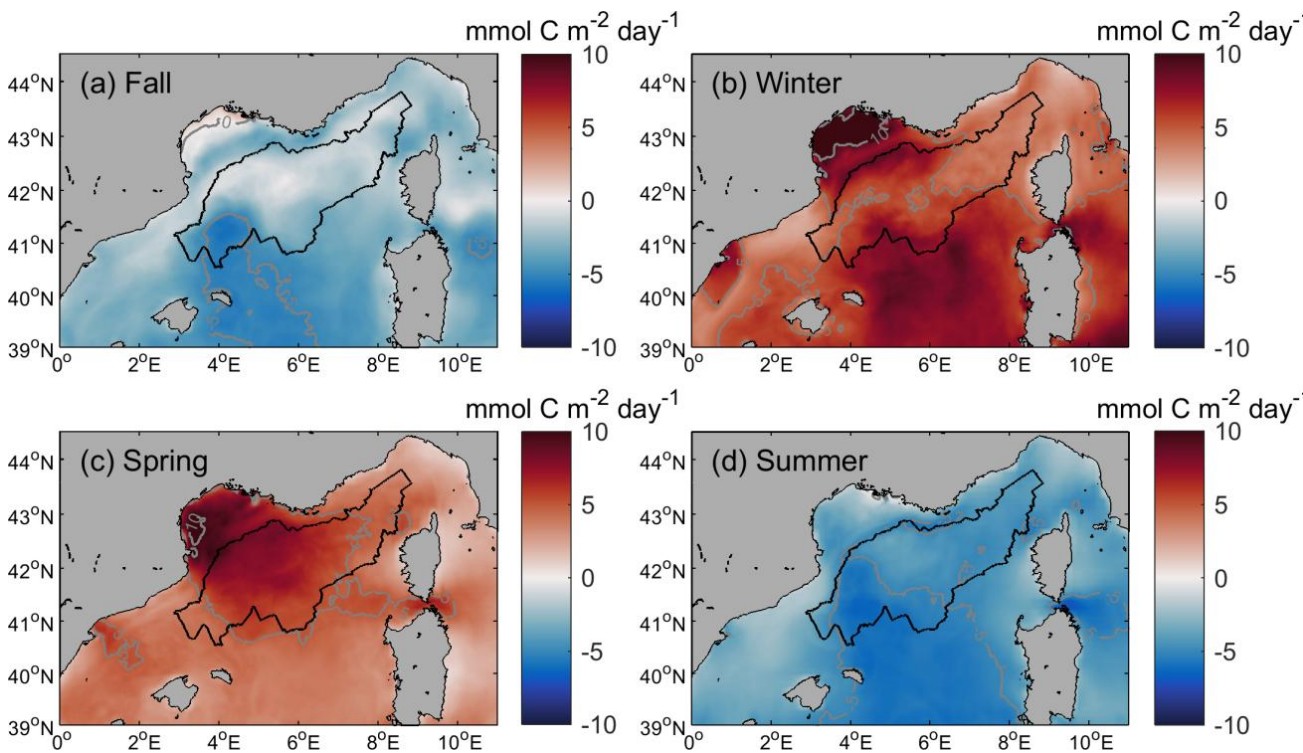

**Figure 9: Seasonal averages of modeled air-to-sea $CO_2$ flux (mmol C m$^{-2}$ day$^{-1}$). Note that the periods of seasons are defined in Sect. 4.1 according to mixed layer depth and biogeochemical processes (Fall: 1 September-27 November, Winter: 2 December-23 March, Spring: 24 March-5 June, Summer: 6 June-31 August). Grey lines indicate $CO_2$ flux isolines (-10, -5, 0, 5, 10 mmol C m$^{-2}$ day$^{-1}$) and the black line the limit of the deep convection area.**

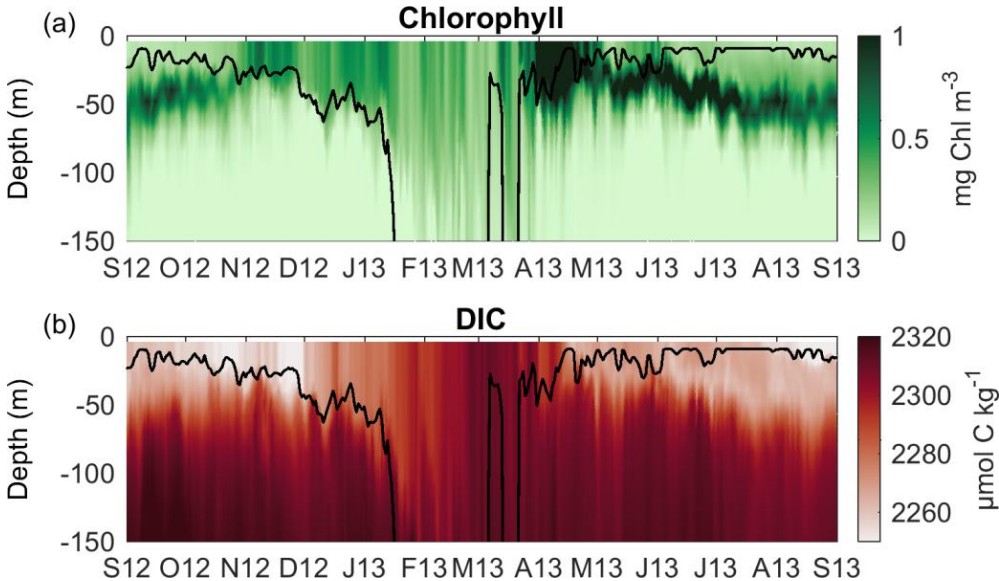

**Figure 10: Time evolution of (a) chlorophyll-a (mg Chl m$^{-3}$) and (b) dissolved inorganic carbon (DIC, μmol C kg$^{-1}$) concentration**
**profile, with mixed-layer depth (m) indicated by the black line, horizontally averaged over the deep convection area.**



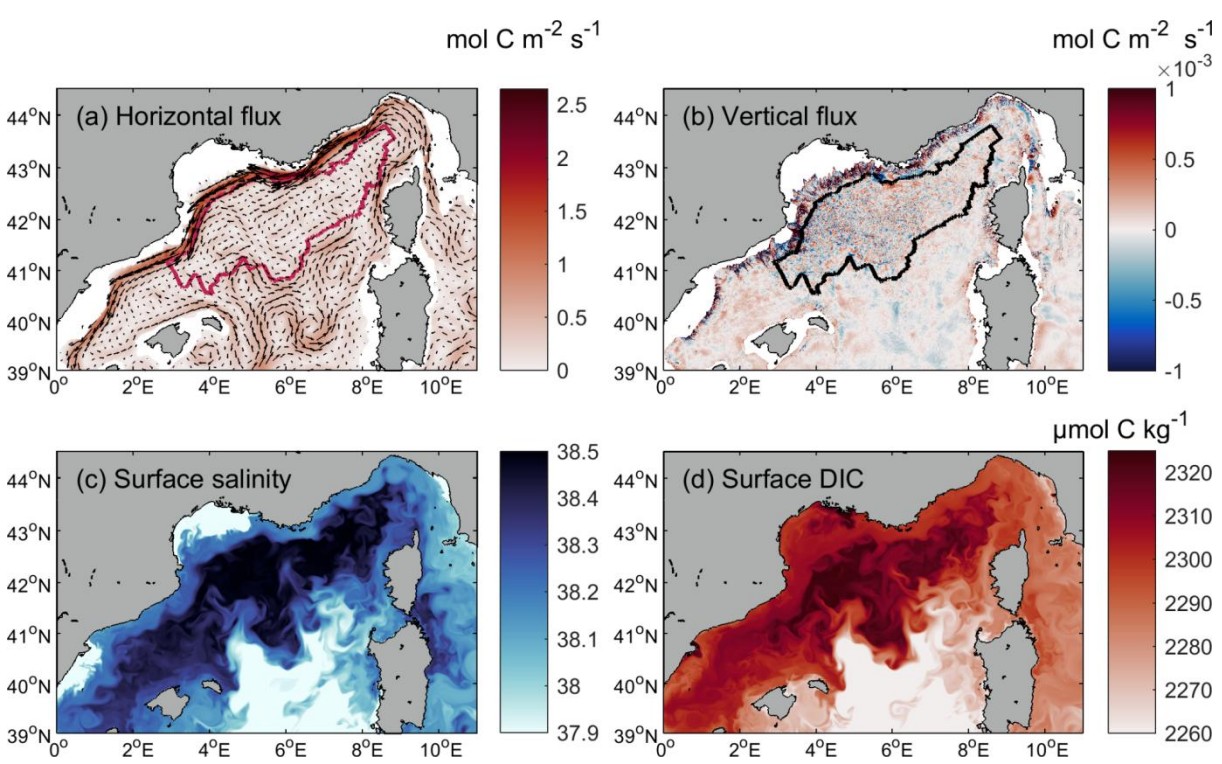


**Figure 11: (a) Winter horizontal flux of dissolved inorganic carbon (DIC, mol C m$^{-2}$ s$^{-1}$), vertically integrated over the upper layer (0-150 m), (b) winter vertical DIC flux (mol C m$^{-2}$ s$^{-1}$) at 150 m, (c) surface salinity and (d) DIC concentration (μmol C kg$^{-1}$) on 4 March 2013. The red and black lines in panel (a) and (b), respectively, indicate the limit of the deep convection area.**




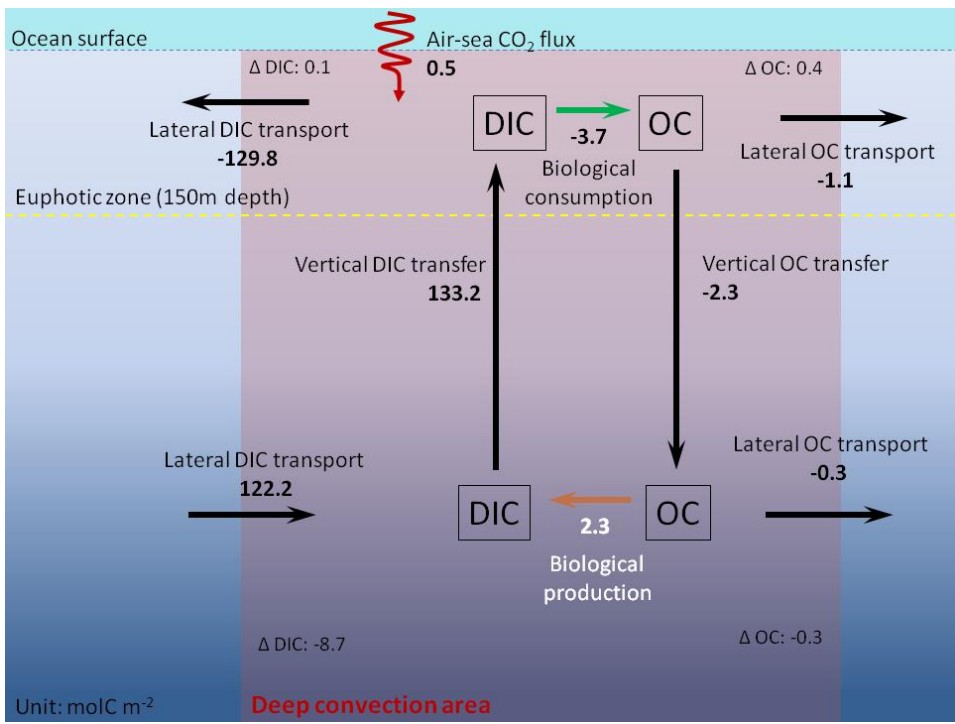

**Figure 12: Scheme of the annual carbon budget for the period September 2012 to September 2013 from the coupled model SYMPHONIE-Eco3M-S. Fluxes are indicated in mol C m$^{-2}$.**


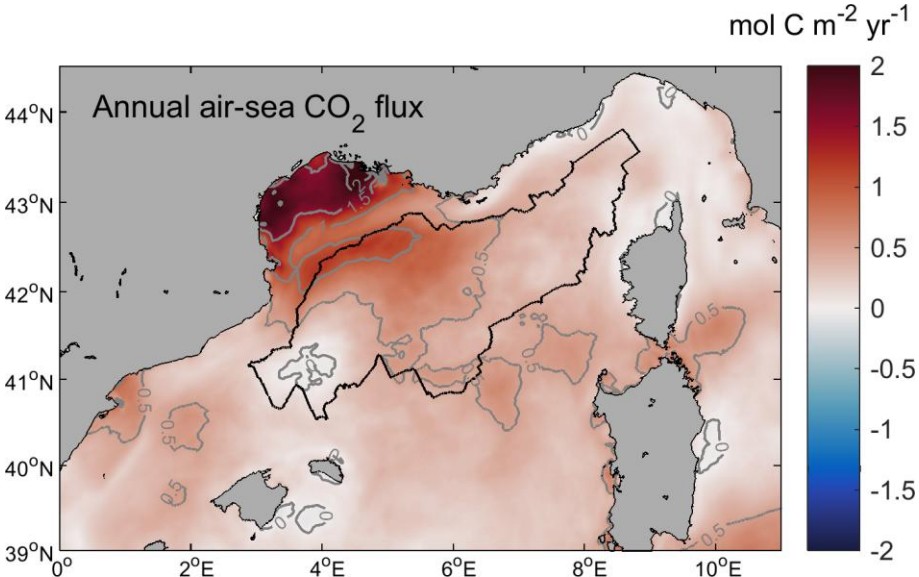

**Figure 13: Modeled annual air-to-sea CO₂ flux (mol C m⁻² yr⁻¹), averaged over the period September 2012-September 2013. Grey lines indicate CO₂ flux isolines (0, 0.5, 1, 1.5 mol C m⁻² yr⁻¹) and the black line the limit of the deep convection area.**





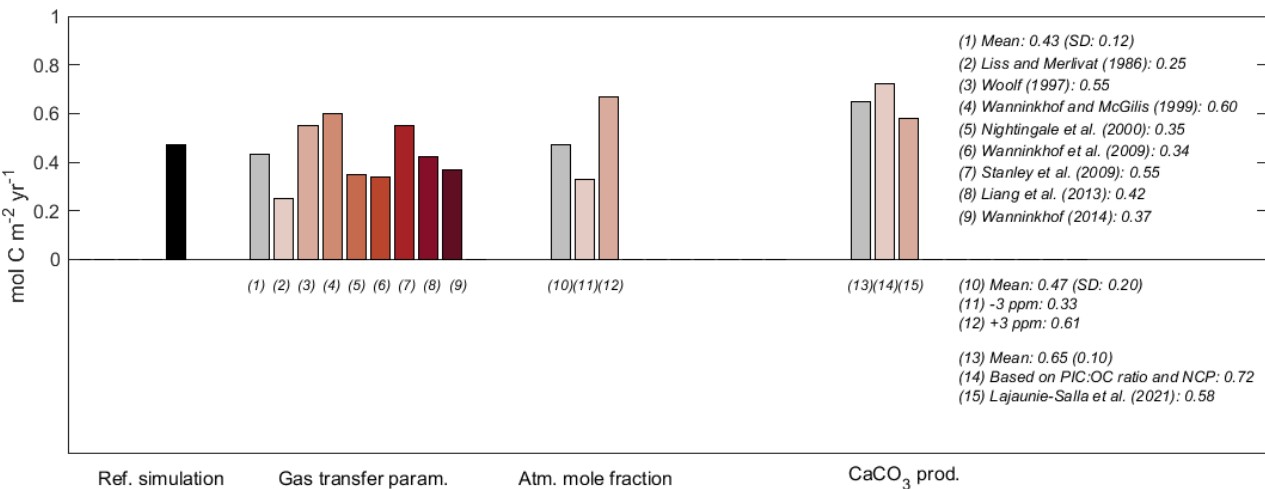

**Figure 14: Sensitivity tests to the parameterization of gas transfer velocity, the variability of the mole fraction of CO$_2$ in the atmosphere, and the calcification processes on the annual CO$_2$ air-sea flux estimate. The black bar indicates the annual estimate in the reference simulation, grey bars the mean value for each of the three sets of sensitivity tests.**