# Peer review of "Seasonal dynamics and annual budget of dissolved inorganic carbon in the northwestern Mediterranean deep convection region"

_Biogeosciences, 2022_

## Referee Comment (RC2)

**Review of Ulses et al. (2022): "Seasonal dynamics and annual budget of dissolved inorganic carbon in the northwestern Mediterranean deep convection region"**

**Summary**

This study by Ulses and coauthors presents a detailed carbon budget in the deep-convection area of the NW Mediterranean Sea for the period between September 2012 and September 2013. Using an ocean biogeochemistry model forced with daily output from a physical model, the authors find that their focus region is a moderate sink of atmospheric CO2 over the study period. In addition, by dividing the study area into the upper 150m and the deep ocean below that, they find that both physical and biological fluxes play an important role in controlling carbon fluxes across seasons.

Overall, the authors did a great job comparing their model results to existing observations and presenting the carbon budget of their study region in great detail. Therefore, the study is generally suitable for publication in Biogeosciences. However, I would like to raise several points, mostly regarding the presentation of the study, which should be addressed before the publication of this manuscript.

Please see the detailed explanation of these major points and all my detailed comments below.

**Main comments**

1. Introduction: Acknowledging that I am not 100% familiar with the literature concerning the Mediterranean Sea, the introduction appears to give a good summary of previous work. However, it fails to make the knowledge gap clear enough in my view. In its current form, it still reads too much as a collection of results from individual studies, making it hard for the reader to figure out what has not been addressed or what the short-comings in each of these previous studies are. As a result, I am a bit lost guessing what exactly the focus of the study by Ulses et al. is until L. 108. I suggest revising the introduction to more clearly state where the knowledge gaps are that are to be addressed in this new study.

2. Description of the model setup: While being methodologically sound from what I understood, the description of the model setup in section 2.1.2 is currently hard to follow. I suggest including a sketch in the revised version of the manuscript illustrating the downscaling approach and providing information on the initialization and run time of the simulations in each of the steps of the setup.

3. Result section 4.1: I admittedly found it quite difficult to keep up with all the provided details in this section. In general, I appreciate the detailed description of the figures, and I generally think the clear division into the different seasons is

good. However, this division means that the reader must constantly jump back and forth between Fig. 6-11, making it very important to have consistent structure and summarizing sentences throughout this section. While such summarizing sentences already exist for some of the seasons (see e.g., winter sub-period 2), they do not for others. I thus suggest that the authors carefully screen the result section again to structure the description of each season as consistently as possible and that they add clear summarizing sentences to each of the seasons. I encourage the authors to work on the paragraph structure (including topic sentences), as this will greatly improve the readability of this part of the manuscript. Lastly, since I really appreciated Fig. 12 as a summary for the annual mean budget, a similar figure for the seasonal budgets (=1 figure, 4 panels) would be a valuable addition to the paper and would serve as guidance for the reader throughout section 4.1.

4. For the sensitivity experiment regarding calcification: I was surprised to see an enhanced oceanic CO2 uptake relative to the reference case in the experiment accounting for calcification. For such an experiment, I would expect less oceanic CO2 uptake, given that the impact of calcification on alkalinity is twice that on DIC (thus increasing seawater pCO2 at the surface). Going back to your method section 2.1.4, I noticed that you only specified the impact of calcification on DIC – did you also include its impact on alkalinity in your sensitivity test? How did you parametrize dissolution at depth? I note that I realize that either way, this will not impact the outcomes of the main findings of this study, but if this was indeed a mistake, I suggest that the authors correct it.

5. Language: I spotted numerous (minor) grammar mistakes, e.g., related to prepositions (see detailed comments below). While this did not impact the readability much, I encourage the authors to carefully check the text again during the revisions.

**Detailed comments:**

L. 22: Maybe better: "seasonal and annual budget"?

L. 26: "reduction of *oceanic* CO2 uptake"

L. 27: I suggest rephrasing this sentence by being more specific: Aren't the physical fluxes (of DIC) always larger than the biological ones? How are both dominant?

L. 28: define "upper"

L. 29: I suggest replacing "air-sea flux" by "oceanic CO2 uptake"

L. 37: "comparable role *to…*" and "processes *for* carbon"; carbon transfer from where to where? Please specify.

L. 42: I suggest rephrasing to "taken up at the ocean surface"

L. 43: If there is an "on the other hand", I am immediately looking for "one the one hand". Maybe better: "at the same time" or "simultaneousy"?

L. 56: moderate phytoplankton bloom

L. 55-56: Is it typical in the Mediterranean science community to refer to the fall bloom as the "first" bloom? I realize this is a matter of defining the start of the growing season, but from all other regions globally, I am used to describing the bloom phenology starting with the strong first bloom in spring after nutrients were replenished in winter and a secondary typically weaker bloom in the fall.

L. 57: "nutrients *to* the euphotic layer"

L. 68: Please add a reference to Fig. 1.

L. 71: "which bring DIC-rich water to the surface"

L. 78: Maybe "complemented" instead of "enriched"?

L. 79: delete "fixed"

L. 81: "drives *an increase in surface* pCO2"

L. 83: model instead of modelling

L. 83-84: I am not sure what this approach means. Can you rephrase this part?

L. 86: biological instead of biology

L. 93: "limited *to*"

L. 92-94: This sentence was very confusing to read due to all the "or". Can you rephrase or split it into two?

L. 94-96: To me, this knowledge does not yet become clear enough from what is written up to this point. I suggest revising the introduction to more clearly highlight the knowledge gaps and why these matter.

L. 104: "by a *positive* net community production"

L. 108: Maybe better: "take advantage of" instead of "benefit from"

L. 112: Throughout the paper, you sometimes say "biological" and sometimes "biogeochemical". I suggest to consistently use one because from what I can see (please correct me if I am wrong), you are always referring to the same processes.

L. 120 & L. 125: Have the different models been evaluated in detail over the bigger study regions in any of these studies? It might help to explicitly state that for the interested reader.

L. 124: How are particle dynamics parametrized in the model? Given that sinking fluxes of biologically-derived particles are an important part of your study, some information on that will be helpful.

L. 131: Before looking up the cited references, it was unclear to me how the version before can resolve the cycling of carbon without including DIC. I suggest clarifying that only particulate organic carbon was included before.

L. 136: "is *the* respiration"

L. 142: "not the case *for* total alkalinity"

L. 146: Maybe add "throughout the water column" if that is what it is.

L. 147-149: Personally, I wouldn't call a paper from 2005 "present knowledge". There are several studies that, albeit of course not perfect, have parametrized it. Thus, I suggest rephrasing this part.

L. 149: "tests *on* this"

L. 165: Please add a reference to Fig. 1.

L. 167: I suggest adding "have been described in detail in X and Y and will be summarized here."

L. 169: It is unclear to me what "hydrodynamic analyses" are. Please clarify and possibly rephrase.

L. 167-175: I found the description of the steps rather difficult to follow. I think adding a flow chart detailing the different steps could help a lot.

L. 179: Given that the model simulates the negative charge and not alkalinity, did you correct the measured alkalinity to correspond to the model tracer? Please clarify.

L. 183: What is a "rigorous mathematical approach"? Please clarify or delete.

L. 189: You only specify what was used for winds here. What about other atmospheric forcing variables (e.g., radiation, humidity, precipitation etc.)? Please be complete.

L. 191: What I am missing here is a description on the model run time in each step. Also, in L. 179 you mention an initialization in summer 2011, while I think (if I understood correctly), the final model was run from September 2012 onwards. Could you clarify? My confusion on this point convinces me even more that a flow chart detailing the model setup procedure would help.

L. 193: I find "DIC flows" and "inventory variations" rather confusing. Maybe "DIC fluxes" and "inventory tendencies"? Please check throughout the text.

L. 194: "for at least 1 day"

L. 201: Given the title of this section, I wonder if Eq. 1 is better to be placed here. Additionally, I think at least the general budget equation (Eq. S1) should be moved to the main text.

L. 203: What do you mean by "internal variation"? Please clarify.

L. 215: Please add a reference to the respective Equation.

L. 216: "*as* 0.5"

L. 220: Please be precise: NCP does not appear as such in Eq. 1.

L. 221: Please state here what the parametrization by Lajaunie-Salla et al. (2021) is. Ideally, the reader should not have to look up other papers to understand what you're doing.

L. 225: sea surface

L. 284: I suggest adding "reflecting a" in front of "period"

L. 298: Does the southern zone include everything south of 41°N or is there a southern limit?

L. 299: I assume the depth profiles have been subsampled to only include the cruise locations shown in Fig. 3. Please clarify.

L. 324: Do you mean "alternating" instead of "alternative"?

L. 325: Where can the direction of the wind be seen? If this is previous knowledge for the region of interest and you therefore decided not to show this explicitly, please make sure it is introduced in the introduction for clarity.

L. 337: Unless I misread something, I think the minus sign should be omitted (the cumulative flux is positive according to Fig. 7).

L. 344: Do you mean "DIC concentration in the ML" or "the DIC flux into the ML"? Please clarify.

L. 441: "episodes of heat *gain*"

L. 466: To me, it is odd to call this flux biological production, when this is in fact remineralization/respiration. I understand why you do it and it is technically correct, but I still suggest rephrasing to avoid confusion.

L. 469: For consistency with how you described the biological component, it would be easier to read if you also reflected the sign convention in your wording here.

L. 474: I suggest deleting "an amount".

L. 486: Please see my comment on the abstract regarding "both dominate". I suggest to also rephrase here.

L. 489: Here and throughout the discussion section: Can you find more descriptive/informative section titles? It is incredibly useful to the reader if the title of each section already conveys information, i.e., ideally the main take-away message.

L. 490-502: As far as I can see, these are results. I am not convinced this part is necessary.

L. 508-509: This sentence is unclear to me. Can you rephrase?

L. 528-530: Here and throughout the text: Try to avoid 1-2 sentence paragraphs.

L. 552: Please see my major comment on these sensitivity experiments.

L. 566: Is there a "yr-1" missing? Additionally, it would help to provide the range based on your model here again to compare to the cited paper more easily.

L. 576: It might be more appropriate to say "physical transport".

L. 577: "the vertical DIC distribution"

L. 581: "greater magnitude" – Please specify the sign.

L. 582: "sea heat loss" Do you mean "ocean heat loss"? Please clarify.

L. 589: Please rephrase "DIC exchange flows".

L. 595: "as illustrated *in*"

L. 608: "slowed down" instead of "braked"

L. 617: "convection" instead of "convention"?

L. 633: "from" instead of "into"?

L. 634: "a lateral outflow"

L. 640-646: It would be a lot easier to compare to the findings of your studies, if you reported these numbers as flux densities instead of as integrated fluxes (or to here report your findings in the same integrated unit).

L. 648: "into" instead of "in"

L. 666: I suggest adding "…and rising atmospheric CO2 levels".

L. 680: budgets

L. 691: What exactly are the first and second part here? Please clarify.

L. 701: "subject to"

**Figures:**

Fig. 3: Please specify for what depth(s) the model output is shown here.

Fig. 4: I suggest adding a legend/title above each column.

Fig. 7: I suggest using the same colors for the same components in all panels, not only in a & b, but also in panel c. Additionally, it is unclear to me why you decided to show the seasonal averages only for the upper layer and not for the deeper layer. Please consider adding the extra panel for completeness.

Fig. 14: Please link the caption more clearly to the figure: which bar is which experiment? Only giving the reference requires the reader to be familiar with every single paper, which will not necessarily be the case (it certainly isn't the case for me).

All figures: Please double-check that the sign convention of all fluxes is defined in the respective caption.

**Supplementary material:**

Eq. S2: "DCA" is not defined in the text.

L. 29: How are sediment fluxes treated in the model? How large are they compared to the other components? Without any further information, it is difficult to judge for the reader to what extent this assumption impacts the role of vertical fluxes (which are treated as the residual and will therefore include any sedimentary contribution).

---

## Author Comment (AC1)

**Seasonal dynamics and annual budget of dissolved inorganic carbon in the northwestern Mediterranean deep convection region**

Caroline Ulses, Claude Estournel, Patrick Marsaleix, Karline Soetaert, Marine Fourrier, Laurent Coppola, Dominique Lefèvre, Franck Touratier, Catherine Goyet, Véronique Guglielmi, Fayçal Kessouri, Pierre Testor, Xavier Durrieu de Madron

**Responses to the comments of the anonymous Reviewer 1**

First we would like to warmly thank Reviewer 1 for his relevant and constructive comments which will help to improve the manuscript.

Answers to reviewers' comments are reported point by point. The questions and comments of the anonymous Reviewer 1 are in black, the answers in blue color and the modifications that we propose for the revised manuscript in red color in italic.

—

The authors investigated the dynamics of dissolved inorganic carbon in the deep convection area of the North-West Mediterranean Sea. The study was based on a good coupling between observations from mooring sites and cruises and 3 D coupled physical-biogeochemical model.
The main findings were that the area:
-was a moderate sink of CO2 (0.47 mol C m-2 yr-1) with an increase during the spring phytoplankton bloom, the air sea flux represented only 12% of net community production in the upper lever of the water column;
- both biological processes and physical transport (vertical and horizontal) played a dominant role in the annual DIC budge;
- winter ventilation had a reducing effect of on the atmospheric CO2 uptake;
- the region acted as a source of DIC for surface and intermediate waters.
Overall the approach is innovative and the results are relevant for a better understanding of the CO2 system dynamics in the NW Mediterranean Sea.

We thank Reviewer 1 for this positive general assessment.

I think that the discussion could be improved by a deeper comparison with one of the few other areas of the Mediterranean Sea where deep convection occurs as the Southern Adriatic Sea. The Adriatic Dense Water formation plays an important role for the sequestration and storage of the anthropogenic carbon, as the anthropogenic CO2 is transferred in the deep waters of the Eastern Mediterranean (Krasakopoulou et al., Deep Sea Res., 2011; Cantoni et al. Mar. Geol. 2016; Ingrosso et al. Deep Sea Res., 2017).

**Response:** We thank Reviewer 1 for this advice and the interesting references. We will add in the discussion a sub-section dedicated to the comparison of our results in terms of air-sea $CO_2$ flux (also in response to a comment of Reviewer 2). In this sub-section we will include comparisons with other studies carried out in the northwestern Mediterranean which was in Section 5.2 in the submitted manuscript, as well as a comparison with the other major deep convection region of the Mediterranean, the South Adriatic.

*"Our estimate is close to the annual flux estimated around 0.5 mol C m$^{-2}$ yr$^{-1}$ by Cossarini et al. (2021) in the South Adriatic Sea, the other deep convection area of the Mediterranean Sea."*

Furthermore, in section "Contribution of northwestern deep convection region to the carbon budget of the Mediterranean Sea", we will add a discussion on the exchanges between the two deep convection areas and the surrounding regions, as follows:

*"Finally, the transfer of DIC in intermediate waters, estimated here at 73 Tg C yr$^{-1}$, could represent up to 11% to the Mediterranean DIC export at the Gibraltar Strait towards the Atlantic Ocean, estimated to range between 680 and 1380 Tg C yr$^{-1}$ (Aït-Ameur and Goyet, 2006), and 100% of the net (difference between Atlantic surface inflow and Mediterranean outflow) DIC outflow, estimated between 20 and 70 Tg C yr$^{-1}$ (Huertas et al., 2009).*
*Our results for the northwestern deep convection area could be compared to those obtained in one of the other major deep water formation areas of the Mediterranean Sea, the Adriatic Sea. This latter has been shown to be a sink of atmospheric $CO_2$ (Cossarini et al., 2021) and a sequestration region of anthropogenic carbon (Krasakopoulou et al., 2011; Palmiéri et al., 2015; Hassoun et al., 2015; Ingrosso et al. 2017) as the study area (Touratier et al., 2016). In particular, experimental studies showed that the deep layer of the South Adriatic Sea was occupied by dense water rich in DIC and anthropogenic carbon formed in the deep convection regions of South Adriatic Pit and Pomo Pit, as well as on the northern shelf (Krasakopoulou et al., 2011; Cantoni et al, 2016; Ingrosso et al. 2017). The deep dense waters could be then transferred towards the Ionian Sea and the Mediterranean general deep circulation. Krasakopoulou et al. (2011) deduced from in situ measurements over February 1995 inorganic carbon fluxes crossing the Otranto Strait which connects the Ionian Sea to the South Adriatic Sea. They estimated that, on an annual basis, the Adriatic Sea could act as a sink of 314 Tg C yr$^{-1}$ of dissolved inorganic carbon for the Ionian Sea. This net flux resulted from an inflow of 1563 Tg C yr$^{-1}$, with 27% in the Levantine Intermediate Water, and an*

*outflow of 1249 Tg C yr$^{-1}$, with 21% in the Adriatic Deep Water. Thus, the northwestern Mediterranean deep convection region and the South Adriatic that includes shallower areas, could have opposite contributions in the deep and intermediate layers of the Mediterranean general circulation. However, our DIC budget assessment (as the budget studies in the Adriatic Sea) is limited to a single year and will need to be extended to a longer period to investigate in particular the question of carbon sequestration."*

Hassoun, A.E.R., Gemayel, A., Krasakopoulou, E., Goyet, E., Saab, C., Guglielmi, M.A.-A., Touratier, V., Falco, C, F., 2015. Acidification of the Mediterranean Sea from anthropogenic carbon penetration. Deep-Sea Res. I 102, 1–15. http://dx.doi.org/10.1016/j.dsr.2015.04.005.

Palmiéri, J., Orr, J.C., Dutay, J.C., Béranger, K., Schneider, A., Beuvier, J., Somot, S., 2015. Simulated anthropogenic CO2 uptake and acidification of the Mediterranean Sea. Biogeosciences 12, 781–802. http://dx.doi.org/10.5194/bg-12-781-2015.

In the Chapter 5.5 "Contribution of north-western deep convection region to the carbon budget of the Mediterranean Sea" the discussion could be improved by taking into account not only the modelling studies but also the experimental studies showing that the Adriatic continental platform acts as a sink for atmospheric CO2 (e. g.:  Turk et al., Jour. Geophys. Res., 2010; Cantoni et al., Est. Coast Shelf Sci.,2012; Catalano et al., Jour. Geophys. Res., 2014; Urbini et. al., Front. Mar. Sci., 2020).

**Response:** We thank Reviewer 1 for these pertinent references. We will complete the discussion on comparisons of the modeled CO$_2$ air-sea fluxes in the new sub-section 5.3, by expanding it to comparisons with the northern continental shelves which were identified as other water formation areas in the Mediterranean Sea:

*"Finally, it is also noteworthy that our estimate is found in the lower range of the annual flux estimated from experimental studies for the northern Adriatic and Aegean shelves, where dense water formation also takes place, and identified as sinks for atmospheric CO$_2$ most of the year and on an annual basis. With respect to the northern Adriatic shelf, our estimate is found close to the estimate of 0.4-0.5 mol C m$^{-2}$ yr$^{-1}$ for year 2014/15 by Urbini et al. (2020) and between about 2 to 4 folds lower than the estimates of 0.8-0.9 mol C m$^{-2}$ yr$^{-1}$ by Urbini et al. (2020) over the year 2016/17, of 1-1.1 mol C m$^{-2}$ yr$^{-1}$ by Catalano et al. (2014) and Cossarini et al. (2015) and of 2.2 mol C m$^{-2}$ yr$^{-1}$ by Cantoni et al. (2012) and Turk et al. (2013). Regarding the northern Aegean Sea, we found a lower winter flux than the one deduced from observations in February 2006 by Krasakopoulou et al. (2009) (4.9 in our study versus 8.6-14.7 mmol C m$^{-2}$ d$^{-1}$). Higher fluxes of CO$_2$ uptake exceeding 1 mol C m$^{-2}$ yr$^{-1}$ were also found for the northern shelves in the modeling studies of Cossarini et al. (2015; 2021). These higher fluxes could be explained by a lower seawater temperature in winter,*

*riverine nutrient inputs favoring intense primary production, and a transport of DIC associated with dense water outflow towards the deep basin (Cantoni et al., 2016; Ingrosso et al., 2017).''*

Cossarini, G., Querin, S., Solidoro, C.: The continental shelf carbon pump in the northern Adriatic Sea (Mediterranean Sea): influence of wintertime variability. Ecol. Model. 314, 118–134. http://dx.doi.org/10.1016/j.ecolmodel.2015.07.024, 2015.

Besides, in Section 4.2, we mentioned the higher air-sea $CO_2$ flux found in our model results on the shelf of the Gulf of Lion, another Mediterranean region where dense shelf water formation and cascading take place. Based on the model configuration implemented by Many et al. (2011), we plan to investigate the seasonal and interannual carbonate system dynamics on this shelf. We think that, in this future work, it would be very interesting to compare the seasonal and annual budget terms, as well as influences of northern winds and river inputs obtained for the Gulf of Lion shelf, with the observational previous works carried out on the northern Adriatic shelf both presenting many similar characteristics (as winter low temperature, continental winds, physical processes) but with a more enclosed morphology and higher river inputs for the northern Adriatic.

In the sensitivity tests including the carbonate production the authors used a PIC/POC ratio of 0.5 but according to the results reported in the cited paper of Miquel et al. (2011) the ratio is subject to wide interannual variations ranging from 0.31 to 0.78. It would be important to know how these natural variations would affect the sensitivity tests.

**Response:** Following the comment of Reviewer 1, we have performed sensitivity tests on carbonate production using the minimum and maximum values of the PIC:POC ratio reported by Miquel et al. (2011) to assess the impact of the natural variations of this ratio on the air-sea flux. The difference between air-sea fluxes computed for these two tests, for the first expression of carbonate production, is equal to 0.07 mol C $m^{-2}$ $yr^{-1}$. We will add the results of these tests in the discussion section on air-sea $CO_2$ flux, Figure 14, Table S1, Sect. 2.1.4 "Sensitivity tests" and in the conclusion. Moreover, we specify that a correction was made in the calculation of the rate of change of alkalinity (the excess negative charge state variable, see the answer to a following comment) that explains the difference in air-sea flux using the mean PIC:POC ratio given in Miquel et al. (2011) between the new version of the manuscript and the previous one.

Section 2.1.4 Sensitivity tests :

*"Following the study of Palevsky and Quay (2017), we first estimated it based on PIC:POC ratio and NCP. Miquel et al. (2011) estimated the PIC:POC ratio at 200 m depth varying between 0.31 and 0.78, with a mean value of 0.5, based on sediment trap measurements at the EMSO-DYFAMED site."*

Discussion section:
*"They show that not taken into account calcification processes could lead to an  overestimation of the annual air-sea $CO_2$ uptake by 16 to 57% with estimates of 0.29 mol C $m^{-2}$ $yr^{-1}$, based on the mean PIC:OC ratio given in Miquel et al. (2011) (varying between 0.19 and 0.36 mol C $m^{-2}$ $yr^{-1}$ based on the measured maximum and minimum PIC:OC ratios, respectively), and 0.40 mol C $m^{-2}$ $yr^{-1}$, based on the parametrization used in Lajaunie-Salla et al. (2021)."*

Conclusion:
*"Moreover, we displayed that neglecting calcification processes could lead to an overestimation by 16 to 57% of the annual uptake, highlighting the need for the refinement of the model in future studies."*

In the conclusion the authors state that the air-sea flux represents only 13% of the upper column Net Community Production (NCP) whereas in the chapter 5.4 that state that the flux represent 12% of NCP. The discrepancy should be solved.

**Response:** The correct value of the air-sea flux / net community production ratio is 13% (=0.47/3.74=12.57%). We apologize for the error. The value will be corrected in Section 5.4.

In the conclusion the authors states that the physical fluxes in the upper layer is of 3.3 mol C m-2 yr-1 but in the figure 12 the difference between the lateral DIC transport and trap vertical DIC transfer amounts to 3.4 mol C m-2 yr-1. The data should be checked.

**Response:** The values of the physical fluxes in the upper layer have been checked. The correct value of the net physical flux is 3.34 mol C $m^{-2}$ $yr^{-1}$. It results from the sum of a vertical input of 133.18 mol C $m^{-2}$ $yr^{-1}$ and of a lateral export of 129.84 mol C $m^{-2}$ $yr^{-1}$. Thus, the values in Figure 12 and in the text were correct.

The authors in the conclusion more clearly the in the discussion (L.547-552) state that calcification processes could lead to an underestimation by 23-58% of the annual uptake but the authors should take into account that the calcification processes although reducing the TCO2 will increase the pCO2 in seawater therefore counteracting the CO2 intake from the atmosphere.

**Response:** We thank Reviewer 1 for raising this point. We acknowledge that there was an error in the sensitivity test on calcification process, by omitting to take into account the process in the rate of change of alkalinity (excess negative charge denoted $\Sigma[-]$). We apologize for this error. We have corrected it by adding in the equation of the rate of change of alkalinity (excess negative charge denoted $\Sigma[-]$) the term of calcium carbonate production added in the DIC equation multiplied by 2 (Middelburg et al., 2019). In the new results, the $CO_2$ air-sea flux is reduced by 16% to 57% when the impact of calcification processes is modeled. We will modify the text and Figure 14 in the discussion section on the sensitivity tests on air-sea $CO_2$ flux, in Sect. 2.1.4 "Sensitivity tests" and in the conclusion.

Middelburg, J. J.: Marine Carbon Biogeochemistry A Primer for Earth System Scientists, Springer B., edited by Springer Briefs in Earth System Sciences, Springer Briefs in Earth System Sciences, 2019.

Section 2.1.4 Sensitivity tests:
*"Thus, if we assume the ratio of calcium carbonate production to NCP is close to PIC:TOC, we added in Eq. 1 a consumption term representing 36% (for the mean value of PIC:POC ratio, 22% and 58% for the minimum and maximum ratio values, respectively) of NCP. This term, multiplied by 2, was added in the equation of the rate of change of the excess negative charge."*

Discussion section:
*"Finally, sensitivity tests taking into account supplementary consumption terms in the equation of DIC and excess of negative charge for $CaCO_3$ precipitation (Sect. 2.1.4) were performed to assess its potential influence on air-sea $CO_2$ flux. They show that not taken into account calcification processes could lead to an  overestimation of the annual air-sea $CO_2$ uptake by 16 to 57% with estimates of 0.29 mol C $m^{-2}$ $yr^{-1}$, based on the mean PIC:POC ratio given by Miquel et al. (2011) (varying between 0.20 and 0.36 mol C $m^{-2}$ $yr^{-1}$ based on the maximum and minimum PIC:POC ratios, respectively), and 0.40 mol C $m^{-2}$ $yr^{-1}$, based on the parametrization used in Lajaunie-Salla et al. (2021)."*

[Figure]

*Figure 14: Sensitivity tests to the parameterization of gas transfer velocity, the variability of the mole fraction of $CO_2$ in the atmosphere, and the calcification processes on the annual $CO_2$ air-sea flux estimate. The black bar indicates the annual estimate in the reference simulation, grey bars the mean value for each of the three sets of sensitivity tests. For the sensitivity tests on the parametrization on gas transfer (from 2 to 9), relation with a quadratic (2), hybrid (3 to 5), cubic (6) wind speed dependency are, respectively, in light pink, yellow and orange, and relations that includes explicit bubbles parametrizations (7 to 9) are in dark pink. For the test (14) on calcification processes, the bar indicates the result found for the mean PIC:POC ratio, while the black line indicates the range using the minimum and maximum PIC:POC ratios.*

Conclusion:

*"Moreover, we displayed that  calcification processes could lead to an underestimation by 16 to 57% of the annual uptake, highlighting the need for the refinement of the model in future studies."*

The authors use the terms "biogeochemical flow "and "physical flow" which are not very appropriate terms as both are related to a mass flow of carbon generated by biological processes or by physical processes (advection, mixing, particle settling). I suggest to find a more appropriate alternative term e.g.:"physical transport".

**Response:** We acknowledge that the term "flow" was often inappropriate and apologize for this. We will replace this by another one when it is not appropriate.

**Specific comments**

**L. 49-50**, "is one of the region where deep convection occurs" a specific reference to the Southern Adriatic SAD should be added.

**Response:** The first sentences of the paragraph will be modified in order to add a specific reference to the Southern Adriatic, as follows:

*"Northern deep basins of the semi-enclosed Mediterranean Sea, i.e. the northwestern region (Fig. 1, Gulf of Lion and Ligurian Sea) and the South Adriatic, located at mid-latitudes  are ones of the regions where deep convection occurs (Ovchinnikov et al., 1985; Mertens and Shott, 1998; Manca and Bregant, 1998; Gačić et al., 2000; Béthoux et al., 2002)."*

Gačić, M., Manca, B.B., Mosetti, R., Scarazzato, P., Viezzoli, D., 2000. Deep water formation experiment in the Adriatic Sea. WWW Page, http://doga.ogs.trieste.it/doga/jwz/deep_water/mtpnews1.html

Manca, B. and D. Bregant: Dense water formation in the Southern Adriatic Sea during winter 1996. Rapp. Comm. Int. Mer Médit., 35, 176-177, 1998.

Ovchinnikov, I.M., Zats, V.I., Krivosheya V.G., Udodov A.I.: Formation of deep eastern Mediterranean water in the Adriatic Sea Oceanology, 25 (6) (1985), pp. 704-707, 1985.

**L. 370-37**, "upward flux of DIC into the upper layer of 41.40 mol C m-2…" The units of a mass flux should be used. They should be expressed as the mass of carbon that passes through a defined cross-sectional area over a period of time.

**Response:** We thank Reviewer 1 for raising this point. In Section 4.1, we give either the daily flux in mmol C $m^{-2}$ $d^{-1}$, or the cumulative flux, i.e. the flux, expressed as an amount of matter per surface per unit of time, multiplied by the considered period of time: mol C $m^{-2}$. In the revised version of the manuscript, we will add "cumulative" before "flux" here, and in the whole "4.1 Seasonal cycle of dissolved inorganic carbon" section when there was an oversight, and we will indicate the period over which the cumulative flux is calculated.

*"The physical fluxes at the limit of the upper layer of the deep convection area showed similar patterns as during autumn, with a cumulative upward flux of DIC into the upper layer of 41.40 mol C $m^{-2}$ over a 2.5 month period, almost counterbalanced by a cumulative lateral outflow of DIC of 40.44 mol C $m^{-2}$ in the upper layer and a cumulative lateral inflow of DIC of 39.90 mol C $m^{-2}$ in the deeper layer."*

**L. 390- 395, L. 469-470; L. 529**. same as above.

**Response:** In Section 4.1, the term "cumulative" was mentioned in L 390 and 394, we will add the period over which the time-integration of flux is done:

*"The cumulative biogeochemical flux reached -1.49 mol C m$^{-2}$ over this sub-period of 68 days."*

*" […] and finally cumulative air-sea flux reached 0.28 mol C m$^{-2}$ over the second winter sub-period of 68 days (a lower value and flux (3.1 versus 7.3 mmol C m$^{-2}$ d$^{-1}$) than over the first winter period)"*

L 469-470 and L529 and in this whole Section 4.2 , the annual fluxes are given. Therefore we will correct the unit of the fluxes by replacing "mol C m$^{-2}$" by "mol C m$^{-2}$ yr$^{-1}$":

*"Figure 12 shows a schematic of the annual budget of dissolved inorganic carbon in the deep convection zone. Our model results show that the deep convection area acted as a moderate $CO_2$ sink for the atmosphere on an annual scale, over the period September 2012-September 2013. We estimate that it absorbed 0.5 mol C m$^{-2}$ yr$^{-1}$ of atmospheric $CO_2$. This uptake of atmospheric $CO_2$ displayed spatial variability (Fig. 13). It was greater than 1 mol C m$^{-2}$ yr$^{-1}$ in the northern edge of the area along the Northern Current flowing over the Gulf of Lion continental slope, and became less than 0.25 mol C m$^{-2}$ yr$^{-1}$ in the western and eastern edge areas. One can notice that the annual rate remained lower than on the Gulf of Lion's shelf, which is beyond the scope of this study. Within the sea, biogeochemical processes induced an annual consumption of 3.7 mol C m$^{-2}$ yr$^{-1}$ of DIC in the upper layer and a  gain of 2.3 mol C m$^{-2}$ yr$^{-1}$ in the deeper layers.*
*Our estimate of net physical fluxes (lateral plus vertical) is an input of 3.3 mol C m$^{-2}$ yr$^{-1}$ in the upper layer and an export of -11.0 mol C m$^{-2}$ yr$^{-1}$ in the deeper layer. Specifically, the model indicates a vertical DIC supply of 133.2 mol C m$^{-2}$ yr$^{-1}$ from the deeper layer to the upper layer, partly offset by a lateral outflow of 129.8 mol C m$^{-2}$ yr$^{-1}$ in the upper layer and an inflow of 122.2 mol C m$^{-2}$ yr$^{-1}$ in the deeper layer. The budget in the deep layer masks different signs of physical fluxes: if the deeper layer is subdivided into an intermediate layer (150 m-800 m) and the deeper most layer (800 m-bottom), we find that the former, the intermediate layer, gained an amount 83.1 mol C m$^{-2}$ yr$^{-1}$ of DIC through vertical transport, while it lost 87.6 mol C m$^{-2}$ yr$^{-1}$ of DIC through lateral export. Finally, our model shows that the convection zone was a source of DIC of 8.7 mol C m$^{-2}$ yr$^{-1}$ for the rest of the western Mediterranean Sea. While the DIC inventory in the upper layer remained stable (decrease of 0.07 mol C m$^{-2}$ yr$^{-1}$), the DIC inventory in the deeper layer experienced a decrease of 8.7 mol C m$^{-2}$ yr$^{-1}$. This loss occurred mainly during deep convection, and to a lesser extent during the preconditioning period (in autumn and early winter).*
*Finally, we complete the inorganic carbon budget with the labile organic carbon fluxes (refractory organic carbon is not considered in our model). We estimate that during the studied period a lateral export of organic carbon of 1.1 mol C m$^{-2}$ yr$^{-1}$ and 0.3 mol C m$^{-2}$ yr$^{-1}$ took place in the upper and deeper layers, respectively. The modeled downward export of organic carbon amounted to 2.3 mol C m$^{-2}$ yr$^{-1}$."*

**L. 452**, "the DIC drawdown due to biological processes decreases and net DIC production events took place": could the authors specify which are the processes driving the DIC production events.

**Response:** Since we haven't deeply analyzed specifically these short events, we will remove this and we propose to rephrase this sentence in the revised manuscript, as follows:

*"From August onwards, the DIC drawdown due to* biogeochemical  *processes decreased,* primary production rate becoming close to respiration rate *(Fig. 6h)."*

**L. 465** "an annual consumption of 3.7 mol C m-2 of DIC": the unit of time is lacking.

**Response:** We agree, we apologize for this oversight. As mentioned in a previous response, the sentence will be modified as follows:

*"Within the sea, biogeochemical processes induced an annual consumption of 3.7 mol C $m^{-2}$ $yr^{-1}$ of DIC in the upper layer and*  gain *of 2.3 mol C $m^{-2}$ $yr^{-1}$ in the deeper layers. "*

**L.549-552.** This sentence is not clear and the CO2 production during calcification should be taken into account.

**Response:** As mentioned in a previous response, we have corrected the sensitivity tests on calcification processes. The sentence will be modify as follows:

*"They show that* not taken into account *calcification processes could lead to an*  overestimation *of the annual air-sea $CO_2$ uptake by* 16 *to* 57% *with estimates* 0.29 *mol C $m^{-2}$ $yr^{-1}$, [...]"*

**L. 578** physical flow? Do the authors mean physical transport?

**Response:** "physical flow" will be replaced by "physical transport".

*"[...] and highlights that physical* transports *play a crucial role in the DIC budget in this highly energetic region."*

**L. 589** DIC exchange flows? Do you mean DIC flows?

**Response:** "DIC exchange flow" will be replaced by "DIC fluxes at the limits of the zone".

*"Moreover, a detailed calculation of the water and DIC*  fluxes at the limits of the deep convection area *allowed us to [...]"*

**Fig. 7.** The units for fluxes are expressed as an inventory: mol C m-2. The mass fluxes should be expressed as the mass of carbon that passes through a defined cross-sectional area over a period of time e.g. mol C m$^{-2}$ y$^{-1}$.

**Response:** Figure 7 shows the cumulative fluxes, i.e. the fluxes expressed as the amount of matter per surface and per unit of time, multiplied by the time period over which the accumulation is calculated. In response to a comment of Reviewer 2 we will remove panel (b) with the cumulative seasonal fluxes. In the remaining panels, the cumulative flux at a day d is the flux, expressed in mol C m$^{-2}$ d$^{-1}$ multiplied by the number of days between the 1$^{st}$ September 2012 and day d.  We will correct the titles of the panels by adding "cumulative" before "fluxes" and complete the caption.

[Figure]

*Figure 7: Time series of cumulative variation in dissolved inorganic carbon (DIC) inventory (black) and cumulative air–sea (red), physical transfer (light and dark blue), and biogeochemical (bright and brown green) flux of dissolved inorganic carbon in the (a) upper (surface to 150 m) and (b) deeper (150 m to bottom) layers, from September 2012 to September 2013. Unit: mol C m$^{-2}$. Positive values represent inputs for the deep convection area. The blue shaded area corresponds to the deep convection period (period when spatially averaged mixed layer depth > 100 m). The DIC inventory on 1$^{st}$ September 2012 was 353 and 5560 mol C m$^{-2}$ in the upper and deeper layers, respectively. The cumulative flux at a day d is the time-integrated flux over the period from the 1$^{st}$ September 2012 to day d.*

**Fig. 12.** The data represented are inventories or fluxes in the latter case they should be expressed as the mass of carbon that passes through a defined cross-sectional area over a period of time e.g. mol C m⁻² y⁻¹.

**Response:** We apologize for this error. The unit on Figure 12 and in its caption will be corrected as follows:

[Figure]

*"Figure 12: Scheme of the annual carbon budget for the period September 2012 to September 2013 from the coupled model SYMPHONIE-Eco3M-S. Fluxes are indicated in mol C m⁻² yr⁻¹."*

---

## Author Comment (AC2)

**Seasonal dynamics and annual budget of dissolved inorganic carbon in the northwestern Mediterranean deep convection region**

Caroline Ulses, Claude Estournel, Patrick Marsaleix, Karline Soetaert, Marine Fourrier, Laurent Coppola, Dominique Lefèvre, Franck Touratier, Catherine Goyet, Véronique Guglielmi, Fayçal Kessouri, Pierre Testor, Xavier Durrieu de Madron

**Responses to the comments of the anonymous Reviewer 2**

First we would like to warmly thank Reviewer 2 for his/her relevant and constructive comments which will help to improve the manuscript.

Answers to reviewers' comments are reported point by point. The questions and comments of the anonymous Reviewer 2 are in black, the responses in blue and the modifications that we propose for the revised manuscript in red in italic.

Review of Ulses et al. (2022): "Seasonal dynamics and annual budget of dissolved inorganic carbon in the northwestern Mediterranean deep convection region"

Summary This study by Ulses and coauthors presents a detailed carbon budget in the deep convection area of the NW Mediterranean Sea for the period between September 2012 and September 2013. Using an ocean biogeochemistry model forced with daily output from a physical model, the authors find that their focus region is a moderate sink of atmospheric CO2 over the study period. In addition, by dividing the study area into the upper 150m and the deep ocean below that, they find that both physical and biological fluxes play an important role in controlling carbon fluxes across seasons.

Overall, the authors did a great job comparing their model results to existing observations and presenting the carbon budget of their study region in great detail. Therefore, the study is generally suitable for publication in Biogeosciences. However, I would like to raise several points, mostly regarding the presentation of the study, which should be addressed before the publication of this manuscript.

Please see the detailed explanation of these major points and all my detailed comments below.

We appreciate this positive general assessment.

**Main comments**

1. Introduction: Acknowledging that I am not 100% familiar with the literature concerning the Mediterranean Sea, the introduction appears to give a good summary of previous work. However, it fails to make the knowledge gap clear enough in my view. In its current form, it still reads too much as a collection of results from individual studies, making it hard for the reader to figure out what has not been addressed or what the short-comings in each of these previous studies are. As a result, I am a bit lost guessing what exactly the focus of the study by Ulses et al. is until L. 108. I suggest revising the introduction to more clearly state where the knowledge gaps are that are to be addressed in this new study.

Response: As suggested by Reviewer 2, we will revise the introduction to more clearly point the gaps in the previous studies on DIC cycle and give earlier in the text the objective of the present study, as follows:

[revised manuscript text omitted]

2. Description of the model setup: While being methodologically sound from what I understood, the description of the model setup in section 2.1.2 is currently hard to follow. I suggest including a sketch in the revised version of the manuscript illustrating the downscaling approach and providing information on the initialization and run time of the simulations in each of the steps of the setup.

Response: We will clarify this description by adding a figure in Supplementary Material to show (1) the domain of the two coupled physical-biogeochemical models, i.e. the parent and child models, and (2) a scheme of the dowscalling strategy.

[Figure]

*Figure S1: (a) Domain and bathymetry (m) of the forcing coupled NEMO-Eco3MS Mediterranean model (red contour) and of the coupled SYMPHONIE-Eco3MS western sub-basin model (blue contour). (b) Scheme of the downscaling strategy from the Mediterranean Sea to the western sub-basin.*

We will also add in Section 2.1.3 "Model setup" of the revised manuscript the run time of each of the three simulations and will move the description of the initialization in step 1b before describing step 2. We hope the description of the downscaling strategy will be clearer after these additional elements and modifications:

*"The implementation of the hydrodynamic simulation and the strategy of downscaling from the Mediterranean Basin to the western sub-basin scale in three stages have been described* in detail in *Estournel et al. (2016) and Kessouri et al (2017) and* will be summarized here (Fig. S1):
*- In a first step* (named step 1a)*, the SYMPHONIE hydrodynamic model, implemented over the Western Mediterranean Sea (delimited by blue lines in the insert of Fig. 1), was initialized and forced at its lateral boundaries with daily hydrodynamic analyses of the configuration PSY2V4R4, based on the NEMO ocean model at a resolution of 1/12° over the Mediterranean Sea by the Mercator Ocean International operational system (Lellouche et al., 2013). This simulation was performed from 1$^{st}$ August 2012 to 31 October 2013.*

*- In parallel (step 1b), the biogeochemical model was computed, in offline mode, at the Mediterranean basin scale, on the same 1/12° NEMO grid (delimited by orange lines in the insert of Fig. 1), using the same NEMO hydrodynamic fields as those used by the SYMPHONIE simulation in step 1a. This simulation was performed from 15 June 2011 to 15 November 2013. The carbonate system module in this configuration was initialized using mean values of dissolved inorganic carbon, total alkalinity observations carried out in 2011 from the Meteor M84/3 (Alvarez et al., 2014), CASCADE (CAscading, Surge, Convection, Advection and Downwelling Events, Touratier et al., 2016), and MOOSE-GE cruises (Testor et al., 2010) and at the EMSO-DYFAMED mooring (Coppola et al., 2021) and BOUSSOLE buoy (Golbol et al., 2020) sites, over bio-regions defined in Kessouri (2015), based on Lavezza et al. (2011). We deduced the concentration of the excess negative charge based on nutrient concentrations initialized using the Medar/Medatlas database as in Kessouri et al. (2017). Recently, Davis and Goyet (2021) described a method based upon the property variability, to precisely quantify the uncertainties at any point of an interpolated data field. This approach could be used in the near-future to improve both the at-sea sampling strategy (Guglielmi et al., 2022a; 2022b), and the accuracy of model initialization.*

*- In a second time (step 2), the Eco3M-S biogeochemical model was implemented over the Western Mediterranean Sea, using the grid and the hydrodynamics fields of the aforementioned SYMPHONIE simulation (step 1a) in offline mode. This simulation was performed from 15 August 2012 to 30 September 2013. The initial state and lateral boundary conditions of the biogeochemical fields are provided by the biogeochemical simulation of the Mediterranean Basin of step 1b. "*

3. Result section 4.1: I admittedly found it quite difficult to keep up with all the provided details in this section. In general, I appreciate the detailed description of the figures, and I generally think the clear division into the different seasons is good. However, this division means that the reader must constantly jump back and forth between Fig. 6-11, making it very important to have consistent structure and summarizing sentences throughout this section. While such summarizing sentences already exist for some of the seasons (see e.g., winter sub-period 2), they do not for others. I thus suggest that the authors carefully screen the result section again to structure the description of each season as consistently as possible and that they add clear summarizing sentences to each of the seasons. I encourage the authors to work on the paragraph structure (including topic sentences), as this will greatly improve the readability of this part of the manuscript. Lastly, since I really appreciated Fig. 12 as a summary for the annual mean budget, a similar figure for the seasonal budgets (=1 figure, 4 panels) would be a valuable addition to the paper and would serve as guidance for the reader throughout section 4.1.

**Response:** In the revised manuscript, we will structure the description of the different seasonal sections as consistently as possible, with a description of the (1) atmospheric and hydrodynamical situation, then of the (2) biogeochemical fluxes, (3) physical fluxes, (4)

air-sea fluxes, and (5) finally of the resulting variation of DIC content, and a summary of the budget in the upper layer. Besides, we will include in Figure 7 a panel with a similar figure as Figure 12 for each season, and remove from Figure 7 the panel (b) to avoid repetitions with the new sub-figure. We will merge Figures 8 and 9 to decrease the number of figures in this part.

[Figure]

*"Figure 7c: Scheme of cumulative seasonal fluxes in mol C m⁻² over the respective periods (fall: 88 days, winter: 116 days, spring : 74 days and summer: 87 days). Resp. stands for respiration and GPP for gross primary production."*

4. For the sensitivity experiment regarding calcification: I was surprised to see an enhanced oceanic CO2 uptake relative to the reference case in the experiment accounting for calcification. For such an experiment, I would expect less oceanic CO2 uptake, given that the impact of calcification on alkalinity is twice that on DIC (thus increasing seawater pCO2 at the surface). Going back to your method section 2.1.4, I noticed that you only specified the impact of calcification on DIC – did you also include its impact on alkalinity in your sensitivity test? How did you parametrize dissolution at depth? I note that I realize that either way, this

will not impact the outcomes of the main findings of this study, but if this was indeed a mistake, I suggest that the authors correct it.

**Response:** We thank Reviewer 2 for raising this point. We acknowledge there was an error in the sensitivity test on calcification process, by omitting to take into account the process in the rate of change of alkalinity (excess negative charge denoted $\Sigma[-]$). We apologize for this error. We have corrected it by adding in the equation of the rate of change of alkalinity (excess negative charge denoted $\Sigma[-]$) the term of calcium carbonate production added in the DIC equation multiplied by 2 (Middelburg et al., 2019). In the new results, the air-sea flux could be reduced by 16% to 57% in considering calcification processes. We will modify the text and Figure 14 in the discussion section on the sensitivity tests on air-sea $CO_2$ flux, in Sect. 2.1.4 "Sensitivity tests" and in the conclusion. The dissolution at depth was not taken into account in these sensitivity tests.

Middelburg, J. J.: Marine Carbon Biogeochemistry A Primer for Earth System Scientists, Springer B., edited by Springer Briefs in Earth System Sciences, Springer Briefs in Earth System Sciences, 2019.

Section 2.1.4 Sensitivity tests :
*"Thus, if we assume the ratio of calcium carbonate production to NCP is close to PIC:TOC, we added in Eq. 1 a consumption term representing 36% (for the mean value of PIC:POC ratio, 22% and 58% for the minimum and maximum ratio values, respectively) of NCP. This term, multiplied by 2, was added in the equation of the rate of change of the excess negative charge."*

Discussion section:
*"Finally, sensitivity tests taking into account supplementary consumption terms in the equation of DIC and excess of negative charge for $CaCO_3$ precipitation (Sect. 2.1.4) were performed to assess its potential influence on air-sea $CO_2$ flux. They show that not taken into account calcification processes could lead to an  overestimation of the annual air-sea $CO_2$ uptake by 16 to 57% with estimates 0.29 mol C m$^{-2}$ yr$^{-1}$, based on the mean PIC:POC ratio given by Miquel et al. (2011) (varying between 0.20 and 0.36 mol C m$^{-2}$ yr$^{-1}$ based on the maximum and minimum PIC:POC ratios, respectively), and 0.40 mol C m$^{-2}$ yr$^{-1}$, based on the parametrization used in Lajaunie-Salla et al. (2021)."*

[Figure]

*Figure 14: Sensitivity tests to the parameterization of gas transfer velocity, the variability of the mole fraction of $CO_2$ in the atmosphere, and the calcification processes on the annual $CO_2$ air-sea flux estimate. The black bar indicates the annual estimate in the reference simulation, grey bars the mean value for each of the three sets of sensitivity tests. For the sensitivity tests on the parametrization on gas transfer (from 2 to 9), relation with a quadratic (2), hybrid (3 to 5), cubic (6) wind speed dependency are, respectively, in light pink, yellow and orange, and relations that includes explicit bubbles parametrizations (7 to 9) are in dark pink. For the test (14) on calcification processes, the bar indicates the result found for the mean PIC:POC ratio, while the black line indicates the range using the minimum and maximum PIC:POC ratios.*

Conclusion:

*"Moreover, we displayed that  calcification processes could lead to an estimation by 16 to 57% of the annual uptake, highlighting the need for the refinement of the model in future studies."*

5. Language: I spotted numerous (minor) grammar mistakes, e.g., related to prepositions (see detailed comments below). While this did not impact the readability much, I encourage the authors to carefully check the text again during the revisions.

**Response:** We warmly thank Reviewer 2 for all the grammar corrections and apologize for these errors. We will carefully check the text again.

**Detailed comments:**

**L. 22:** Maybe better: "seasonal and annual budget"?

**Response:** The sentence will be changed as suggested in the revised manuscript.

**L. 26:** "reduction of oceanic CO2 uptake"

**Response:** The sentence will be changed as suggested in the revised manuscript.

**L. 27:** I suggest rephrasing this sentence by being more specific: Aren't the physical fluxes (of DIC) always larger than the biological ones? How are both dominant?

**Response:** The vertical and horizontal fluxes are always both one order of magnitude higher than the net biogeochemical fluxes (respiration minus primary production). However, the net physical flux, i.e. vertical flux plus lateral flux, is of the same order of magnitude as the net biogeochemical flux for each season ; it is higher (in intensity) than the biogeochemical flux in winter and summer (cumulative fluxes: 4.45 versus -1.53 mol C m$^{-2}$ over the winter period, 0.79 versus -0.57 mol C m$^{-2}$ over the summer period), and smaller (in intensity) in fall and spring (cumulative fluxes: 0.49 versus 0.56 mol C m$^{-2}$ over the fall period, -0.80 versus -2.19 mol C m$^{-2}$ over the summer period). At the annual scale, the net physical flux is 3.3 mol C m$^{-2}$ yr$^{-1}$ and represents 88% of the net biogeochemical flux, while the air-sea flux is one order of magnitude smaller and represents 13% of the biogeochemical flux. We will rephrase the sentence, as follows:

*"We highlight the  major role in the annual dissolved inorganic carbon budget of both  biogeochemical and physical fluxes that amount to 3.3 mol C m$^{-2}$ yr$^{-1}$ and -3.7 mol C m$^{-2}$ yr$^{-1}$, respectively, and are one order of magnitude higher than the $CO_2$ air-sea flux ."*

**L. 28:** define "upper"

**Response:** We will specify "upper" as follows:

*"The upper layer (from surface to 150 m depth) of the northwestern deep convection region [...]"*

**L. 29:** I suggest replacing "air-sea flux" by "oceanic CO2 uptake"

**Response:** The sentence will be changed as suggested in the revised manuscript.

**L. 37:** "comparable role to…" and "processes for carbon"; carbon transfer from where to where? Please specify.

**Response:** The sentence will be changed for more clarity, as follows:

"*Physical mechanisms can* *quantitatively* *play a comparable role to that of*  biogeochemical *processes on*  CO$_2$ air-sea flux *at regional and global scales …*"

**L. 42:** I suggest rephrasing to "taken up at the ocean surface"

**Response:** The sentence will be changed as suggested in the revised manuscript.

**L. 43**: If there is an "on the other hand", I am immediately looking for "one the one hand". Maybe better: "at the same time" or "simultaneousy"?

**Response:** "on the other hand' will be replaced by "furthermore" in the revised manuscript.

**L. 56:** moderate phytoplankton bloom

**Response:** The sentence will be changed as suggested in the revised manuscript.

**L. 55-56**: Is it typical in the Mediterranean science community to refer to the fall bloom as the "first" bloom? I realize this is a matter of defining the start of the growing season, but from all other regions globally, I am used to describing the bloom phenology starting with the strong first bloom in spring after nutrients were replenished in winter and a secondary typically weaker bloom in the fall.

**Response:** Some of previous studies which determined the date of the onset in the Mediterranean Sea (Bernardello et al., 2012; Lavigne et al., 2013) were based on the work by Henson et al. (2009) who determined the bloom start in the North Atlantic Sea by adjusting the method of Siegel et al (2002) and considering the 1$^{st}$ September as the beginning of the annual period, to capture the start of the subtropical bloom that occurs in autumn. Using satellite derived-chlorophyll data, Lavigne et al. (2013) found a bloom starting in autumn (late November / early December) in all the Mediterranean bioregions defined by D'Ortenzio and Ribera d'Alcala (2009). Kessouri et al. (2018) calculated the date of the bloom onset in the Western Mediterranean Sea using the same biogeochemical model (without the

carbonate system module) as used in this study. They also found a start bloom in autumn for the three considered regions (deep convection zone, shallow convection zone and stratified region). In their results, contrary to the two other regions, in the deep convection region the bloom is interrupted during the deep mixing period and a second bloom start was found when the water column stratified. However, in other studies, the description of the annual chlorophyll cycle is described from January to December and thus the spring bloom is mentioned before the autumnal bloom (Bosc et al., 2004). Our study was performed in the continuity of Kessouri et al. (2018) and thus we preferred keeping the same annual period. In the revised version, we will remove "first" and "secondary" from the sentence.

Henson, S. A., Dunne, J., & Sarmiento, J.: Decadal variability in North Atlantic phytoplankton blooms. Journal of Geophysical Research, 114, C04013. https://doi.org/10.1029/2008JC005139, 2009.

Lavigne, H., D'Ortenzio, F., Migon, C., Claustre, H., Testor, P., Ribera d'Alcala, M., et al.: Enhancing the comprehension of mixed layer depth control on the Mediterranean phytoplankton phenology. Journal of Geophysical Research: Oceans, 118, 3416–3430. https://doi.org/10.1002/jgrc.20251, 2013.

Siegel, D. A., Doney, S. C., & Yoder, J. A.: The North Atlantic spring phytoplankton bloom and Sverdrup's critical depth hypothesis. Science, 296, 730–733. https://doi.org/10.1126/science.1069174, 2002.

**L. 57:** "nutrients to the euphotic layer"

**Response:** The sentence will be changed as suggested in the revised manuscript.

**L. 68**: Please add a reference to Fig. 1.

**Response:** The reference to Fig. 1 will be added in the sentence as suggested in the revised manuscript.

**L. 71:** "which bring DIC-rich water to the surface"

**Response:** The sentence will be changed as suggested in the revised manuscript.

**L. 78**: Maybe "complemented" instead of "enriched"?

**Response:** The sentence will be changed as suggested in the revised manuscript.

**L. 79**: delete "fixed"

**Response:** The sentence will be changed as suggested in the revised manuscript.

**L. 81:** "drives an increase in surface pCO2"

**Response:** The sentence will be changed as suggested in the revised manuscript.

**L. 83**: model instead of modelling

**Response:** The sentence will be changed as suggested in the revised manuscript.

**L. 83-84:** I am not sure what this approach means. Can you rephrase this part?

**Response:** In response to one of the main comments and in revising the introduction, this sentence has been simplified as follows:

*"D'Ortenzio et al. (2008) and Cossarini et al. (2021), based on a 1D model and a 3D model, respectively, found that the whole deep convection region is a major sink of atmospheric $CO_2$ in the open Mediterranean Sea"*

D'Ortenzio et al. (2008) implemented a 1D model in cells of 0.5° x 0.5° horizontal resolution covering the Mediterranean Sea, with no lateral connection between the cells.

**L. 86**: biological instead of biology

**Response:** The sentence will be changed as suggested in the revised manuscript.

**L. 93:** "limited to"

**Response:** The sentence will be changed as suggested in the revised manuscript.

**L. 92-94:** This sentence was very confusing to read due to all the "or". Can you rephrase or split it into two?

**Response:** In revising the introduction, this sentence has been merged with the following one, as follows:

*"In the previous studies, the 3D dynamics of the $CO_2$ system over an annual cycle has never been specifically explored for the whole northwestern deep convection region and a complete DIC budget is still lacking for this region."*

**L. 94-96**: To me, this knowledge does not yet become clear enough from what is written up to this point. I suggest revising the introduction to more clearly highlight the knowledge gaps and why these matter.

**Response:** As we answered at one of the main comments, we will revise the introduction to more clearly highlight the knowledge gaps.

**L. 104:** "by a positive net community production"

**Response:** The sentence will be changed as suggested in the revised manuscript.

**L. 108:** Maybe better: "take advantage of" instead of "benefit from"

**Response:** The sentence will be changed as suggested in the revised manuscript.

**L. 112:** Throughout the paper, you sometimes say "biological" and sometimes "biogeochemical". I suggest to consistently use one because from what I can see (please correct me if I am wrong), you are always referring to the same processes.

**Response:** The term "biological" will be replaced by "biogeochemical" throughout the paper when referring to the same processes.

**L. 120 & L. 125:** Have the different models been evaluated in detail over the bigger study regions in any of these studies? It might help to explicitly state that for the interested reader.

**Response:** The biogeochemical model implemented in the whole Mediterranean and forced by the outputs of the hydrodynamic model NEMO operated by Mercator was assessed by Kessouri (2015) in terms of spatial and temporal surface chlorophyll and vertical distribution of chlorophyll and inorganic nutrient. This will be specified in Section 2.1.1 "The coupled hydrodynamic-biogeochemical-chemical model". The western Mediterranean biogeochemical model was assessed over the western Mediterranean in Kessouri et al. (2018) through comparisons with satellite chlorophyll data.

*"The model has been used to study biogeochemical processes in the NW (northwestern) Mediterranean deep convection area (Herrmann et al., 2013; Auger et al., 2014; Ulses et al., 2016; 2021; Kessouri et al., 2017; 2018) and in the whole Mediterranean Sea (Kessouri, 2015)."*

Kessouri, F.: Cycles biogéochimiques de la Mer Méditerranée : processus et bilans, Ph.D. thesis, Université Toulouse 3, 2015.

**L. 124:** How are particle dynamics parametrized in the model? Given that sinking fluxes of biologically-derived particles are an important part of your study, some information on that will be helpful.

**Response:** To take into account particle dynamics in the model, we consider a constant settling velocity, $w_s$, for the slow and fast sinking particulate organic matter and for micro-phytoplankton. The values of the settling velocity will be given in Section 2.1.1. The settling of particles is taken into account in the following advection-diffusion equation allowing the calculation of the "physical" rate of change of the concentration $C$, the concentration of each biogeochemical state variable:

$$\frac{\partial C}{\partial t} + \frac{\partial uC}{\partial x} + \frac{\partial vC}{\partial y} + \frac{\partial (w - w_s)C}{\partial z} = \frac{\partial}{\partial z}\left(K_z \frac{\partial C}{\partial z}\right) + F_C$$

where u, v and w are the three components of the current velocity, $K_z$ is the vertical diffusivity and $F_c$ is the source or sink term from rivers, atmosphere and sediment.

*"Particulate organic detritus and microphytoplankton have a constant settling velocity (1 m $d^{-1}$ for slow sinking detritus and microphytoplankton, and 90 m $d^{-1}$ for fast sinking detritus)."*

**L. 131:** Before looking up the cited references, it was unclear to me how the version before can resolve the cycling of carbon without including DIC. I suggest clarifying that only particulate organic carbon was included before.

**Response:** As suggested by Reviewer 2 we will add a sentence to clarify this point.

*"In previous versions of the model, particulate and dissolved organic carbon was considered, but the dynamics of dissolved inorganic carbon was not described."*

**L. 136:** "is the respiration"

**Response:** The sentence will be changed as suggested in the revised manuscript.

**L. 142:** "not the case for total alkalinity"

**Response:** The sentence will be changed as suggested in the revised manuscript.

**L. 146**: Maybe add "throughout the water column" if that is what it is.

**Response:** We will add "throughout the water column" in this sentence as suggested in the revised manuscript.

**L. 147-149:** Personally, I wouldn't call a paper from 2005 "present knowledge". There are several studies that, albeit of course not perfect, have parametrized it. Thus, I suggest rephrasing this part.

**Response:** This part will be rephrased, as follows:

*"The  $CaCO_3$ precipitation  is difficult to parametrize  in a model (Aumont et al., 2005). However, we are aware that future refinements will have to take it [...]"*

**L. 149:** "tests on this"

**Response:** The sentence will be changed as suggested in the revised manuscript.

**L. 165:** Please add a reference to Fig. 1.

**Response:** A reference to Fig. 1 will be added in the revised manuscript.

**L. 167:** I suggest adding "have been described in detail in X and Y and will be summarized here."

**Response:** The sentence will be changed as suggested in the revised manuscript.

**L. 169:** It is unclear to me what "hydrodynamic analyses" are. Please clarify and possibly rephrase.

**Response:** Hydrodynamic analyses represent here the hydrodynamic solutions from the NEMO numerical model computed with the Mercator near real time configuration PSY2V2R4 that embeds assimilation of data in order to constrain and bring realism to the numerical solution. We will replace "analyses" by "fields" in the text.

**L. 167-175:** I found the description of the steps rather difficult to follow. I think adding a flow chart detailing the different steps could help a lot.

**Response:** As answered to one of the main comments, to clarify this point we will add a figure with a scheme of the 3 steps in Supplementary material.

**L. 179:** Given that the model simulates the negative charge and not alkalinity, did you correct the measured alkalinity to correspond to the model tracer? Please clarify.

**Response:** We apologize for the confusion. Yes, we deduced the initial values of the excess negative charge using measurements of total alkalinity and nutrients concentrations based on Eq. 2. We will add a sentence to clarify this point:

*"To deduce the excess negative charge from total alkalinity (Eq. 2), we also used the nutrient concentration data from the Medar/Medatlas database as in Kessouri et al. (2017)."*

**L. 183:** What is a "rigorous mathematical approach"? Please clarify or delete.

**Response:** We will delete "rigorous mathematical approach" and rephrase the sentence.

*"Recently, Davis and Goyet (2021)  described a  method based upon the property variability, to precisely quantify the uncertainties at any point of an interpolated data field."*

**L. 189:** You only specify what was used for winds here. What about other atmospheric forcing variables (e.g., radiation, humidity, precipitation etc.)? Please be complete.

**Response:** We will be complete and add all the other forcing variables needed for the gas transfer velocity knowing that the hydrodynamic model uses other atmospheric variables such as air temperature, precipitation, longwave and shortwave radiation:

*"To compute the gas transfer velocity, we used the 3-hour wind speed, pressure, and humidity provided by the ECMWF model on a 1/8° grid, in consistency with the hydrodynamic simulation."*

**L. 191:** What I am missing here is a description on the model run time in each step. Also, in L. 179 you mention an initialization in summer 2011, while I think (if I understood correctly), the final model was run from September 2012 onwards. Could you clarify? My confusion on this point convinces me even more that a flow chart detailing the model setup procedure would help.

**Response:** We apologize for the confusions. The biogeochemical simulation over the whole Mediterranean Sea (step 1b) was performed from 15 June 2011 to 15 November 2013. We initialized the $CO_2$ system module using interpolated data as it was described L 178-183. The biogeochemical simulation over the western Mediterranean (step 2) was performed over the period from 15 August 2012 to 30 September 2013, and was initialized using the model outputs of the whole Mediterranean Sea simulation. To avoid confusions, as indicated in the response of one of the main comments, we will add the model run time of each step and will move the description of the initialization of step 1b before describing step 2.

**L. 193:** I find "DIC flows" and "inventory variations" rather confusing. Maybe "DIC fluxes" and "inventory tendencies"? Please check throughout the text.

**Response:** We will rephrase the sentence and change "flows" by "fluxes" throughout the text. We could replace "inventory" by "stock" or "content" if it is clearer.

*"We computed DIC  fluxes and the resulting content variation  for the whole deep convection area."*

**L. 194:** "for at least 1 day"

**Response:** The sentence will be changed as suggested in the revised manuscript.

**L. 201:** Given the title of this section, I wonder if Eq. 1 is better to be placed here. Additionally, I think at least the general budget equation (Eq. S1) should be moved to the main text.

**Response:** We would prefer to keep Eq. 1 in Section 2.1.1, since it gives the biogeochemical rate of change of the state variable DIC at the model grid points. As suggested, we will replace the text describing the budget *"The biological term of the budget […] upper layer is given in Supplementary Material (Text S1)"* in section "Study area and computation of DIC balance" by Text S1.

**L. 203**: What do you mean by "internal variation"? Please clarify.

**Response:** "internal variation" meant variation of the content of DIC during a considered period. It is given in Eq. S1 of the submitted version. This term will be removed here. We will also replace it throughout the text.

**L. 215:** Please add a reference to the respective Equation.

**Response:** We will add a reference to the Equation in the revised manuscript.

**L. 216:** "as 0.5"

**Response:** This sentence will be modified to take into account a comment of Reviewer 1.

*"Miquel et al. (2011) estimated the PIC:POC ratio varying between 0.31 and 0.78, with a mean value of~to~0.5 at 200 m depth based on sediment trap measurements at the EMSO-DYFAMED site."*

**L. 220:** Please be precise: NCP does not appear as such in Eq. 1.

**Response:** We agree, the sentence was confusing, We will move "in Eq. 1" as follows:

*" [...] we added in Eq. 1 a consumption term representing 35% of NCP ~in Eq. 1~."*

**L. 221:** Please state here what the parametrization by Lajaunie-Salla et al. (2021) is. Ideally, the reader should not have to look up other papers to understand what you're doing.

**Response:** In Lajaunie-Salla et al. (2021), carbonate precipitation, named *Precip*, is given by the following equation:

$$Precip = k_{precip} \frac{(\Omega_c - 1)}{0.4 + (\Omega_c - 1)} \sum_{i=1}^{3} \left( GPP_i - RespPhy_i \right)$$

where $k_{precip}$ is the PIC:POC ratio and $\Omega c$ the aragonite saturation, which we set at 3.5 based on Schneider et al. (2007).

We will add the equation in the text:
*"In a second sub-test, we added a CaCO3 production term based on the parametrization used in the Gulf of Lion's shelf modeling study by Lajaunie-Salla et al. (2021) (their Table A4, $Precip = k_{precip} \frac{(\Omega_c - 1)}{0.4 + (\Omega_c - 1)} \sum_{i=1}^{3} \left( GPP_i - RespPhy_i \right)$, where $k_{precip}$ is the PIC:POC ratio and $\Omega c$ the aragonite saturation, set at 3.5 based on Schneider et al. (2007))."*

**L. 225:** sea surface

**Response:** The sentence will be changed as suggested in the revised manuscript.

**L. 284**: I suggest adding "reflecting a" in front of "period"

**Response:** The sentence will be changed as suggested in the revised manuscript.

**L. 298:** Does the southern zone include everything south of 41°N or is there a southern limit?

**Response:** The southern zone includes all stations south of the convection zone. There is no southern limit.

**L. 299:** I assume the depth profiles have been subsampled to only include the cruise locations shown in Fig. 3. Please clarify.

**Response:** The modeled mean profiles shown in Fig. 5 correspond to the average of the modeled profiles extracted at the same location and date as the measurement stations. This will specify in the text:

*"Comparisons were performed by extracting model outputs at the same date and location as measurements."*

**L. 324:** Do you mean "alternating" instead of "alternative"?

**Response:** Yes, the sentence will be changed as suggested.

**L. 325:** Where can the direction of the wind be seen? If this is previous knowledge for the region of interest and you therefore decided not to show this explicitly, please make sure it is introduced in the introduction for clarity.

**Response:** We will specify the wind direction in the introduction, on Figure 1 and/or in Fig. S1 (of the submitted version of the manuscript) by adding two panels with maps of wind speed and direction.

**L. 337:** Unless I misread something, I think the minus sign should be omitted (the cumulative flux is positive according to Fig. 7).

**Response:** In fall, the cumulative air-sea flux is negative (Fig. 7b of the submitted version), we are sorry if it was not clear on Figure 7b. As recommended in one of the main comments, we will add a figure with schemes of the seasonal budget for which the direction of the flux will be clearer.

**L. 344:** Do you mean "DIC concentration in the ML" or "the DIC flux into the ML"? Please clarify.

**Response:** We meant a decrease in " DIC concentration" visible at the end of October and end of November in Figure 10b. We will slightly modify the sentence, as follows:

*"This led notably temporally to  low in DIC concentration into the mixed layer end of October and end of November (Fig. 10b)."*

**L. 441:** "episodes of heat gain"

**Response:** The sentence will be changed as suggested in the revised manuscript.

**L. 466:** To me, it is odd to call this flux biological production, when this is in fact remineralization/respiration. I understand why you do it and it is technically correct, but I still suggest rephrasing to avoid confusion.

**Response:** We agree that "production" can be confusing. We will replace this term by 'gain' here and a more appropriate term throughout the text and in figures:

*"Within the sea, biogeochemical processes induced an annual consumption of 3.7 mol C m$^{-2}$ yr$^{-1}$ of DIC in the upper layer and a  gain of 2.3 mol C m$^{-2}$ yr$^{-1}$ in the deeper layers."*

**L. 469:** For consistency with how you described the biological component, it would be easier to read if you also reflected the sign convention in your wording here.

**Response:** We will modify the sentence to reflect the sign convention as follows:

*"Our estimate of net physical fluxes (lateral plus vertical) is an input of 3.3 mol C m$^{-2}$ yr$^{-1}$ in the upper layer and an export of -11.0 mol C m$^{-2}$ yr$^{-1}$ in the deeper layer."*

**L. 474:** I suggest deleting "an amount".

**Response:** The sentence will be changed as suggested in the revised manuscript.

**L. 486:** Please see my comment on the abstract regarding "both dominate". I suggest to also rephrase here.

**Response:** We will also rephrase in the introduction of the discussion section, by merging the two last sentences:

*"Our results show that  biogeochemical and physical processes,  through their impacts on DIC concentration,  have both a major role in the intensity and sign of the air-sea exchanges in the deep convection area."*

**L. 489:** Here and throughout the discussion section: Can you find more descriptive/informative section titles? It is incredibly useful to the reader if the title of each section already conveys information, i.e., ideally the main take-away message.

**Response:** We will modify titles of the discussion section, as follows:

- *"5.1 The pCO2"* to *"5.1 Assessment of the seasonal cycle of the pCO2"*
- *"5.2 The air-sea CO$_2$ flux"* to *"5.2 Estimate of the annual air-sea CO$_2$ flux and its uncertainties"* and *"5.3 Comparisons on air-sea CO$_2$ flux in different Mediterranean regions"*
- *"5.3 Physical flows in the deep convection area"* to *"5.4 The major influence of physical transport in the deep convection area"*
- *"5.4 Net community production and air-sea fluxes"* to *"5.5 Net community production and air-sea fluxes relationships"*

**L. 490-502:** As far as I can see, these are results. I am not convinced this part is necessary.

**Response:** We will remove most of this part. Some elements will be kept to make easier the comparisons with previous studies.

**L. 508-509:** This sentence is unclear to me. Can you rephrase?

**Response:** We will rephrase this sentence as follows:

*"The high frequency measurements at the CARIOCA buoy described by Hood and Merlivat (2001) and Merlivat et al. (2018) indicated  an interannual variability of 4-5 weeks in the date  at which the pCO$_2$ difference changes sign,  depending on air-sea heat flux variations and the timing of the bloom onset."*

**L. 528-530:** Here and throughout the text: Try to avoid 1-2 sentence paragraphs.

**Response:** We will avoid this as much as possible throughout the text.

**L. 552:** Please see my major comment on these sensitivity experiments.

**Response:** As already answered to one of the main comments, we have corrected this error in the equation of the rate of change of alkalinity (excess negative charge denoted Σ[−]) and have again performed the sensitivity tests. In the new results, the air-sea flux could be reduced by 16% to 57% if carbonate production is taken into account. We will modify the text and Figure 14 in the discussion section on the sensitivity tests, in Sect. 2.1.4 "Sensitivity tests" and in the conclusion.

Discussion section:

*They show that not taken into account calcification processes could lead to an  overestimation of the annual air-sea $CO_2$ uptake by 16 to 57% with estimates 0.29 mol C $m^{-2}$ $yr^{-1}$, based on the mean PIC:POC ratio given by Miquel et al. (2011) (varying between 0.20 and 0.36 mol C $m^{-2}$ $yr^{-1}$ based on the maximum and minimum PIC:OC ratios, respectively), and 0.40 mol C $m^{-2}$ $yr^{-1}$, based on the parametrization used in Lajaunie-Salla et al. (2021)."*

**L. 566:** Is there a "yr-1" missing? Additionally, it would help to provide the range based on your model here again to compare to the cited paper more easily.

**Response:** Yes, we will correct the unit by adding a "yr-1" and add in the following sentence the range of the model estimates to make the comparison more easy:

*"The larger homogeneity in our estimates (varying between -0.1 and 1.2 mol C $m^{-2}$ $yr^{-1}$ inside the deep convection area) could be partly ascribed to the horizontal diffusion and advection that were accounted for in our model."*

**L. 576:** It might be more appropriate to say "physical transport".

**Response:** The sentence will be changed as suggested in the revised manuscript.

**L. 577:** "the vertical DIC distribution"

**Response:** The sentence will be changed as suggested in the revised manuscript.

**L. 581:** "greater magnitude" – Please specify the sign.

**Response:** We will specify the sign:

*"They both show a similar seasonal cycle with greater magnitude (positive for the vertical transport and negative for the lateral transport) in fall, the preconditioning phase [...]"*

**L. 582:** "sea heat loss" Do you mean "ocean heat loss"? Please clarify.

**Response:** We will replace "sea heat loss" by "sea surface heat loss".

**L. 589:** Please rephrase "DIC exchange flows".

**Response:** We will replace "DIC exchange flows" by "DIC fluxes at the limits of the deep convection area".

**L. 595:** "as illustrated in"

**Response:** The sentence will be changed as suggested in the revised manuscript.

**L. 608:** "slowed down" instead of "braked"

**Response:** The sentence will be changed as suggested in the revised manuscript.

**L. 617:** "convection" instead of "convention"?

**Response:** We will correct this error in the revised manuscript.

**L. 633:** "from" instead of "into"?

**Response:** The DIC budget shows a lateral DIC transport from the surrounding region into the deep convection region in the deep layer (Figure 12). We will change the sentence as follows:

*"More specifically, we found that the lateral exchanges with the surrounding region were characterized by a net lateral input of carbon into the deep layers of the deep convection region, [...]"*

**L. 634:** "a lateral outflow"

**Response:** The sentence will be changed in the revised manuscript.

**L. 640-646:** It would be a lot easier to compare to the findings of your studies, if you reported these numbers as flux densities instead of as integrated fluxes (or to here report your findings in the same integrated unit).

**Response:** Our estimate was reported L 645 in the same unit: 0.4 Tg C yr$^{-1}$. We will slightly change the sentence as follows:

*"Thus the NW Mediterranean deep convection area, which represents 2.5% of the Mediterranean Sea surface, and which, we estimate* here *absorbed* at the surface *0.4 Tg C yr$^{-1}$, could strongly contribute to the uptake of atmospheric $CO_2$ in the open Mediterranean Sea."*

**L. 648:** "into" instead of "in"

**Response:** The sentence will be changed as suggested in the revised manuscript.

**L. 666:** I suggest adding "…and rising atmospheric CO2 levels".

**Response:** We will add this in the sentence as suggested in the revised manuscript.

**L. 680:** budgets

**Response:** The sentence will be changed as suggested in the revised manuscript.

**L. 691:** What exactly are the first and second part here? Please clarify.

**Response:** The sentence will be changed as follows:

*"The region was marked by a deficit of $CO_2$ compared to the atmosphere from* November to early June *, which led to a 7-month ingassing of atmospheric $CO_2$"*

**L. 701:** "subject to"

**Response:** The sentence will be changed as suggested in the revised manuscript.

**Figures:**

**Fig. 3:** Please specify for what depth(s) the model output is shown here.

**Response:** In the caption, we indicated that the model outputs are "modeled at 3 m depth".

**Fig. 4:** I suggest adding a legend/title above each column.

**Response:** We will add "Observations' and "Model" above the first and second column, respectively.

[Figure]

*Figure 4: Surface dissolved inorganic carbon (DIC) concentration (µmol kg⁻¹) observed (left) and modeled (right) over the (a,b) DEWEX Leg1 (1-21 February 2013), (c,d) DEWEX Leg2 (5-24 April 2013), and (e,f)*

*MOOSE-GE (11 June-9 July 2013) cruise periods. The correlation coefficient (R), root mean square error (RMSE), and bias between surface observed and modeled DIC are indicated in (b,d,f).*

**Fig. 7:** I suggest using the same colors for the same components in all panels, not only in a & b, but also in panel c. Additionally, it is unclear to me why you decided to show the seasonal averages only for the upper layer and not for the deeper layer. Please consider adding the extra panel for completeness.

**Response:** The color for the different components in Fig. 7 was the same color as the same components shown in Fig. 6:
- biogeochemical fluxes in the upper layer in bright green,
- biogeochemical fluxes in the deeper layer in green/brown,
- physical fluxes in the upper layer in light blue,
- physical fluxes in the deeper layer in dark blue.

As recommended in one of the main comments, we will add a figure with seasonal budget schemes showing the budget in the upper and deeper layer, and to avoid repetitions we will remove Fig 7b.

**Fig. 14:** Please link the caption more clearly to the figure: which bar is which experiment? Only giving the reference requires the reader to be familiar with every single paper, which will not necessarily be the case (it certainly isn't the case for me).

**Response:** To clarify this figure, we will move the titles of the experiment in the top of the figure. We will also classify and color the bars according to the type of parametrization of the gas transfer velocity instead of the date of paper publication for the first set of experiments, and we will add the type of the parameterization in the caption.

[Figure]

*Figure 14: Sensitivity tests to the parameterization of gas transfer velocity, the variability of the mole fraction of $CO_2$ in the atmosphere, and the calcification processes on the annual $CO_2$ air-sea flux estimate. The black bar indicates the annual estimate in the reference simulation, grey bars the mean value for each of the three sets of sensitivity tests. For the sensitivity tests on the parametrization on gas transfer (from 2 to 9), relation with a quadratic (2), hybrid (3 to 5), cubic (6) wind speed dependency are respectively in light pink, yellow and orange, and relations that includes explicit bubbles parametrizations (7 to 9) are in dark pink. For the test (14) on calcification processes, the bar indicates the result found for the mean PIC:POC ratio, while the black line indicates the range using the minimum and maximum PIC:POC ratios.*

**All figures:** Please double-check that the sign convention of all fluxes is defined in the respective caption.

**Response:** We will check this.

Supplementary material: Eq. S2: "DCA" is not defined in the text.

**Response:** DCA was defined in L. 5 of the Supplementary material. We will move its definition just after the equation (that will be moved in the main text as recommended in a previous comment):

*"where (x,y,z) belongs to the upper layer (150 m to the surface) of the DCA (deep convection area)."*

**L. 29:** How are sediment fluxes treated in the model? How large are they compared to the other components? Without any further information, it is difficult to judge for the reader to

what extent this assumption impacts the role of vertical fluxes (which are treated as the residual and will therefore include any sedimentary contribution).

**Response:** The fluxes of dissolved inorganic carbon, nutrients and oxygen at the sea-sediment interface were calculated using a simplified version of the vertically-integrated dynamic sediment model described in Soetaert et al. (2000). The parameters of the model were set following the study of Pastor et al. (2011) in the Gulf of Lion shelf. The same model was used by Many et al. (2011) who showed that the model results were consistent with previous observational and modeling studies on the Gulf of Lion shelf. In this study, we found a POC deposit of 0.1 mol m$^{-2}$ yr$^{-1}$ in the deep convection area. This is in the same order, but smaller than the sediment flux estimated at 0.2 mol C m$^{-2}$ yr$^{-1}$ by Stabholz et al. (2013) near the bottom in the deep convection area. The authors reported an increase of the flux by one to two orders of magnitude during a winter characterized by deep convection. They attributed this increase to resuspension events induced by strong bottom currents. Durrieu de Madron et al. (2023) also pointed out the influence of dense shelf water cascading which can be responsible for supplementary organic carbon deposit flux. In the model, the efflux of DIC resulting from the sediment organic carbon remineralization is calculated during the simulation and taken into account in the budget but is negligible compared to all the other terms. Further comparison analyses will be needed in the future to verify the model in the deep region. Moreover, a coupling with sediment transport model would allow improving the description of the deposition flux of organic carbon and the modifications in the sediment resulting from resuspension events. In the revised manuscript, we will specify how the fluxes are calculated at the sea-sediment interface and indicate that we found a negligible annual DIC flux.

Durrieu de Madron X., D. Aubert, B. Charrière, S. Kunesch, C. Menniti, O. Radakovitch, and J. Sola. 2023. Impact of dense water formation on the transfer of particles and trace metals from the coast to the deep in the northwestern Mediterranean. Water, 15, 2: 301. doi: 10.3390/w15020301.

Stabholz, M., Durrieu de Madron, X., Canals, M., Khripounoff, A., Taupier-Letage, I., Testor, P., Heussner, S., Kerhervé, P., Delsaut, N., Houpert, L., Lastras, G., and Dennielou, B.: Impact of open-ocean convection on particle fluxes and sediment dynamics in the deep margin of the Gulf of Lions, Biogeosciences, 10, 1097–1116, https://doi.org/10.5194/bg-10-1097-2013, 2013.

---

## Author Response (AR1)

**Seasonal dynamics and annual budget of dissolved inorganic carbon in the northwestern Mediterranean deep convection region**

Caroline Ulses, Claude Estournel, Patrick Marsaleix, Karline Soetaert, Marine Fourrier, Laurent Coppola, Dominique Lefèvre, Franck Touratier, Catherine Goyet, Véronique Guglielmi, Fayçal Kessouri, Pierre Testor, Xavier Durrieu de Madron

**Responses to the Reviewers' comments**

Answers to reviewers' comments are reported point by point. Reviews are included in black font, answers in blue font and the modifications done in the revised manuscript in italic red font. We indicated the line number where the modifications have been done in the manuscript with track changes.

**Responses to the comments of the anonymous Reviewer 1**

First we would like to warmly thank Reviewer 1 for his/her relevant and constructive comments which helped to improve the manuscript.

—

The authors investigated the dynamics of dissolved inorganic carbon in the deep convection area of the North-West Mediterranean Sea. The study was based on a good coupling between observations from mooring sites and cruises and 3 D coupled physical-biogeochemical model.
The main findings were that the area:
-was a moderate sink of $CO_2$ (0.47 mol C $m^{-2}$ $yr^{-1}$) with an increase during the spring phytoplankton bloom, the air sea flux represented only 12% of net community production in the upper lever of the water column;
- both biological processes and physical transport (vertical and horizontal) played a dominant role in the annual DIC budge;
- winter ventilation had a reducing effect of on the atmospheric $CO_2$ uptake;
- the region acted as a source of DIC for surface and intermediate waters.
Overall the approach is innovative and the results are relevant for a better understanding of the CO2 system dynamics in the NW Mediterranean Sea.

**Response:** We thank Reviewer 1 for this positive general assessment.

I think that the discussion could be improved by a deeper comparison with one of the few other areas of the Mediterranean Sea where deep convection occurs as the Southern Adriatic Sea. The Adriatic Dense Water formation plays an important role for the sequestration and storage of the anthropogenic carbon, as the anthropogenic CO2 is transferred in the deep waters of the Eastern Mediterranean (Krasakopoulou et al., Deep Sea Res., 2011; Cantoni et al. Mar. Geol. 2016; Ingrosso et al. Deep Sea Res., 2017).

**Response:** We thank Reviewer 1 for this advice and the interesting references. We have added in the discussion a sub-section dedicated to the comparison of our results in terms of air-sea $CO_2$ flux (also in response to a comment of Reviewer 2). In this sub-section, we have included comparisons with other studies carried out in the northwestern Mediterranean which was in Section 5.2 in the previously submitted manuscript, as well as a comparison with studies in the other major deep convection region of the Mediterranean, the South Adriatic in L. 687-689.

L. 687: *"Our estimate is close to the annual flux estimated around 0.5 mol C $m^{-2}$ $yr^{-1}$ by Cossarini et al. (2021) in the South Adriatic Sea, another deep convection area of the Mediterranean Sea."*

Furthermore, in section "Contribution of northwestern deep convection region to the carbon budget of the Mediterranean Sea", we have added, L. 787-800, a discussion on the exchanges between the two deep convection areas and the surrounding regions, as follows:

L. 787: *"Our results for the northwestern deep convection area could be compared to those obtained in one of the other major deep water formation areas of the Mediterranean Sea, the Adriatic Sea. This latter has been shown to be a sink of atmospheric $CO_2$ (Cossarini et al., 2021) and a sequestration region of anthropogenic carbon (Krasakopoulou et al., 2011; Palmiéri et al., 2015; Hassoun et al., 2015; Ingrosso et al. 2017) as the study area (Touratier et al., 2016). In particular, experimental studies showed that the deep layer of the South Adriatic Sea was occupied by dense water rich in DIC and anthropogenic carbon formed in the deep convection regions of South Adriatic Pit and Pomo Pit, as well as on the northern shelf (Krasakopoulou et al., 2011; Cantoni et al, 2016; Ingrosso et al. 2017). The deep dense waters could be then transferred towards the Ionian Sea and the Mediterranean general deep circulation. Krasakopoulou et al. (2011) deduced from in situ measurements over February 1995 inorganic carbon fluxes crossing the Otranto Strait which connects the Ionian Sea to the South Adriatic Sea. They estimated that, on an annual basis, the Adriatic Sea could act as a sink of 314 Tg C $yr^{-1}$ of dissolved inorganic carbon for the Ionian Sea. This net flux*

*resulted from an inflow of 1563 Tg C yr$^{-1}$, with 27% in the Levantine Intermediate Water, and an outflow of 1249 Tg C yr$^{-1}$, with 21% in the Adriatic Deep Water. Thus, the northwestern Mediterranean deep convection region and the South Adriatic that includes shallower areas, could have opposite contributions in the deep and intermediate layers of the Mediterranean general circulation. However, our DIC budget assessment is limited to a single year and will need to be extended to a longer period to investigate in particular the question of carbon sequestration."*

Hassoun, A.E.R., Gemayel, A., Krasakopoulou, E., Goyet, E., Saab, C., Guglielmi, M.A.-A., Touratier, V., Falco, C, F., 2015. Acidification of the Mediterranean Sea from anthropogenic carbon penetration. Deep-Sea Res. I 102, 1–15. http://dx.doi.org/10.1016/j.dsr.2015.04.005.

Palmiéri, J., Orr, J.C., Dutay, J.C., Béranger, K., Schneider, A., Beuvier, J., Somot, S., 2015. Simulated anthropogenic CO2 uptake and acidification of the Mediterranean Sea. Biogeosciences 12, 781–802. http://dx.doi.org/10.5194/bg-12-781-2015.

In the Chapter 5.5 "Contribution of north-western deep convection region to the carbon budget of the Mediterranean Sea" the discussion could be improved by taking into account not only the modelling studies but also the experimental studies showing that the Adriatic continental platform acts as a sink for atmospheric $CO_2$ (e. g.:  Turk et al., Jour. Geophys. Res., 2010; Cantoni et al., Est. Coast Shelf Sci.,2012; Catalano et al., Jour. Geophys. Res., 2014; Urbini et. al., Front. Mar. Sci., 2020).

**Response:** We thank Reviewer 1 for the suggestion and these pertinent references. We have completed the discussion on comparisons of the modeled air-sea $CO_2$ fluxes in the new sub-section 5.3, by expanding it to comparisons with the northern continental shelves which were identified as other water formation areas in the Mediterranean Sea, in L 689-700:

L. 689: *"Finally, it is also noteworthy that our estimate is found in the lower range of the annual flux estimated from experimental studies for the northern Adriatic and Aegean shelves, where dense water formation also takes place, and identified as sinks for atmospheric $CO_2$ most of the year and on an annual basis. With respect to the northern Adriatic shelf, our estimate is found close to the estimate of 0.4-0.5 mol C m$^{-2}$ yr$^{-1}$ for year 2014/15 by Urbini et al. (2020) and between about 2 to 4 folds lower than the estimates of 0.8-0.9 mol C m$^{-2}$ yr$^{-1}$ by Urbini et al. (2020) over the year 2016/17, of 1-1.1 mol C m$^{-2}$ yr$^{-1}$ by Catalano et al. (2014) and Cossarini et al. (2015) and of 2.2 mol C m$^{-2}$ yr$^{-1}$ by Cantoni et al. (2012) and Turk et al. (2010). Regarding the northern Aegean Sea, we found a lower winter flux than the one deduced from observations in February 2006 by Krasakopoulou et al. (2009) (8.6-14.7 mmol C m$^{-2}$ day$^{-1}$ versus 4.9 mmol C m$^{-2}$ day$^{-1}$ in our study). Our estimates*

*are also lower than the CO$_2$ uptake exceeding 1 mol C m$^{-2}$ yr$^{-1}$ found for the northern shelves in the modeling studies of Cossarini et al. (2015; 2021). The higher fluxes over the continental shelves compared to our study area could be explained by a lower seawater temperature in winter, riverine nutrient inputs favoring intense primary production, and a transport of DIC associated with dense water outflow towards the deep basin (Cantoni et al., 2016; Ingrosso et al., 2017)."*

Cossarini, G., Querin, S., Solidoro, C.: The continental shelf carbon pump in the northern Adriatic Sea (Mediterranean Sea): influence of wintertime variability. Ecol. Model. 314, 118–134. http://dx.doi.org/10.1016/j.ecolmodel.2015.07.024, 2015.

Besides, in Section 4.2 "Annual carbon budget", we mentioned the higher air-sea CO$_2$ flux found in our model results on the shelf of the Gulf of Lion, another Mediterranean region where dense shelf water formation and cascading take place. Based on the model configuration implemented by Many et al. (2021), we plan to investigate the seasonal and interannual carbonate system dynamics on this shelf. We think that, in this future work, it would be very interesting to compare the seasonal and annual budget terms, as well as influences of northern winds and river inputs obtained for the Gulf of Lion shelf, with the observational previous works carried out on the northern Adriatic shelf both presenting many similar characteristics (as winter low temperature, continental winds, physical processes), but with a more enclosed morphology and higher river inputs for the northern Adriatic.

In the sensitivity tests including the carbonate production the authors used a PIC/POC ratio of 0.5 but according to the results reported in the cited paper of Miquel et al. (2011) the ratio is subject to wide interannual variations ranging from 0.31 to 0.78. It would be important to know how these natural variations would affect the sensitivity tests.

**Response:** Following the comment of Reviewer 1, we have performed sensitivity tests on carbonate production using the minimum and maximum values of the PIC:POC ratio reported by Miquel et al. (2011), to assess the impact of the natural variations of this ratio on the air-sea flux. The difference between air-sea fluxes computed for these two tests is equal to 0.07 mol C m$^{-2}$ yr$^{-1}$. We have added the results of these tests in the discussion section on air-sea CO$_2$ flux, Figure 14, Table S1, Sect. 2.1.4 "Sensitivity tests" and in the conclusion. Moreover, we specify that a correction was made in the calculation of the rate of change of alkalinity (the excess negative charge state variable, see the answer to a following comment) that explains the difference in air-sea flux using the mean value of

PIC:POC ratio given in Miquel et al. (2011), between the new version of the manuscript and the previous one.

Section 2.1.4 "Sensitivity tests", L. 284-290:

"Following the study of Palevsky and Quay (2017), we first estimated it based on PIC:POC ratio and NCP. Miquel et al. (2011) estimated *that* the PIC:POC *ratio at 200 m depth varied between 0.31 and 0.78, with a mean value of* 0.5, based on sediment trap measurements at the EMSO-DYFAMED site. [...] Thus, *by* assum*ing* the ratio of calcium carbonate production to NCP is close to *the* PIC:TOC *ratio* we added *in Eq. 1* a consumption term representing 3*6*% of NCP for the mean value of PIC:POC ratio*, and 22% and 55% for the minimum and maximum ratio values, respectively.*``

Discussion section, new Section 5.2 "Estimate of the annual air-sea $CO_2$ flux and its uncertainties, L. 661-665:

"They show that *not taken into account* calcification processes could lead to an *overestimation* of the annual air-sea $CO_2$ uptake by *16* to *57*% with estimates *of 0.29* mol C $m^{-2}$ $yr^{-1}$, based on *the mean* PIC:POC ratio *and NCP (varying between 0.20 and 0.36 mol C m$^{-2}$ yr$^{-1}$ based on the measured maximum and minimum PIC:POC ratios, respectively), and 0.40* mol C $m^{-2}$ $yr^{-1}$, based on the parametrization used in Lajaunie-Salla et al. (2021)."

Conclusion, in L. 847-849:

"Moreover, we displayed that *neglecting* calcification processes could lead to an overestimation by 216 to 557% of the annual uptake, highlighting the need for the refinement of the model in future studies."

In the conclusion the authors state that the air-sea flux represents only 13% of the upper column Net Community Production (NCP) whereas in the chapter 5.4 that state that the flux represent 12% of NCP. The discrepancy should be solved.

**Response:** The correct value of the air-sea flux / net community production ratio is 13% (=0.47/3.74=12.57%). We apologize for the error. The value has been corrected in Section 5.5 (5.4 in the previously submitted manuscript), L. 735.

In the conclusion the authors states that the physical fluxes in the upper layer is of 3.3 mol C m-2 yr-1 but in the figure 12 the difference between the lateral DIC transport and the vertical DIC transfer amounts to 3.4 mol C m-2 yr-1. The data should be checked.

**Response:** The values of the physical fluxes in the upper layer have been checked. The correct value of the net physical flux is 3.34 mol C m$^{-2}$ yr$^{-1}$. It results from the sum of a vertical input of 133.18 mol C m$^{-2}$ yr$^{-1}$ and of a lateral export of 129.84 mol C m$^{-2}$ yr$^{-1}$. Thus, the values in Figure 12 and in the text were correct in the previous version of the manuscript.

The authors in the conclusion more clearly the in the discussion (L.547-552) state that calcification processes could lead to an underestimation by 23-58% of the annual uptake but the authors should take into account that the calcification processes although reducing the TCO2 will increase the pCO$_2$ in seawater therefore counteracting the CO$_2$ intake from the atmosphere.

**Response:** We thank Reviewer 1 for raising this point. We acknowledge that there was an error in the sensitivity test on calcification process, by omitting to take into account the process in the rate of change of alkalinity (excess negative charge denoted $\sum[-]$). We apologize for this error. We have corrected it by adding in the equation of the rate of change of alkalinity (excess negative charge denoted $\sum[-]$) the term of calcium carbonate production added in the DIC equation multiplied by 2 (Middelburg et al., 2019). In the new results, the air-sea CO$_2$ flux is reduced by 16% to 57% when the impact of calcification processes is modeled. We have modified the text and Figure 14 in the discussion section on the sensitivity tests on air-sea CO$_2$ flux, in Sect. 2.1.4 "Sensitivity tests", and in the conclusion.

Middelburg, J. J.: Marine Carbon Biogeochemistry A Primer for Earth System Scientists, Springer B., edited by Springer Briefs in Earth System Sciences, Springer Briefs in Earth System Sciences, 2019.

Section 2.1.4 "Sensitivity tests", in L. 288:
*"Thus, by assuming the ratio of calcium carbonate production to NCP is close to the PIC:TOC ratio, we added in Eq. 1 a consumption term representing 36% of NCP for the mean value of PIC:POC ratio, and 22% and 55% for the minimum and maximum ratio values, respectively. This term, multiplied by 2, was added in the equation of the rate of change of the excess negative charge (Middelburg, 2019)."*

Discussion section, Section 5.2 "Estimate of the annual air sea CO$_2$ flux et and its uncertainties, in L. 659-665:
*"Finally, sensitivity tests taking into account supplementary consumption terms in the equation of DIC and excess of negative charge for CaCO$_3$ precipitation (Sect. 2.1.4) were performed to assess its potential influence on air-sea CO$_2$ flux. They show that not taken into*

*account calcification processes could lead to an  overestimation of the annual air-sea CO₂ uptake by 16 to 57% with estimates of 0.29 mol C m⁻² yr⁻¹, based on the mean PIC:POC ratio and NCP (varying between 0.20 and 0.36 mol C m⁻² yr⁻¹ based on the maximum and minimum PIC:POC ratios, respectively), and of 0.40 mol C m⁻² yr⁻¹, based on the parametrization used in Lajaunie-Salla et al. (2021)."*

[Figure]

**Figure 14: Sensitivity tests to the parameterization of gas transfer velocity, the variability of the mole fraction of CO₂ in the atmosphere, and the calcification processes, on the annual air-sea CO₂ flux estimate. The black bar indicates the annual estimate in the reference simulation, grey bars the mean value for each of the three sets of sensitivity tests. For the sensitivity tests on the parametrization of gas transfer (from 2 to 9), relations with a quadratic (2), hybrid (3 to 5), cubic (6) wind speed dependency are, respectively, in light pink, yellow and orange, and relations that include explicit bubble parametrizations (7 to 9) are in dark pink. For the test (14) on calcification processes, the bar indicates the result found for the mean PIC:POC ratio, while the black line indicates the range using the minimum and maximum PIC:POC ratios.**

Conclusion, in L. 847-849:

*"Moreover, we displayed that neglecting calcification processes could lead to an overestimation by 16 to 57% of the annual uptake, highlighting the need for the refinement of the model in future studies."*

The authors use the terms "biogeochemical flow "and "physical flow" which are not very appropriate terms as both are related to a mass flow of carbon generated by biological processes or by physical processes (advection, mixing, particle settling). I suggest to find a more appropriate alternative term e.g.:"physical transport".

**Response:** We acknowledge that the term "flow" was often inappropriate and apologize for this. We have replaced this by "flux", "transport", "export" and "input" when it was not appropriate.

**Specific comments**

**L. 49-50**, "is one of the region where deep convection occurs" a specific reference to the Southern Adriatic SAD should be added.

**Response:** The first sentences of the paragraph have been modified in order to add a specific reference to the Southern Adriatic, L. 53-56, as follows:

*"The northwestern  Mediterranean Sea (Gulf of Lion and Ligurian Sea, Fig. 1), alongside with the South Adriatic,  is one of the regions where deep convection occurs (Ovchinnikov et al., 1985; Mertens and Schott, 1998; Manca and Bregant, 1998; Gačić et al., 2000; Béthoux et al., 2002)."*

Gačić, M., Manca, B.B., Mosetti, R., Scarazzato, P., Viezzoli, D., 2000. Deep water formation experiment in the Adriatic Sea. WWW Page, http://doga.ogs.trieste.it/doga/jwz/deep_water/mtpnews1.html

Manca, B. and D. Bregant: Dense water formation in the Southern Adriatic Sea during winter 1996. Rapp. Comm. Int. Mer Médit., 35, 176-177, 1998.

Ovchinnikov, I.M., Zats, V.I., Krivosheya V.G., Udodov A.I.: Formation of deep eastern Mediterranean water in the Adriatic Sea Oceanology, 25 (6) (1985), pp. 704-707, 1985.

**L. 370-37**, "upward flux of DIC into the upper layer of 41.40 mol C m-2…" The units of a mass flux should be used. They should be expressed as the mass of carbon that passes through a defined cross-sectional area over a period of time.

**Response:** We thank Reviewer 1 for raising this point. In Section 4.1, we give either the daily flux in mmol C $m^{-2}$ $day^{-1}$, or the cumulative flux, i.e. the flux, expressed as an amount of matter per surface per unit of time, multiplied by the considered period of time: mol C $m^{-2}$. In the revised version of the manuscript, we have added "cumulative" before "flux" L. 459-463, and in the whole "4.1 Seasonal cycle of dissolved inorganic carbon" section when there was an oversight, and we have indicated the period over which the cumulative flux is calculated.

*"The physical fluxes at the limit of the upper layer of the deep convection area showed similar patterns as during autumn, with a cumulative upward flux of DIC into the upper*

*layer of 41.40 mol C m$^{-2}$ over a 2.5 month period, almost counterbalanced by a cumulative lateral outflow of DIC of 40.44 mol C m$^{-2}$ in the upper layer and a cumulative lateral inflow of DIC of 39.90 mol C m$^{-2}$ in the deeper layer."*

**L. 390- 395, L. 469-470; L. 529**. same as above.

**Response:** In Section 4.1, the term "cumulative" was mentioned in L. 390 and 394 of the previously submitted version, we have added the period over which the time-integration of flux is done:

*"The cumulative biogeochemical flux reached -1.49 mol C m$^{-2}$ over this sub-period of 67 days."*

*" […] and finally cumulative air-sea flux reached 0.28 mol C m$^{-2}$ over the second winter sub-period of 67 days (a lower value and flux (3.1 versus 7.3 mmol C m$^{-2}$ day$^{-1}$) than over the first winter period)"*

L. 469-470 and L. 529 of the previously submitted version and in this whole Section 4.2, the annual fluxes are given and therefore we have corrected the unit of the fluxes by replacing "mol C m$^{-2}$" by "mol C m$^{-2}$ yr$^{-1}$", L. 571-593:

*"We estimate that it absorbed 0.5 mol C m$^{-2}$ yr$^{-1}$ of atmospheric CO$_2$. This uptake of atmospheric CO$_2$ displayed spatial variability (Fig. 12). It was greater than 1 mol C m$^{-2}$ yr$^{-1}$ in the northern edge of the area along the Northern Current flowing over the Gulf of Lion continental slope, and became less than 0.25 mol C m$^{-2}$ yr$^{-1}$ in the western and eastern edge areas. One can notice that the annual rate remained lower than on the Gulf of Lion's shelf, which is beyond the scope of this study. Within the sea, biogeochemical processes induced an annual DIC consumption of 3.7 mol C m$^{-2}$ yr$^{-1}$ of DIC in the upper layer and a production DIC gain of 2.3 mol C m$^{-2}$ yr$^{-1}$ in the deeper layers.*
*Our estimate of net physical fluxes (lateral plus vertical) is an input of 3.3 mol C m$^{-2}$ yr$^{-1}$ in the upper layer and an export of -11.0 mol C m$^{-2}$ yr$^{-1}$ in the deeper layer. Specifically, the model indicates a vertical DIC supply of 133.2 mol C m$^{-2}$ yr$^{-1}$ from the deeper layer to the upper layer, partly offset by a lateral outflow of 129.8 mol C m$^{-2}$ yr$^{-1}$ in the upper layer and an inflow of 122.2 mol C m$^{-2}$ yr$^{-1}$ in the deeper layer. The budget in the deep layer masks different signs of physical fluxes: if the deeper layer is subdivided into an intermediate layer (150 m-800 m) and the deeper most layer (800 m-bottom), we find that the former, the intermediate layer, gained an amount of 83.1 mol C m$^{-2}$ yr$^{-1}$ of DIC through vertical transport, while it lost 87.6 mol C m$^{-2}$ yr$^{-1}$ of DIC through lateral export. Finally, our model shows that the convection zone was a source of DIC of 8.7 mol C m$^{-2}$ yr$^{-1}$ for the rest of the western Mediterranean Sea. While the DIC inventory in the upper layer remained stable (decrease of 0.07 mol C m$^{-2}$ yr$^{-1}$), the DIC inventory in the deeper layer experienced a*

*decrease of 8.7 mol C m$^{-2}$ yr$^{-1}$. This loss occurred mainly during deep convection, and to a lesser extent during the preconditioning period (in autumn and early winter).*
*Finally, we complete the inorganic carbon budget with the labile organic carbon fluxes (refractory organic carbon is not considered in our model). We estimate that during the studied period a lateral export of organic carbon of 1.1 mol C m$^{-2}$ yr$^{-1}$ and 0.3 mol C m$^{-2}$ yr$^{-1}$ took place in the upper and deeper layers, respectively. The modeled downward export of organic carbon amounted to 2.3 mol C m$^{-2}$ yr$^{-1}$."*

**L. 452**, "the DIC drawdown due to biological processes decreases and net DIC production events took place": could the authors specify which are the processes driving the DIC production events.

**Response:** Since we haven't deeply analyzed specifically these short events, we have removed this and have rephrased this sentence in the revised manuscript, L. 553-554, as follows:

*"From August onwards, the DIC drawdown due to biogeochemical  processes decreased, the primary production rate becoming close to the respiration rate (Fig. 6h)."*

**L. 465** "an annual consumption of 3.7 mol C m$^{-2}$ of DIC": the unit of time is lacking.

**Response:** We agree, we apologize for this oversight. As mentioned in a previous response, the sentence has been modified as follows, in L. 575-576:

*"Within the sea, biogeochemical processes induced an annual DIC consumption of 3.7 mol C m$^{-2}$ yr$^{-1}$  in the upper layer and  DIC gain of 2.3 mol C m$^{-2}$ yr$^{-1}$ in the deeper layers. "*

**L.549-552**. This sentence is not clear and the CO$_2$ production during calcification should be taken into account.

**Response:** As mentioned in a previous response, we have corrected the sensitivity tests on calcification processes. The sentence has been modified as follows, in L. 661-665:

*"They show that not taken into account calcification processes could lead to an  overestimation of the annual air-sea CO$_2$ uptake by 16 to 57% with estimates of 0.29 mol C m$^{-2}$ yr$^{-1}$, based on the mean PIC:POC ratio and NCP (varying between 0.20 and 0.36 mol C m$^{-2}$ yr$^{-1}$ based on the maximum and minimum PIC:POC ratios,*

*respectively), and of 0.40 mol C m$^{-2}$ yr$^{-1}$, based on the parametrization used in Lajaunie-Salla et al. (2021)."*

**L. 578** physical flow? Do the authors mean physical transport?

**Response:** "physical flow" has been replaced by "physical transport" in L. 704.

*"[...] and highlights that physical transports play a crucial role in the DIC budget in this highly energetic region."*

**L. 589** DIC exchange flows? Do you mean DIC flows?

**Response:** "DIC exchange flow" has been replaced by "DIC fluxes at the limits of the zone", in L. 715.

*"Moreover, a detailed calculation of the water and DIC  fluxes at the limits of the deep convection area allowed us to [...]"*

**Fig. 7.** The units for fluxes are expressed as an inventory: mol C m-2. The mass fluxes should be expressed as the mass of carbon that passes through a defined cross-sectional area over a period of time e.g. mol C m$^{-2}$ y$^{-1}$.

**Response:** Figure 7 shows the cumulative fluxes, i.e. the fluxes expressed as the amount of matter per surface and per unit of time, multiplied by the time period over which the accumulation is calculated. In response to a comment of Reviewer 2 we have removed panel (b) with the cumulative seasonal fluxes. In the remaining panels, the cumulative flux at a day d is the flux, expressed in mol C m$^{-2}$ day$^{-1}$ multiplied by the number of days between the 1$^{st}$ September 2012 and day d. We have corrected the titles of the two panels by adding "cumulative" before "fluxes" and have completed the caption.

[Figure]

**Figure 7:** *Time series of variation in dissolved inorganic carbon (DIC) inventory since the 1st September 2012 (black) and cumulative air–sea (red), physical (light and dark blue), and biogeochemical (bright and brown green) flux of dissolved inorganic carbon in the (a) upper (surface to 150 m) and (b) deeper (150 m to bottom) layers, from September 2012 to September 2013. The cumulative flux at a day d is the time-integrated flux over the period from the 1st September 2012 to day d.* Unit: mol C m$^{-2}$. Positive values of fluxes represent DIC inputs for the deep convection area. The blue shaded area corresponds to the deep convection period (period when spatially averaged mixed layer depth > 100 m). The DIC inventory on 1st September 2012 was 353 and 5560 mol C m$^{-2}$ in the upper and deeper layers, respectively.

**Fig. 12.** The data represented are inventories or fluxes in the latter case they should be expressed as the mass of carbon that passes through a defined cross-sectional area over a period of time e.g. mol C m$^{-2}$ y$^{-1}$.

**Response:** We apologize for this error. The unit on Figure 12 (Figure 11 in the new version of the manuscript) and in its caption has been corrected as follows:

[Figure]

*"**Figure 11**: Scheme of the annual carbon budget for the period September 2012 to September 2013 from the coupled model SYMPHONIE-Eco3M-S. Fluxes are indicated in mol C m$^{-2}$ yr$^{-1}$. The direction of the arrows indicates the direction of the fluxes and positive values of fluxes represent DIC inputs for the deep convection area (positive vertical fluxes represent inputs for the upper layer)."*

**Responses to the comments of the anonymous Reviewer 2**

First, we would like to warmly thank Reviewer 2 for his/her relevant and constructive comments which will help to improve the manuscript.

Review of Ulses et al. (2022): "Seasonal dynamics and annual budget of dissolved inorganic carbon in the northwestern Mediterranean deep convection region"

**Summary**

This study by Ulses and coauthors presents a detailed carbon budget in the deep convection area of the NW Mediterranean Sea for the period between September 2012 and September 2013. Using an ocean biogeochemistry model forced with daily output from a physical model, the authors find that their focus region is a moderate sink of atmospheric $CO_2$ over the study period. In addition, by dividing the study area into the upper 150m and the deep ocean below that, they find that both physical and biological fluxes play an important role in controlling carbon fluxes across seasons.

Overall, the authors did a great job comparing their model results to existing observations and presenting the carbon budget of their study region in great detail. Therefore, the study is generally suitable for publication in Biogeosciences. However, I would like to raise several points, mostly regarding the presentation of the study, which should be addressed before the publication of this manuscript.

Please see the detailed explanation of these major points and all my detailed comments below.

We appreciate this positive general assessment.

**Main comments**

1. Introduction: Acknowledging that I am not 100% familiar with the literature concerning the Mediterranean Sea, the introduction appears to give a good summary of previous work. However, it fails to make the knowledge gap clear enough in my view. In its current form, it still reads too much as a collection of results from individual studies, making it hard for the reader to figure out what has not been addressed or what the short-comings in each of these previous studies are. As a result, I am a bit lost guessing what exactly the focus of the study by Ulses et al. is until L. 108. I suggest revising the introduction to more clearly state where the knowledge gaps are that are to be addressed in this new study.

Response: As suggested by Reviewer 2, we have revised the introduction to more clearly point the gaps in the previous studies on DIC cycle and give earlier in the text the objective of the present study, as follows, in L. 53-112:

[revised manuscript text omitted]

2. Description of the model setup: While being methodologically sound from what I understood, the description of the model setup in section 2.1.2 is currently hard to follow. I suggest including a sketch in the revised version of the manuscript illustrating the downscaling approach and providing information on the initialization and run time of the simulations in each of the steps of the setup.

**Response:** We have clarified this description by adding a figure in Supplementary Material to show (1) the domain of the two coupled physical-biogeochemical models, i.e. the parent and child models, and (2) a scheme of the downscaling strategy.

[Figure]

*Figure S1: (a) Domain and bathymetry (m) of the forcing coupled NEMO-Eco3MS Mediterranean model (red contour) and of the coupled SYMPHONIE-Eco3MS western basin model (blue contour). (b) Scheme of the downscaling strategy from the Mediterranean Sea to the western basin.*

We have also added in Section 2.1.3 "Model setup" of the revised manuscript the run time of each of the three simulations and have moved the description of the initialization in step 1b before describing step 2. We hope the description of the downscaling strategy will be clearer after these modifications and additional elements, L. 189-213:

*"The implementation of the hydrodynamic simulation and the strategy of downscaling from the Mediterranean Basin to the western sub-basin scale in three stages (Fig. S1) have been described in detail in Estournel et al. (2016) and Kessouri et al (2017) and will be summarized here:*
*- In a first step (step 1a, Fig. S1), the SYMPHONIE hydrodynamic model, implemented over the Western Mediterranean sub-basin (delimited by blue lines in the insert of Fig. 1), was initialized and forced at its lateral boundaries with daily hydrodynamic  fields of the configuration PSY2V4R4, based on the NEMO ocean model at a resolution of 1/12° over the Mediterranean Basin (delimited by orange lines in the insert of Fig. 1) by the Mercator Ocean International operational system (Lellouche et al., 2013). This simulation was performed from 1ˢᵗ August 2012 to 31 October 2013.*
*- In parallel (step 1b, Fig. S1), the biogeochemical model was computed, in offline mode, at the Mediterranean basin scale, on the same 1/12° NEMO grid, using the same NEMO hydrodynamic fields as those used by the SYMPHONIE simulation in step 1a. This simulation*

*was performed from 15 June 2011 to 15 November 2013. The carbonate system module in this configuration was initialized using mean values of dissolved inorganic carbon, total alkalinity observations carried out in 2011 from the Meteor M84/3 (Alvarez et al., 2014), CASCADE (CAscading, Surge, Convection, Advection and Downwelling Events, Touratier et al., 2016), and MOOSE-GE cruises (Testor et al., 2010) ,as well as from the EMSO-DYFAMED mooring (Coppola et al., 2021) and BOUSSOLE buoy (Golbol et al., 2020) sites, over bio-regions defined in Kessouri (2015), based on Lavezza et al. (2011). We deduced the concentration of the excess negative charge based on nutrient concentrations initialized using the Medar/Medatlas database as in Kessouri et al. (2017). Recently, Davis and Goyet (2021) described a method based upon the property variability, to precisely quantify the uncertainties at any point of an interpolated data field. This approach could be used in the near-future to improve both the at-sea sampling strategy (Guglielmi et al., 2022a; 2022b), and the accuracy of model initialization.*

*- In a second time (step 2, Fig. S1), the Eco3M-S biogeochemical model was implemented over the western Mediterranean sub-basin, using the grid and the hydrodynamics fields of the aforementioned SYMPHONIE simulation (step 1a) in offline mode. This simulation was performed from 15 August 2012 to 30 September 2013. The initial state and lateral boundary conditions of the biogeochemical fields are provided by the biogeochemical simulation of the Mediterranean Sea of step 1b."*

3. Result section 4.1: I admittedly found it quite difficult to keep up with all the provided details in this section. In general, I appreciate the detailed description of the figures, and I generally think the clear division into the different seasons is good. However, this division means that the reader must constantly jump back and forth between Fig. 6-11, making it very important to have consistent structure and summarizing sentences throughout this section. While such summarizing sentences already exist for some of the seasons (see e.g., winter sub-period 2), they do not for others. I thus suggest that the authors carefully screen the result section again to structure the description of each season as consistently as possible and that they add clear summarizing sentences to each of the seasons. I encourage the authors to work on the paragraph structure (including topic sentences), as this will greatly improve the readability of this part of the manuscript. Lastly, since I really appreciated Fig. 12 as a summary for the annual mean budget, a similar figure for the seasonal budgets (=1 figure, 4 panels) would be a valuable addition to the paper and would serve as guidance for the reader throughout section 4.1.

**Response:** In the revised manuscript, we have structured the description of the different seasonal sections as consistently as possible, with a description of (1) the atmospheric and hydrodynamic situation, then of (2) the biogeochemical fluxes, (3) the physical fluxes, (4) the air-sea fluxes, and finally of (5) the resulting variation of DIC content, and a summary of the budget in the upper layer. Besides, we have included in Figure 7 a panel with a similar figure

as Figure 12 for each season, and have removed from Figure 7 the panel (b) to avoid redundancy with the new sub-figure. We have merged Figures 8 and 9 to decrease the number of figures in this part.

[Figure]

*"Figure 7c: Scheme of cumulative seasonal fluxes in mol C m-2 over the respective periods (fall: 88 days, winter: 116 days, spring: 74 days and summer: 87 days). Resp. stands for respiration and GPP for gross primary production. The direction of the arrows indicates the direction of the fluxes and positive values correspond to DIC inputs for the deep convection area."*

4. For the sensitivity experiment regarding calcification: I was surprised to see an enhanced oceanic CO2 uptake relative to the reference case in the experiment accounting for calcification. For such an experiment, I would expect less oceanic CO2 uptake, given that the impact of calcification on alkalinity is twice that on DIC (thus increasing seawater pCO2 at the surface). Going back to your method section 2.1.4, I noticed that you only specified the impact of calcification on DIC – did you also include its impact on alkalinity in your sensitivity test? How did you parametrize dissolution at depth? I note that I realize that either way, this will not impact the outcomes of the main findings of this study, but if this was indeed a mistake, I suggest that the authors correct it.

**Response:** We thank Reviewer 2 for raising this point. We acknowledge there was an error in the sensitivity test on calcification process, by omitting to take into account the process in the rate of change of alkalinity (excess negative charge denoted $\sum[-]$). We apologize for this error. We have corrected it by adding in the equation of the rate of change of alkalinity (excess negative charge denoted $\sum[-]$) the term of calcium carbonate production added in the DIC equation multiplied by 2 (Middelburg et al., 2019). In the new results, the air-sea flux could be reduced by 16% to 57% in considering calcification processes. We have modified the text and Figure 14 (Figure 13 in the new version of the manuscript) in the discussion section on the sensitivity tests on air-sea $CO_2$ flux, in Sect. 2.1.4 "Sensitivity tests" and in the conclusion. Regarding the dissolution at depth it was not taken into account in these sensitivity tests.

Middelburg, J. J.: Marine Carbon Biogeochemistry A Primer for Earth System Scientists, Springer B., edited by Springer Briefs in Earth System Sciences, Springer Briefs in Earth System Sciences, 2019.

Section 2.1.4 "Sensitivity tests", in L. 288-291:
*"Thus, by assuming the ratio of calcium carbonate production to NCP is close to the PIC:TOC ratio, we added in Eq. 1 a consumption term representing 36% of NCP for the mean value of PIC:POC ratio, and 22% and 55% for the minimum and maximum ratio values, respectively. This term, multiplied by 2, was added in the equation of the rate of change of the excess negative charge (Middelburg, 2019)."*

Discussion section, Section 5.2 "Estimate of the annual air sea $CO_2$ flux et and its uncertainties, in L. 659-665:
*"Finally, sensitivity tests taking into account supplementary consumption terms in the equation of DIC and excess of negative charge for $CaCO_3$ precipitation (Sect. 2.1.4) were performed to assess its potential influence on air-sea $CO_2$ flux. They show that not taken into account calcification processes could lead to an  overestimation of the annual air-sea $CO_2$ uptake by 16 to 57% with estimates of 0.29 mol C m$^{-2}$ yr$^{-1}$, based on the mean PIC:POC ratio and NCP (varying between 0.20 and 0.36  mol C m$^{-2}$ yr$^{-1}$ based on the maximum and minimum PIC:POC ratios, respectively), and of 0.40 mol C m$^{-2}$ yr$^{-1}$, based on the parametrization used in Lajaunie-Salla et al. (2021)."*

[Figure]

*Figure 14: Sensitivity tests to the parameterization of gas transfer velocity, the variability of the mole fraction of $CO_2$ in the atmosphere, and the calcification processes, on the annual air-sea $CO_2$ flux estimate. The black bar indicates the annual estimate in the reference simulation, grey bars the mean value for each of the three sets of sensitivity tests.* ***For the sensitivity tests on the parametrization of gas transfer (from 2 to 9), relations with a quadratic (2), hybrid (3 to 5), cubic (6) wind speed dependency are, respectively, in light pink, yellow and orange, and relations that include explicit bubble parametrizations (7 to 9) are in dark pink. For the test (14) on calcification processes, the bar indicates the result found for the mean PIC:POC ratio, while the black line indicates the range using the minimum and maximum PIC:POC ratios.***

Conclusion, in L. 847-849:

*"Moreover, we displayed that* neglecting *calcification processes could lead to an* over*estimation by* 16 *to* 57*% of the annual uptake, highlighting the need for the refinement of the model in future studies."*

5. Language: I spotted numerous (minor) grammar mistakes, e.g., related to prepositions (see detailed comments below). While this did not impact the readability much, I encourage the authors to carefully check the text again during the revisions.

**Response:** We warmly thank Reviewer 2 for all the grammar corrections and apologize for these errors. We have carefully checked the text again.

**Detailed comments:**

**L. 22:** Maybe better: "seasonal and annual budget"?

**Response:** The sentence has been changed as suggested in the revised manuscript, L. 24.

**L. 26:** "reduction of oceanic $CO_2$ uptake"

**Response:** The sentence has been changed as suggested in the revised manuscript, L. 28.

**L. 27:** I suggest rephrasing this sentence by being more specific: Aren't the physical fluxes (of DIC) always larger than the biological ones? How are both dominant?

**Response:** The vertical and horizontal fluxes are always both one order of magnitude higher than the net biogeochemical fluxes (community respiration minus gross primary production). However, the net physical flux, i.e. vertical flux plus lateral flux, is of the same order of magnitude as the net biogeochemical flux for each season (Figures 7 and 12 in the previously submitted manuscript); their magnitude is higher than the one of biogeochemical flux in winter and summer (cumulative fluxes: 4.45 versus -1.53 mol C m$^{-2}$ over the winter period, 0.79 versus -0.57 mol C m$^{-2}$ over the summer period), and smaller in fall and spring (cumulative fluxes: 0.49 versus 0.56 mol C m$^{-2}$ over the fall period, -0.80 versus -2.19 mol C m$^{-2}$ over the summer period). At the annual scale, the net physical flux is 3.3 mol C m$^{-2}$ yr$^{-1}$ and represents 88% of the net biogeochemical consumption flux, while the air-sea flux is one order of magnitude smaller and represents 13% of the biogeochemical flux. We have rephrased the sentence, as follows, in L. 29-32:

*"We highlight the  major role in the annual dissolved inorganic carbon budget of both the  biogeochemical and physical fluxes that amount to -3.7 mol C m$^{-2}$ yr$^{-1}$ and 3.3 mol C m$^{-2}$ yr$^{-1}$, respectively, and are one order of magnitude higher than the air-sea $CO_2$ flux."*

**L. 28:** define "upper"

**Response:** We have specified "upper" as follows, L. 32:

*"The upper layer (from the surface to 150 m depth) of the northwestern deep convection region […]"*

**L. 29:** I suggest replacing "air-sea flux" by "oceanic $CO_2$ uptake"

**Response:** The sentence has been changed as suggested in the revised manuscript, L. 34.

**L. 37:** "comparable role to…" and "processes for carbon"; carbon transfer from where to where? Please specify.

**Response:** The sentence has been changed for more clarity, as follows, L. 41-42:

*"Physical mechanisms can  play a comparable role to that of  biogeochemical processes on  air-sea $CO_2$ flux at regional and global scales …"*

**L. 42:** I suggest rephrasing to "taken up at the ocean surface"

**Response:** The sentence has been changed as suggested in the revised manuscript, L. 46.

**L. 43**: If there is an "on the other hand", I am immediately looking for "one the one hand". Maybe better: "at the same time" or "simultaneousy"?

**Response:** "on the other hand' has been replaced by "furthermore" in the revised manuscript, L. 48.

**L. 56:** moderate phytoplankton bloom

**Response:** The sentence has been changed as suggested in the revised manuscript, L. 67.

**L. 55-56**: Is it typical in the Mediterranean science community to refer to the fall bloom as the "first" bloom? I realize this is a matter of defining the start of the growing season, but from all other regions globally, I am used to describing the bloom phenology starting with the strong first bloom in spring after nutrients were replenished in winter and a secondary typically weaker bloom in the fall.

**Response:** Some previous studies which determined the date of the onset in the Mediterranean Sea (Bernardello et al., 2012; Lavigne et al., 2013) were based on the work by Henson et al. (2009) who determined the bloom start in the North Atlantic Sea by adjusting the method of Siegel et al (2002) and considering the 1st September as the beginning of the annual period, to capture the start of the subtropical bloom that occurs in autumn. Using satellite derived-chlorophyll data, Lavigne et al. (2013) found a bloom starting in autumn (late November / early December) in all the Mediterranean bioregions defined by D'Ortenzio and Ribera d'Alcala (2009). Kessouri et al. (2018) calculated the date

of the bloom onset in the Western Mediterranean Sea using the same biogeochemical model (without the carbonate system module) as used in this study. They also found a start bloom in autumn for the three considered western Mediterranean regions (deep convection zone, shallow convection zone and stratified region). In their results, contrary to the two other regions, in the deep convection region the bloom is interrupted during the deep mixing period and a second bloom start was found when the water column stratified again. However, in other studies, the description of the annual chlorophyll cycle is described from January to December and thus the spring bloom is mentioned before the autumnal bloom (Bosc et al., 2004). Our study was performed in the continuity of Kessouri et al. (2018) and thus we preferred keeping the same annual period. In the revised version, we have removed "first" and "secondary" from the sentence in L. 67-68.

Bosc, E., Bricaud, A., Antoine, D.: Seasonal and interannual variability in algal biomass and primary production in the Mediterranean Sea, as derived from 4 years of SeaWiFS observations. Global Biogeochemical Cycles, 18, GB1005. https://doi.org/10.1029/2003GB002034, 2004.

D'Ortenzio, F., Ribera d'Alcala, M.: On the trophic regimes of the Mediterranean Sea: A satellite analysis. Biogeosciences, 6, 139–148. https://doi.org/10.5194/bg-6-139-2009, 2009.

Henson, S. A., Dunne, J., Sarmiento, J.: Decadal variability in North Atlantic phytoplankton blooms. Journal of Geophysical Research, 114, C04013. https://doi.org/10.1029/2008JC005139, 2009.

Lavigne, H., D'Ortenzio, F., Migon, C., Claustre, H., Testor, P., Ribera d'Alcala, M., et al.: Enhancing the comprehension of mixed layer depth control on the Mediterranean phytoplankton phenology. Journal of Geophysical Research: Oceans, 118, 3416–3430. https://doi.org/10.1002/jgrc.20251, 2013.

Siegel, D. A., Doney, S. C., and Yoder, J. A.: The North Atlantic spring phytoplankton bloom and Sverdrup's critical depth hypothesis. Science, 296, 730–733. https://doi.org/10.1126/science.1069174, 2002.

**L. 57:** "nutrients to the euphotic layer"

**Response:** The sentence has been changed as suggested in the revised manuscript, L. 68.

**L. 68**: Please add a reference to Fig. 1.

**Response:** The reference to Fig. 1 has been added in the sentence in the revised manuscript, L. 80, as suggested.

**L. 71:** "which bring DIC-rich water to the surface"

**Response:** The sentence has been changed as suggested in the revised manuscript, L. 82.

**L. 78**: Maybe "complemented" instead of "enriched"?

**Response:** The sentence has been removed to make more concise the description of the previous studies in the Mediterranean Sea in the revised manuscript, as suggested in the first main comment.

**L. 79**: delete "fixed"

**Response:** As with the previous point, the sentence has been deleted in the revised manuscript.

**L. 81:** "drives an increase in surface $pCO_2$"

**Response:** As with the previous points, the sentence has been deleted in the revised manuscript.

**L. 83**: model instead of modelling

**Response:** As with the previous points, the sentence has been deleted in the revised manuscript.

**L. 83-84:** I am not sure what this approach means. Can you rephrase this part?

**Response:** In response to the fist main comment and in revising the introduction, this sentence has been simplified as follows, in L. 96-99:

*"Finally, D'Ortenzio et al. (2008) and Cossarini et al. (2021), based on 1D models and a 3D model, respectively, found that the whole deep convection region is a major sink of atmospheric $CO_2$ in the open Mediterranean Sea"*

D'Ortenzio et al. (2008) implemented a 1D model in cells of 0.5° x 0.5° horizontal resolution covering the Mediterranean Sea, with no lateral connection between the cells.

**L. 86**: biological instead of biology

**Response:** As with the previous points, the sentence has been deleted in the revised manuscript.

**L. 93:** "limited to"

**Response:** As with the previous points, the sentence has been deleted in the revised manuscript.

**L. 92-94:** This sentence was very confusing to read due to all the "or". Can you rephrase or split it into two?

**Response:** In revising the introduction, this sentence has been merged with the following one, as follows, L. 108-112:

*"In the previous studies, the 3D dynamics of the $CO_2$ system over an annual cycle has never been specifically explored for the whole northwestern deep convection region and a complete DIC budget is still lacking for this region."*

**L. 94-96**: To me, this knowledge does not yet become clear enough from what is written up to this point. I suggest revising the introduction to more clearly highlight the knowledge gaps and why these matter.

**Response:** We have modified the introduction to more clearly highlight the knowledge gaps. We hope it is clearer in the revised version of the introduction.

**L. 104:** "by a positive net community production"

**Response:** The sentence has been changed as suggested in the revised manuscript, L. 120.

**L. 108:** Maybe better: "take advantage of" instead of "benefit from"

**Response:** The sentence has been changed as suggested in the revised manuscript, L. 124.

**L. 112:** Throughout the paper, you sometimes say "biological" and sometimes "biogeochemical". I suggest to consistently use one because from what I can see (please correct me if I am wrong), you are always referring to the same processes.

**Response:** The term "biological" has been replaced by "biogeochemical" throughout the paper when referring to the same processes.

**L. 120 & L. 125:** Have the different models been evaluated in detail over the bigger study regions in any of these studies? It might help to explicitly state that for the interested reader.

**Response:** The biogeochemical model implemented over the whole Mediterranean and forced by the outputs of the operational hydrodynamic model NEMO operated by Mercator was assessed by Kessouri (2015) in terms of spatial and temporal surface chlorophyll and vertical distribution of chlorophyll and inorganic nutrient. This has been specified in Section 2.1.1 "The coupled hydrodynamic-biogeochemical-chemical model", L. 145. The western Mediterranean biogeochemical model was assessed over the western Mediterranean in Kessouri et al. (2018) through comparisons with satellite chlorophyll data.

L. 143: *"The model has been used to study biogeochemical processes in the NW (northwestern) Mediterranean deep convection area (Herrmann et al., 2013; Auger et al., 2014; Ulses et al., 2016; 2021; Kessouri et al., 2017; 2018)* and in the whole Mediterranean Sea (Kessouri, 2015)."

Kessouri, F.: Cycles biogéochimiques de la Mer Méditerranée : processus et bilans, Ph.D. thesis, Université Toulouse 3, 2015.

**L. 124:** How are particle dynamics parametrized in the model? Given that sinking fluxes of biologically-derived particles are an important part of your study, some information on that will be helpful.

**Response:** To take into account particle dynamics in the model, we consider a constant settling velocity, $w_s$, for the slow and fast sinking particulate organic matter and for micro-phytoplankton. The values of the settling velocity have been given in Section 2.1.1, L. 141-143:

*"Particulate organic detritus and micro-phytoplankton have a constant settling velocity (1 m day$^{-1}$ for slow sinking detritus and micro-phytoplankton, and 90 m day$^{-1}$ for fast sinking detritus)."*

The settling of particles is taken into account using the following advection-diffusion equation allowing the calculation of the "physical" rate of change of the concentration *C*, the concentration of each biogeochemical state variable:

$$\frac{\partial C}{\partial t} + \frac{\partial uC}{\partial x} + \frac{\partial vC}{\partial y} + \frac{\partial (w - w_s)C}{\partial z} = \frac{\partial}{\partial z}\left(K_z \frac{\partial C}{\partial z}\right) + F_c$$

where u, v and w are the three components of the current velocity, $K_z$ is the vertical diffusivity and $F_c$ is the source or sink term from rivers, atmosphere and sediment.

**L. 131:** Before looking up the cited references, it was unclear to me how the version before can resolve the cycling of carbon without including DIC. I suggest clarifying that only particulate organic carbon was included before.

**Response:** As suggested by Reviewer 2 we have added a sentence to clarify this point, L. 146-147.

*"In previous versions of the model, particulate and dissolved organic carbon was considered, but the dynamics of dissolved inorganic carbon was not described."*

**L. 136:** "is the respiration"

**Response:** The sentence has been changed as suggested in the revised manuscript, L. 157.

**L. 142:** "not the case for total alkalinity"

**Response:** The sentence has been changed as suggested in the revised manuscript, L. 163.

**L. 146**: Maybe add "throughout the water column" if that is what it is.

**Response:** We have added "throughout the water column" in this sentence, as suggested in the revised manuscript L. 167-168.

**L. 147-149:** Personally, I wouldn't call a paper from 2005 "present knowledge". There are several studies that, albeit of course not perfect, have parametrized it. Thus, I suggest rephrasing this part.

**Response:** This part has been rephrased L. 168-170, as follows:

"*Regarding the  CaCO$_3$ precipitation,  we are aware that future refinements will have to take* this […]"

**L. 149:** "tests on this"

**Response:** The sentence has been changed as suggested in the revised manuscript, L. 171.

**L. 165:** Please add a reference to Fig. 1.

**Response:** A reference to the insert in Fig. 1 has been added in the revised manuscript, L. 186.

**L. 167:** I suggest adding "have been described in detail in X and Y and will be summarized here."

**Response:** The sentence has been changed as suggested in the revised manuscript, L. 190-191.

**L. 169**: It is unclear to me what "hydrodynamic analyses" are. Please clarify and possibly rephrase.

**Response:** Hydrodynamic analyses represent here the hydrodynamic solutions from the NEMO numerical model computed with the Mercator near real time configuration

PSY2V2R4 that embeds assimilation of data in order to constrain and increase realism to the numerical solution. We have replaced "analyses" by "fields" in the text, L. 194.

**L. 167-175:** I found the description of the steps rather difficult to follow. I think adding a flow chart detailing the different steps could help a lot.

**Response:** As answered to the second main comment, to clarify this point we have added a figure with a scheme of the three steps in the Supplementary Material (new Fig. S1).

**L. 179:** Given that the model simulates the negative charge and not alkalinity, did you correct the measured alkalinity to correspond to the model tracer? Please clarify.

**Response:** We apologize for the confusion. Yes, we deduced the initial values of the excess negative charge based on Eq. 2, using measurements of total alkalinity and nutrients concentrations. We have added a sentence to clarify this point, L. 204-206:

*"To deduce the excess negative charge from total alkalinity (Eq. 2), we also used the nutrient concentration data from the Medar/Medatlas database as in Kessouri et al. (2017)."*

**L. 183:** What is a "rigorous mathematical approach"? Please clarify or delete.

**Response:** We have deleted "rigorous mathematical approach" and have rephrased the sentence, in L. 206-208, as follows:

*"Recently, Davis and Goyet (2021)  described a  method based upon the property variability, to precisely quantify the uncertainties at any point of an interpolated data field."*

**L. 189:** You only specify what was used for winds here. What about other atmospheric forcing variables (e.g., radiation, humidity, precipitation etc.)? Please be complete.

**Response:** We have completed and added all the other forcing variables needed for the gas transfer velocity calculation, in L. 226, knowing that the hydrodynamic model uses other atmospheric variables such as air temperature, precipitation, longwave and shortwave radiation:

*"To compute the gas transfer velocity, we used the 3-hour wind speed, pressure, and humidity provided by the ECMWF model on a 1/8° grid, in consistency with the hydrodynamic simulation."*

**L. 191:** What I am missing here is a description on the model run time in each step. Also, in L. 179 you mention an initialization in summer 2011, while I think (if I understood correctly), the final model was run from September 2012 onwards. Could you clarify? My confusion on this point convinces me even more that a flow chart detailing the model setup procedure would help.

**Response:** We apologize for the confusions. As answered to previous comments, we have added a figure with a scheme of the three steps in the Supplementary Material (new Fig. S1) in which we have specified the period of each simulation. The biogeochemical simulation over the whole Mediterranean Sea (step 1b in the new Fig. S1) was performed from 15 June 2011 to 15 November 2013. We initialized the $CO_2$ system module using interpolated data as it was described L. 178-183 in the previously submitted manuscript. The biogeochemical simulation over the Western Mediterranean (step 2 in the new Fig. S1) was performed over the period from 15 August 2012 to 30 September 2013, and was initialized using the model outputs of the whole Mediterranean Sea simulation (step 1b). To avoid confusions, we have moved the description of the initialization of step 1b before describing step 2, L. 200-208.

**L. 193:** I find "DIC flows" and "inventory variations" rather confusing. Maybe "DIC fluxes" and "inventory tendencies"? Please check throughout the text.

**Response:** We have rephrased the sentence, in L. 233, and have changed "flows" by "fluxes", "transport", "export" or "input" throughout the text. If Reviewer 2 suggests the terms "change" or "time evolution" are clearer than "variation", and the terms "stock" or "content" are clearer than "inventory", we will follow her/his recommendations.

*"We computed DIC  fluxes and the resulting variation in the DIC inventory for the whole deep convection area."*

**L. 194:** "for at least 1 day"

**Response:** The sentence has been changed, as suggested, in the revised manuscript L. 234.

**L. 201:** Given the title of this section, I wonder if Eq. 1 is better to be placed here. Additionally, I think at least the general budget equation (Eq. S1) should be moved to the main text.

**Response:** We would prefer to keep Eq. 1 in Section 2.1.1, since it gives the biogeochemical rate of change of the state variable DIC at the model grid points. As suggested, we have replaced the text describing the budget "The biological term of the budget […] upper layer is given in Supplementary Material (Text S1)" in section "Study area and computation of DIC balance" by Text S1, in L. 240-267.

**L. 203**: What do you mean by "internal variation"? Please clarify.

**Response:** "internal variation" meant variation of the content of DIC during a considered period. It is given in Eq. S1 of the previously submitted version (Eq. 4. in the revised version) The sentence has been removed in the new version of the manuscript by answering the previous comment. We have also replaced it in the caption of Figure 7.

**L. 215:** Please add a reference to the respective Equation.

**Response:** We have added a reference to Eq. 1 in the revised manuscript, L. 283.

**L. 216:** "as 0.5"

**Response:** This sentence has been modified to take into account a comment of Reviewer 1, L. 285-286.

*"Miquel et al. (2011) estimated that the PIC:POC ratio at 200 m depth varied between 0.31 and 0.78, with a mean value of to 0.5, based on sediment trap measurements at the EMSO-DYFAMED site."*

**L. 220:** Please be precise: NCP does not appear as such in Eq. 1.

**Response:** We agree, the sentence was confusing, We have moved "in Eq. 1" L. 289-291, as follows:

*" [...] we added in Eq. 1 a consumption term representing 36% of NCP for the mean value of PIC:POC ratio, and 22% and 55% for the minimum and maximum ratio values, respectively in Eq. 1."*

**L. 221:** Please state here what the parametrization by Lajaunie-Salla et al. (2021) is. Ideally, the reader should not have to look up other papers to understand what you're doing.

**Response:** In Lajaunie-Salla et al. (2021), carbonate precipitation, named *Precip*, is given by the following equation:

$$\text{Precip} = k_{\text{precip}} \frac{(\Omega_c - 1)}{0.4 + (\Omega_c - 1)} \sum_{i=1}^{3}(GPP_i - RespPhy_i)$$

where $k_{\text{precip}}$ is the PIC:POC ratio and $\Omega c$ the aragonite saturation, which we set at 3.5 based on Schneider et al. (2007).

We have added the equation in the text, L. 293:
*"In a second sub-test, we added a CaCO₃ production term based on the parametrization used in the Gulf of Lion's shelf modeling study by Lajaunie-Salla et al. (2021) (their Table A4, $Precip = k_{precip}\frac{(\Omega_c-1)}{0.4+(\Omega_c-1)}\sum_{i=1}^{3}(GPP_i - RespPhy_i)$, where $k_{precip}$ is the PIC:POC ratio and $\Omega c$ the aragonite saturation, set at 3.5 based on Schneider et al., (2007))."*

**L. 225:** sea surface

**Response:** The sentence has been changed as suggested in the revised manuscript, L. 297.

**L. 284**: I suggest adding "reflecting a" in front of "period"

**Response:** The sentence has been changed as suggested in the revised manuscript, L. 356.

**L. 298:** Does the southern zone include everything south of 41°N or is there a southern limit?

**Response:** The southern zone includes all stations south of the convection zone. There is no southern limit.

**L. 299:** I assume the depth profiles have been subsampled to only include the cruise locations shown in Fig. 3. Please clarify.

**Response:** The modeled mean profiles shown in Fig. 5 correspond to the average of the modeled profiles extracted at the same location and date as the measurement stations. This has been specified in the new text, in L. 371-372:

*"Comparisons were performed by extracting model outputs at the same date and location as measurements."*

**L. 324:** Do you mean "alternating" instead of "alternative"?

**Response:** Yes, the sentence has been changed, L. 395, as suggested.

**L. 325:** Where can the direction of the wind be seen? If this is previous knowledge for the region of interest and you therefore decided not to show this explicitly, please make sure it is introduced in the introduction for clarity.

**Response:** We have specified the wind direction in Figure S2e-f of the new version of the Supplementary Material, by adding two panels with maps of wind velocity.

**L. 337:** Unless I misread something, I think the minus sign should be omitted (the cumulative flux is positive according to Fig. 7).

**Response:** In fall, the cumulative air-to-sea flux is negative, we are sorry if it was not clear on Figure 7b of the previously submitted version. As recommended in the third main comment, we have added a figure (Figure 7c in the new manuscript) with schemes of the seasonal budgets for which the direction of the flux will be clearer.

**L. 344:** Do you mean "DIC concentration in the ML" or "the DIC flux into the ML"? Please clarify.

**Response:** We meant a decrease in "DIC concentration" visible at the end of October and end of November in Figure 10b (Fig. 9b in the revised manuscript). We have slightly modified the sentence in L. 415-416, as follows:

*"This led, notably, temporally, to a  low  DIC concentration in the mixed layer at the end of October and end of November (Fig. 9b)."*

**L. 441:** "episodes of heat gain"

**Response:** The sentence has been changed as suggested in the revised manuscript, L. 541.

**L. 466:** To me, it is odd to call this flux biological production, when this is in fact remineralization/respiration. I understand why you do it and it is technically correct, but I still suggest rephrasing to avoid confusion.

**Response:** We agree that the term "production" can be confusing. We have replaced this term by 'gain' here, L. 575-576, and a more appropriate term throughout the text and in figures:

*"Within the sea, biogeochemical processes induced an annual DIC consumption of 3.7 mol C m$^{-2}$ yr$^{-1}$ of DIC in the upper layer and a productionDIC gain of 2.3 mol C m$^{-2}$ yr$^{-1}$ in the deeper layers."*

**L. 469:** For consistency with how you described the biological component, it would be easier to read if you also reflected the sign convention in your wording here.

**Response:** We have modified the sentence to reflect the sign convention, L. 579-580, as follows:

*"Our estimate of net physical fluxes (lateral plus vertical) is an input of 3.3 mol C m$^{-2}$ yr$^{-1}$ into the upper layer and an export of -11.0 mol C m$^{-2}$ yr$^{-1}$ infrom the deeper layer."*

**L. 474:** I suggest deleting "an amount".

**Response:** The sentence has been changed as suggested in the revised manuscript, L. 584.

**L. 486:** Please see my comment on the abstract regarding "both dominate". I suggest to also rephrase here.

**Response:** We have also rephrased the introduction of the discussion section, by merging the two last sentences, in L. 596-599:

*"Our results show that  biogeochemical and physical processes,  through their impacts on DIC concentration,  have both a major role in the intensity and sign of the air-sea exchanges in the deep convection area."*

**L. 489:** Here and throughout the discussion section: Can you find more descriptive/informative section titles? It is incredibly useful to the reader if the title of each section already conveys information, i.e., ideally the main take-away message.

**Response:** We have modified titles of the discussion section, as follows:

- *"5.1 The pCO2"* **to** *"5.1 Assessment of the seasonal cycle of the pCO$_2$"*
- *"5.2 The air-sea CO$_2$ flux"* **to** *"5.2 Estimate of the annual air-sea CO$_2$ flux and its uncertainties"* **and** *"5.3 Comparisons on air-sea CO$_2$ flux with previous studies in the Mediterranean Sea"*
- *"5.3 Physical flows in the deep convection area"* **to** *"5.4 The major influence of physical transport in the DIC budget of the deep convection area"*
- *"5.4 Net community production and air-sea fluxes"* **to** *"5.5 Net community production and air-sea fluxes relationships"*

**L. 490-502:** As far as I can see, these are results. I am not convinced this part is necessary.

**Response:** We have removed most of this part. Some elements have been kept to make the comparisons easier with previous studies. To be consistent with this modification and the change of titles of the subsection of the discussion, we have also removed the first paragraph of the following section on air-sea CO$_2$ flux.

**L. 508-509:** This sentence is unclear to me. Can you rephrase?

**Response:** We have rephrased this sentence, in L. 619-622, as follows:

*"The high frequency measurements at the CARIOCA buoy described by Hood and Merlivat (2001) and Merlivat et al. (2018) indicated  an interannual variability of 4-5 weeks in the date  at which the pCO$_2$ difference changes sign,  depending on the interannual variability of air-sea heat flux  and  of the bloom onset."*

**L. 528-530:** Here and throughout the text: Try to avoid 1-2 sentence paragraphs.

**Response:** We have avoided this as much as possible throughout the text.

**L. 552:** Please see my major comment on these sensitivity experiments.

**Response:** As already answered to the fourth main comment, we have corrected this error in the equation of the rate of change of alkalinity (excess negative charge denoted $\sum[-]$) and have again performed the sensitivity tests. In the new results, the air-sea flux could be reduced by 16% to 57% if carbonate production is taken into account. We have modified the text, L. 659-665 (as well as Figure 14 (Figure 13 in the revised version) in the discussion section on the sensitivity tests, in Sect. 2.1.4 "Sensitivity tests" and in the conclusion):

*"Finally, sensitivity tests taking into account supplementary consumption terms in the equation of DIC and excess of negative charge for $CaCO_3$ precipitation (Sect. 2.1.4) were performed to assess its potential influence on air-sea $CO_2$ flux. They show that not taken into account calcification processes could lead to an  overestimation of the annual air-sea $CO_2$ uptake by 16 to 57% with estimates of 0.29 mol C $m^{-2}$ $yr^{-1}$, based on the mean PIC:POC ratio and NCP (varying between 0.20 and 0.36 mol C $m^{-2}$ $yr^{-1}$ based on the maximum and minimum PIC:POC ratios, respectively), and of 0.40 mol C $m^{-2}$ $yr^{-1}$, based on the parametrization used in Lajaunie-Salla et al. (2021)."*

**L. 566:** Is there a "$yr^{-1}$" missing? Additionally, it would help to provide the range based on your model here again to compare to the cited paper more easily.

**Response:** Yes, we have corrected the unit by adding a "$yr^{-1}$" (L. 680) and have added in the following sentence the range of the model estimates to make the comparison easier, in L.680-681:

*"The larger homogeneity in our estimates (varying between -0.1 and 1.2 mol C $m^{-2}$ $yr^{-1}$ inside the deep convection area) could be partly ascribed to the horizontal diffusion and advection that were accounted for in our model."*

**L. 576:** It might be more appropriate to say "physical transport".

**Response:** The title has been changed as suggested in the revised manuscript, L. 701.

**L. 577:** "the vertical DIC distribution"

**Response:** The sentence has been changed as suggested in the revised manuscript, L. 702.

**L. 581:** "greater magnitude" – Please specify the sign.

**Response:** We have specified the sign of the fluxes in L. 705-707, as follows:

*"They both show a similar seasonal cycle with greater magnitude (positive for the vertical transport and negative for the lateral transport with regard to the upper layer) in fall, the preconditioning phase [...]"*

**L. 582:** "sea heat loss" Do you mean "ocean heat loss"? Please clarify.

**Response:** We have replaced "sea heat loss" by "sea surface heat loss", L. 707.

**L. 589:** Please rephrase "DIC exchange flows".

**Response:** We have replaced "DIC exchange flows" by "DIC fluxes at the limits of the deep convection area", L. 715.

**L. 595:** "as illustrated in"

**Response:** The sentence has been changed as suggested in the revised manuscript in L. 722.

**L. 608:** "slowed down" instead of "braked"

**Response:** The sentence has been changed as suggested in the revised manuscript, L. 736.

**L. 617:** "convection" instead of "convention"?

**Response:** We have corrected this error in the revised manuscript, L. 744.

**L. 633:** "from" instead of "into"?

**Response:** The DIC budget shows a lateral DIC transport from the surrounding region into the deep layer of the deep convection region (Figure 11 in the revised manuscript). We have changed the sentence, L. 759-763, as follows:

*"More specifically, we found that the lateral exchanges with the surrounding region were characterized by a net lateral input of total carbon into the deep layers of the deep convection region, although organic carbon was exported towards the surrounding region, and a net lateral export of both organic and inorganic carbon in upper water masses (Fig. 11)."*

**L. 634:** "a lateral outflow"

**Response:** The sentence has been changed in the revised manuscript L. 762.

**L. 640-646:** It would be a lot easier to compare to the findings of your studies, if you reported these numbers as flux densities instead of as integrated fluxes (or to here report your findings in the same integrated unit).

**Response:** Our estimate was reported as integrated fluxes L. 530 and L. 645 of the previously submitted manuscript: 0.4 Tg C $yr^{-1}$. We have slightly changed the sentence L. 773-774, as follows:

*"Thus the NW Mediterranean deep convection area, which represents 2.5% of the Mediterranean Sea surface, and which we estimate here absorbed at the sea surface 0.4 Tg C $yr^{-1}$, could strongly contribute to the uptake of atmospheric $CO_2$ in the open Mediterranean Sea."*

**L. 648:** "into" instead of "in"

**Response:** The sentence has been changed as suggested in the revised manuscript, L. 777.

**L. 666:** I suggest adding "…and rising atmospheric CO2 levels".

**Response:** We have added this in the sentence as suggested in the revised manuscript, L. 809.

**L. 680:** budgets

**Response:** The sentence has been changed as suggested in the revised manuscript, L. 823.

**L. 691:** What exactly are the first and second part here? Please clarify.

**Response:** The sentence has been changed L. 834-836, as follows:

*"The region was marked by a deficit of $CO_2$ compared to the atmosphere from November to early June the second part of fall to the first part of spring, which led to a 7-month ingassing of atmospheric $CO_2$"*

**L. 701:** "subject to"

**Response:** The sentence has been changed as suggested in the revised manuscript, L. 844.

**Figures:**

**Fig. 3:** Please specify for what depth(s) the model output is shown here.

**Response:** In the caption, we indicated that the model outputs are "modeled at 3 m depth" in the previously submitted version. We have added the depth of the model outputs also in the legend in the top of the figure.

[Figure]

*Figure 3: Time series of (a) temperature, (b) salinity, (c) pCO₂, (d) DIC, (e) total alkalinity, and (f) pH at total scale, modeled at 3 m depth (line in black), observed (small red dots at BOUSSOLE site and pink points at EMSO-DYFAMED site between 3 and 14 m depth) and computed with CANYON-MED neural networks (small blue dots at BOUSSOLE at 3 m, blue squares at EMSO-DYFAMED site between 3 and 14 m depth, error bars are indicated in gray). Correlation coefficient, RMSE and bias between model outputs and BOUSSOLE observations are indicated in (a), (b) and (c).*

**Fig. 4:** I suggest adding a legend/title above each column.

**Response:** We have added "Observations' and "Model" above the first and second column, respectively.

[Figure]

*Figure 4: Surface dissolved inorganic carbon (DIC) concentration (µmol kg⁻¹) observed (left) and modeled (right) over the (a,b) DEWEX Leg1 (1-21 February 2013), (c,d) DEWEX Leg2 (5-24 April 2013), and (e,f) MOOSE-GE (11 June-9 July 2013) cruise periods. The correlation coefficient (R), root mean square error (RMSE), and bias between surface observed and modeled DIC are indicated in (b,d,f).*

**Fig. 7:** I suggest using the same colors for the same components in all panels, not only in a & b, but also in panel c. Additionally, it is unclear to me why you decided to show the seasonal averages only for the upper layer and not for the deeper layer. Please consider adding the extra panel for completeness.

**Response:** The color for the different components in Fig. 7 was the same color as the same components shown in Fig. 6:
- biogeochemical fluxes in the upper layer in bright green,
- biogeochemical fluxes in the deeper layer in green/brown,
- physical fluxes in the upper layer in light blue,
- physical fluxes in the deeper layer in dark blue.

As recommended in the third main comment, we have added a sub-figure with seasonal budget schemes showing the budgets in the upper and deeper layer and, to avoid redundancies, we have removed Fig. 7b of the previous version.

**Fig. 14:** Please link the caption more clearly to the figure: which bar is which experiment? Only giving the reference requires the reader to be familiar with every single paper, which will not necessarily be the case (it certainly isn't the case for me).

**Response:** To clarify this figure, we have moved the titles of the experiment in the top of the figure. For the first set of experiments, we have also classified and colored the bars according to the type of parametrization of the gas transfer velocity instead of the date of paper publication, and we have added the type of the parameterization in the caption. We have also added the type of parametrization of the gas transfer velocity in Table S1.

[Figure]

*Figure 13: Sensitivity tests to the parameterization of gas transfer velocity, the variability of the mole fraction of $CO_2$ in the atmosphere, and the calcification processes, on the annual air-to-sea $CO_2$ flux estimate. The black bar indicates the annual estimate in the reference simulation, grey bars the mean value for each of the three sets of sensitivity tests. For the sensitivity tests on the parametrization on gas transfer (from 2 to 9), relation with a quadratic (2), hybrid (3 to 5), cubic (6) wind speed dependency are respectively in light pink, yellow and orange, and relations that include explicit bubbles parametrizations (7 to 9) are in red. For the test (14) on calcification processes, the bar indicates the result found for the mean PIC:POC ratio, while the black line indicates the range using the minimum and maximum PIC:POC ratios.*

**All figures:** Please double-check that the sign convention of all fluxes is defined in the respective caption.

**Response:** We have checked this.

Supplementary material: Eq. S2: "DCA" is not defined in the text.

**Response:** DCA was defined in L. 5 of the previous Supplementary Material. We have moved its definition just after the equation (that has been moved in the main text of the revised manuscript as recommended in a previous comment), L. 249:

*"where (x,y,z) belongs to the upper layer (150 m to the surface) of the DCA (deep convection area)."*

**L. 29:** How are sediment fluxes treated in the model? How large are they compared to the other components? Without any further information, it is difficult to judge for the reader to what extent this assumption impacts the role of vertical fluxes (which are treated as the residual and will therefore include any sedimentary contribution).

**Response:** The fluxes of dissolved inorganic carbon, nutrients and oxygen at the sea-sediment interface were calculated using a simplified version of the vertically-integrated dynamic sediment model described in Soetaert et al. (2001). The parameters of the model were set following the study of Pastor et al. (2011) in the Gulf of Lion shelf. The same model was used by Many et al. (2021) who showed that the model results were consistent with previous observational and modeling studies on the Gulf of Lion shelf. In this study, we found a particulate organic carbon deposit of 0.1 mol m$^{-2}$ yr$^{-1}$ in the deep convection area. This is in the same order, but smaller than the sediment flux estimated at 0.2 mol C m$^{-2}$ yr$^{-1}$ by Stabholz et al. (2013) near the bottom in the deep convection area. The authors reported an increase in the flux by one to two orders of magnitude during a winter characterized by deep convection. They attributed this increase to resuspension events induced by strong bottom currents. Durrieu de Madron et al. (2023) also pointed out the influence of dense shelf water cascading which can be responsible for supplementary organic carbon deposit flux. In the model, the efflux of DIC resulting from the sediment organic carbon remineralization is calculated during the simulation and taken into account in the budget but is negligible compared to all the other terms. Further comparison analyses will be needed in the future to verify the model in the deep region. Moreover, a coupling with sediment transport model would allow improving the description of the deposition flux of organic carbon and the modifications in the sediment resulting from resuspension events, not taken into account currently in the model. In the revised manuscript, we have specified how the fluxes are calculated at the sea-sediment interface in L. 229-231, and have indicated that we found a negligible annual DIC efflux in L. 266-267.

Durrieu de Madron X., D. Aubert, B. Charrière, S. Kunesch, C. Menniti, O. Radakovitch, and J. Sola. 2023. Impact of dense water formation on the transfer of particles and trace metals from the coast to the deep in the northwestern Mediterranean. Water, 15, 2: 301. doi: 10.3390/w15020301.

Stabholz, M., Durrieu de Madron, X., Canals, M., Khripounoff, A., Taupier-Letage, I., Testor, P., Heussner, S., Kerhervé, P., Delsaut, N., Houpert, L., Lastras, G., and Dennielou, B.: Impact of open-ocean convection on particle fluxes and sediment dynamics in the deep margin of the Gulf of Lions, Biogeosciences, 10, 1097–1116, https://doi.org/10.5194/bg-10-1097-2013, 2013.

---

## Author Response (AR2)

**Seasonal dynamics and annual budget of dissolved inorganic carbon in the northwestern Mediterranean deep convection region**

Caroline Ulses, Claude Estournel, Patrick Marsaleix, Karline Soetaert, Marine Fourrier, Laurent Coppola, Dominique Lefèvre, Franck Touratier, Catherine Goyet, Véronique Guglielmi, Fayçal Kessouri, Pierre Testor, Xavier Durrieu de Madron

**Responses to the Reviewer' comments**

Dear Editor,
We thank you for the consideration of the revised manuscript.
We have considered all the reviewer's comments and have addressed them point by point. Reviews are included in black font, answers in blue font and the modifications done in the revised manuscript in *italic red font*. We have also indicated the line number where the modifications have been done in the revised manuscript with track changes.

**Responses to the comments of Reviewer #1**

We warmly thank Reviewer #1 for his constructive comments and suggestions which help to improve the manuscript.

Review of Ulses et al. (2022): "Seasonal dynamics and annual budget of dissolved inorganic carbon in the northwestern Mediterranean deep convection region"

The new version greatly improved the ms resolving the main weak points pointed out by the reviewers. The discussion has now a wider perspective at the Mediterranean scale.
Therefore I am favourable for the publication of the ms.

We appreciate this positive general comment.

There are only a few minor points that need to be corrected or clarified:

Figure 2: Why in the scheme of the model is considered the phytoplankton deposition but not the zooplankton deposition? The zooplankton contribute to POM through faecal pellets, dead organism not only egestion why only the latter is considered? Bacteria can contribute to DOM formation through exudation, excretion, cellular lysis why these pathways are not considered in the model.

**Response:** The description of the dynamics of zooplankton and bacteria in the Eco3M-S model is based on the model of Anderson and Pondaven (2003), adapted in the Ligurian Sea by Raick et al. (2005). In our model, described in Auger et al. (2011), Ulses et al. (2016) and the Supplement Material in Many et al. (2021), a fraction of the food grazed by zooplankton is not assimilated, lost through messy feeding, corresponding to the breakage of preys (Anderson and Duclow, 2001) and supplies to the dissolved organic matter compartment. The term "egestion" is considered as a constant fraction of the ingested food that is released and supplied to the POM compartments; this includes faecal pellets. The flux "mortality" from the zooplankton compartment to the POM compartment (indicated by a green arrow to the left in Figure 2 diagram), corresponding to dead organisms, is also taken into account in the model.

Only particulate organic matter and micro-phytoplankton have a settling velocity. Living zooplanktons have no settling velocity as in several biogeochemical models applied in the Mediterranean Sea (for instance Levy et al., 1998; Lacroix and Grégoire, 2002; Lazzari et al., 2002; Raick et al., 2005; Palmiéri, 2014; Guyennon et al., 2015). However they indirectly contribute to POM deposition through the production of "egested" material (i.e. faecal pellets) and their mortality. Furthermore, the study of Isla et al. (2015) emphasized the importance of diel vertical migration of zooplankton in the POM export from surface to deep waters. We plan to implement this process in future works to improve the POM and zooplankton vertical distribution.

Regarding the bacteria dynamics, the excretion term in the model is a flux from bacterial to dissolved inorganic matter; cellular lysis is part of the non-grazing bacterial mortality term directed to the DOM compartment. The bacterial exudation was not taken into account in the model of Anderson and Pondaven (2003) and the adapted version by Raick et al. (2005), and was not yet added here in our model. We are conscious that the addition of this process could improve the description of the DOM dynamics and should be tested in future works.

Anderson, T.R., Ducklow, H.W.: Microbial loop carbon cycling in ocean environments studied using a simple steady-state model. Aquatic Microbial Ecology 26, 37 – 49, 2001

Anderson, T.R., Pondaven, P.: Non-redfield carbon and nitrogen cycling in the Sargasso Sea: pelagic imbalances and export flux. Deep-Sea Research I 50, 573 – 591, 2003.

Guyennon, A., Baklouti, M., Diaz, F., Palmieri, J., Beuvier, J., Lebaupin-Brossier, C., Arsouze, T., Beranger, K., Dutay, J. C., and Moutin, T.: New insights into the organic carbon export in the Mediterranean Sea from 3-D modeling, Biogeosciences, 12, 7025–7046,https://doi.org/10.5194/bg-12-7025-2015, 2015.

Isla, A., Scharek, R., Latasa, M.: Zooplankton diel vertical migration and contribution to deep active carbon flux in the NW Mediterranean, Journal of Marine Systems, 143, 86-97, https://doi.org/10.1016/j.jmarsys.2014.10.017, 2015.

Lacroix, G., Grégoire, M.: Revisited ecosystem model MODECOGeL) of the Ligurian Sea: seasonal and interannual variability due to atmospheric forcing. Journal of Marine Systems 37 (4), 229 – 258, 2002.

Levy, M., Mémery, L., and André, J. M.: Simulation of primary production and export fluxes in the Northwestern Mediterranean Sea, J. Mar. Res., 56, 197–238, 1998.

Palmiéri J.: Modélisation biogéochimique de la mer Méditerranée avec le modèle régional couplé NEMO-MED12/PISCES. Ph.D. thesis; . URL: http://www.theses.fr/2014VERS0061/document; thesis supervised by Dutay, J.-C. Bopp, L. et B., K., Sciences de l'environnement Versailles-St Quentin en Yvelines, 2014.

Raick, C., Delhez, E. J. M., Soetaert, K., and Grégoire, M.: Study of the seasonal cycle of the biogeochemical processes in the Ligurian Sea using a 1D interdisciplinary model, J. Marine Syst., 55, 177–203, 2005.

It is not clear to me the difference in the word used Transport and transfer in the figures n.7 and 11 why not to use the same transport for both vertical and horizontal exchanges?

**Response:** We agree with the reviewer: it is clearer to use the same term for both fluxes. "transfer" has been changed to "transport" in Figures 7 and 11.

L. 99 contribution should be "contributions"

**Response:** The sentence has been changed as suggested in the revised manuscript, L. 100.

L. 114. It is not clear from where the settling velocity for detritus were derived.

**Response:** The values of settling velocities for detritus (i.e. 1 m day$^{-1}$ for slow sinking detritus and 90 m day$^{-1}$ for fast sinking detritus) were based on the previous modeling studies in the northwestern Mediterranean Sea of Levy et al. (1998) (1 and 100 m day$^{-1}$ for slow and fast sinking particles, respectively), Lacroix and Grégoire (2002) (1.5 and 95 m day$^{-1}$ for slow and fast sinking particles, respectively), Raick et al. (2005) (1 m day$^{-1}$ for slow sinking particles) and adapted after calibration tests. Levy et al. (1998) set the value of the settling velocity of fast sinking particles using the estimate of 92 m day$^{-1}$ by Miquel et al. (1994) derived from

sediment trap measurements in the Ligurian Sea. Their value for slow sinking particles corresponds to a medium value ranging between the settling velocity of diatoms (from 0.1 to 2.1 m day$^{-1}$, according to Andersen and Nival (1988)) and of small detritus (1 m day$^{-1}$). We have added the references in the revised manuscript, L. 115-116:

*"Particulate organic detritus and micro-phytoplankton have a constant settling velocity (1 m day$^{-1}$ for slow sinking detritus and micro-phytoplankton, and 90 m day$^{-1}$ for fast sinking detritus, based on the modeling studies of Levy et al. (1998), Lacroix and Grégoire (2002), Raick et al. (2005) and calibration tests (Auger et al., 2011; Kessouri et al., 2017))."*

Andersen, V., Nival, P.: A pelagic ecosystem model simulating production and sedimentation of biogenic particles: role of salps and copepods. Mar. Ecol. Prog. Ser., 44, 37–50, 1988.

Miquel, J.-C., Flower, S. W., La Rosa, J., Buat-Menard, P.: Dynamics of the downward flux of particles and carbon in the open Northwestern Mediterranean Sea. Deep-Sea Res., 41, 243–262, 1994.

L. 311-312. The authors should add the increase of temperature to the other factors (oligotrophy, high stratification and domination of respiration) to explain the summer drop of pHT. Moreover the authors could better explain the effect of high stratification to the variation of pH.

**Response:** The increase in temperature and weak vertical mixing in summer induce a high stratification that causes oligotrophy near the surface. These conditions favor a prevalence of respiration over photosynthesis, and thus a production of DIC that induces a drop in pH. We have realized that this sentence was not clear. We have modified it and, as suggested by the reviewer, we have added the increase of temperature to explain the drop of pH in summer, in line with the previous study of Hagens and Middleburg (2016) who calculated the contribution of temperature, dissolved inorganic carbon, total alkalinity and salinity in pH variation (L. 313-315):

*"The pH$_T$ seasonal variation in observations, simulation and CANYON-MED results all indicate a drop in summer,  mostly due to the increase in temperature (Hagens and Middleburg, 2016)."*

Hagens M., Middleburg J.J.: Attributing seasonal pH variability in surface ocean waters to governing factors", Geophysical Research Letters, 43, 12528–12537, 2016.

L.405- "physical supplies" do the authors mean "physical transport"?

**Response:** Yes. "physical supplies" has been replaced by "physical transport", L. 407 and also L. 31-32.

L. 417 I suggest changing vertical and horizontal exchanged transports with "vertical and horizontal exchanges" and in the following line change "transfer" with "transport"

**Response:** We have made the suggested modifications L. 419-420, and also L. 459.

L. 470-471. "The loss of DIC by physical fluxes resulted from a loss by lateral transport and a gain through upward transport." To make the sentence more understandable I would suggest to write "from a prevalence of the loss through lateral transport with respect to the gain through the vertical transport"

**Response:** The sentence has been changed as suggested L. 472-473.

L. 563. A reference about the POC:PIC ratio could be added.

**Response:** The reference for the mean, minimum and maximum POC:PIC ratios, Miquel et al. (2011), has been added L. 564 and 565.

L. 677-683. The huge difference of the contribution to the Mediterranean DIC outflow should be discussed.

**Response:** In this study we estimated a lateral DIC flux of 109 and 73 Tg C $yr^{-1}$ in the surface (surface to 150 m) and intermediate (150 m to 800 m) layers, respectively, from the deep convection area to the surrounding area. We compared these fluxes with the DIC inflow and outflow estimates at the Gibraltar Strait by Aït-Ameur and Goyet (2006). Based on cruise DIC measurements and on previous estimates of water mass transport at the Gibraltar Strait, they obtained a DIC inflow from the Atlantic Ocean in the surface layer ranging between 660 and 1310 Tg C $yr^{-1}$, and a Mediterranean DIC export towards the Atlantic Ocean at the intermediate depths, ranging between 680 and 1380 Tg C $yr^{-1}$. The large range in their DIC flux estimates can be explained by the high discrepancies between the mass transports estimates used in their calculations. The upper range values are based on the estimates of water mass inflow and outflow by Béthoux (1979) of 1.7 Sv and 1.6 Sv, respectively. These values are twice as high as the estimates of 0.81 Sv and 0.76 Sv for the inflow and outflow, respectively, by Baschek et al. (2002), from which Aït-Ameur and Goyet (2006) derived the low values of the DIC fluxes. The review study of Jordà et al. (2017) found a mean value of

the previous estimates derived from observations of 0.86 ± 0.1 Sv and 0.80 ± 0.08 Sv, closer to the lower estimates of Baschek et al. (2002).

The DIC fluxes that we estimated from the surface and intermediate layers of the deep convection area towards the adjacent area represent between 8 and 22% and between 5 and 11% of the DIC inflow and outflow at the Gibraltar Strait, respectively, estimated by Aït-Ameur and Goyet (2006). The high values of these percentages derived from the low water mass transport estimates appear more reliable according to the synthesis study of Jordà et al. (2017). As suggested by Reviewer #1, we have added a discussion on this aspect in the revised manuscript, L. 686-693:

*"The large discrepancy found here in the percentages of DIC fluxes at Gibraltar Strait that DIC outflows from the deep convection area represent, is due to the large range in the DIC flux estimates at the Gibraltar Strait found by Aït-Ameur and Goyet (2006). This latter discrepancy is in turn explained by the large range in the estimates of water mass transport used in their computations, which can be attributed to the complex topography and strong tidal currents at the Strait (Jordà et al., 2017). In their synthesis study on the exchanges at the Gibraltar Strait, Jordà et al. (2017) found a mean value of the water flux estimates based on observations close to those proposed by Baschek et al. (2002), from which Aït-Ameur and Goyet (2006) derived the lower values of DIC fluxes. This suggests that the higher percentage values that we calculated would be more reliable."*

Baschek, B., Send, U., Garcia-Lafuente, J., Candela, J.: Transport estimates in the Strait of Gibraltar with a tidal inverse model. Journal of Geophysical Research 106, 31033–31044, 2002.

Jordà, G., Von Schuckmann, K., Josey, S.A., Caniaux, G., García-Lafuente, J., Sammartino, S., Özsoy, E., Polcher, J., Notarstefano, G., Poulain, P.-M., Adloff, F., Salat, J., Naranjo, C., Schroeder, K., Chiggiato, J., Sannino, G, Macías, D.: The Mediterranean Sea heat and mass budgets: Estimates, uncertainties and perspectives, Progress in Oceanography, 156, 174-208,https://doi.org/10.1016/j.pocean.2017.07.001, 2017.

L. 753-755. It is confusing to compare the exchanges also with the Eastern Mediterranean, which would be the pathway of these exchanges.

**Response:** We agree that the sentence was confusing. We have changed this sentence L. 765-766, as well as the last sentence of the abstract L. 35-36, as follows:

*"The transfer of DIC into the adjacent surface and intermediate water masses could mitigate the atmospheric CO$_2$ uptake also in the surrounding open sea of the sub-basin, and represent  up to 22 and 11%, respectively, of  the DIC exchanges with the  Atlantic Ocean at the Strait of Gibraltar."*

L. 827. In the reference of Catalano the authors should add et al. as not alL the coauthors are cited.

**Response:** We apologize for this oversight. We have corrected the reference, L. 840-842.

In the references check the subscripts for CO2.

**Response:** We have corrected the subscripts for $CO_2$ in the references, L. 801-1110.